# RNase III CLASH in MRSA uncovers sRNA regulatory networks coupling metabolism to toxin expression

Stuart W. McKellar [1], Ivayla Ivanova[1], Pedro Arede[1], Rachel L. Zapf[2], Noémie Mercier[3], Liang-Cui Chu [1], Daniel G. Mediati[4], Amy C. Pickering [5], Paul Briaud[2], Robert G. Foster [6], Grzegorz Kudla [6], J. Ross Fitzgerald [5], Isabelle Caldelari [3], Ronan K. Carroll [2,7], Jai J. Tree[4] & Sander Granneman [1]✉

Methicillin-resistant *Staphylococcus aureus* (MRSA) is a bacterial pathogen responsible for significant human morbidity and mortality. Post-transcriptional regulation by small RNAs (sRNAs) has emerged as an important mechanism for controlling virulence. However, the functionality of the majority of sRNAs during infection is unknown. To address this, we performed UV cross-linking, ligation, and sequencing of hybrids (CLASH) in MRSA to identify sRNA-RNA interactions under conditions that mimic the host environment. Using a double-stranded endoribonuclease III as bait, we uncovered hundreds of novel sRNA-RNA pairs. Strikingly, our results suggest that the production of small membrane-permeabilizing toxins is under extensive sRNA-mediated regulation and that their expression is intimately connected to metabolism. Additionally, we also uncover an sRNA sponging interaction between RsaE and RsaI. Taken together, we present a comprehensive analysis of sRNA-target interactions in MRSA and provide details on how these contribute to the control of virulence in response to changes in metabolism.

---

[1] Centre for Synthetic and Systems Biology, University of Edinburgh, Edinburgh EH9 3BF, UK. [2] Department of Biological Sciences, Ohio University, Athens, OH 45701, USA. [3] Université de Strasbourg, CNRS, Architecture et Réactivité de l'ARN, UPR9002, F-67000 Strasbourg, France. [4] School of Biotechnology and Biomolecular Sciences, University of New South Wales, Sydney 2052 NSW, Australia. [5] The Roslin Institute and Edinburgh Infectious Diseases, University of Edinburgh, Easter Bush Campus, Edinburgh, Scotland, UK. [6] MRC Human Genetics Unit, University of Edinburgh, Edinburgh EH4 2XU, UK. [7] The Infectious and Tropical Disease Institute, Ohio University, Athens, OH 45701, USA. ✉email: Sander.Granneman@ed.ac.uk

The dynamic means by which bacteria respond to stress facilitates their survival in a diverse range of environments. Survival relies on transcriptional networks whose plasticity allows bacteria to adapt their transcriptome on near-instantaneous time-scales[1]. However, it is now becoming established that effective responses are dependent upon post-transcriptional regulatory mechanisms involving RNA-binding proteins (RBPs), cis-acting riboswitches and non-coding RNAs. In particular, non-coding RNAs, termed small RNAs (sRNAs), regulate the translational efficiency and stability of targeted mRNAs and can also be linked directly to transcriptional control[2,3]. Recent research into Gram-negative bacteria such as *Escherichia coli* and *Salmonella enterica* has shown the wide variety of roles that sRNAs play in mediating adaptive processes[4–6], but our understanding of the biology of sRNAs in Gram-positive species, such as *Staphylococcus aureus*, lags far behind.

Expression of sRNAs can either be cis to the target RNA, i.e., encoded on the opposite strand, or trans at separate genomic loci. While cis-encoded sRNAs are crucial players in specific toxin-antitoxin systems in *S. aureus*[7,8], trans-encoded sRNAs have the capacity to regulate numerous RNA targets involved in separate signalling pathways. To date, around 500 transcripts have been annotated as potential sRNAs in *S. aureus*[9], however, it is unclear how many of these are genuine. Indeed, one study suggests that there are only around 50 trans-acting sRNAs that are expressed as individual transcriptional units[10].

The best characterised *S. aureus* sRNA is RNAIII, which is the main effector molecule of the quorum sensing, *agr* operon. At sufficient cellular densities, RNAIII is induced and then regulates a myriad of virulence-related targets[11]. For example, RNAIII uses its distinctive 'UCCC' seed motifs to prevent translation of the *spa, rot*, and *coa* mRNAs by binding to G-rich Shine-Dalgarno sequences[12–14]. Additionally, RNAIII is also known to stimulate translation of *hla*, encoding for the haemolytic alpha toxin, thus showing that sRNAs can also promote the translation of mRNAs. Other sRNAs with established biology are several of the Rsa family, which are known to be involved in stress responses and regulating metabolism[15–17]. For example, RsaE regulates the expression of genes involved in respiration and the TCA cycle[15–18], while RsaI is involved in sugar uptake, sugar metabolism, and biofilm formation[19].

The function of sRNAs is often mediated through RBPs that regulate or stabilise ('chaperone') the base-pairing interactions between sRNAs and their targets. Hfq and ProQ in Gram-negative species are the best studied RNA chaperones, however, the importance of Hfq in Gram-positive species is unclear while ProQ does not have a homologue (reviewed[20]). Thus, how sRNA-target interactions are regulated in Gram-positive bacteria may be mechanistically different from that in Gram-negative bacteria. Indeed, it has been hypothesised that sRNA-target interactions in *S. aureus* may involve more extensive base-pairing than those in Gram-negatives to circumvent the lack of a global chaperone[21]. Interestingly, in *S. aureus*, a multitude of proteins have been shown to bind sRNAs[21,22]. For example, duplexes between RNAIII and its targets, such as *rot, spa* and *coa*, can be targeted by endoribonuclease III (RNase III), leading to degradation of the mRNA[12–14,23–26]. Immunoprecipitation and sequencing experiments also showed that RNase III binds a large number of sRNAs[27], implying this ribonuclease plays an important role in sRNA-mediated regulation of gene expression. A brief overview of sRNA biogenesis and functionality is pictorialised in Supplementary Fig. 1.

*S. aureus* encounters a variety of different and hostile environments when it infects a host. Previous studies have implied that sRNAs could play an important role during host infection as sRNA expression levels can change significantly under infection conditions[28–30]. However, it is unclear how the vast majority of these sRNAs contribute to the infection process, underscoring the need for more detailed functional analyses. As a first step in unravelling their function, we have adapted the Cross-linking, Ligation And Sequencing of Hybrids (CLASH)[31–34] technology for *S. aureus*. To enrich for sRNA-RNA duplexes, we used RNase III as a bait protein and performed RNase III CLASH on cells in conditions mimicking the host environment. In addition to previously known sRNA-RNA interactions, we identified hundreds of novel sRNA-RNA interactions that were condition-specific, suggesting that RNase III plays a much larger role in sRNA-mediated regulation than anticipated. In addition, many mRNA-mRNA and sRNA-sRNA interactions were identified, revealing that many metabolic pathways are connected through RNA-RNA interactions.

One strategy that *S. aureus* adopts to adjust to the host environment is to express various toxins to acquire essential nutrients by lysing host cells, to kill innate immune cells, or to escape from the intracellular environment. Our data suggest that the expression of toxins is subjected to extensive sRNA-mediated regulation. A striking discovery was the regulation of the alpha phenol soluble modulin (PSMα) toxins by RsaE. Here we show that RsaE base-pairing with the *psmα* transcripts enhances the production of cytolytic toxins and increases *S. aureus'* ability to lyse erythrocytes. Furthermore, we show that the activity of RsaE in the host environment is directly controlled by another sRNA, RsaI. We demonstrate that RsaI acts as an sRNA sponge[35] and functions in concert with RNase III to inactivate RsaE activity in the bloodstream.

Taken together, our data greatly expands the repertoire of sRNA-target interactions in *S. aureus* and provides details on how these contribute to adjusting virulence in response to changes in metabolism.

## Results

**Validation of in vitro models for the transition to the bloodstream and intracellular environment**. To understand how sRNAs contribute to adaptation of *S. aureus* to the intracellular and host bloodstream environment, we performed an RNA-RNA proximity-dependent ligation method termed CLASH[32] to identify directly sRNA-RNA interactions in strains JKD6009 and USA300. As these strains differ in their evolutionary history and represent distinct clonal lineages (ST239 and ST8 respectively), we hypothesised that RNA-RNA interactions conserved across them would be more likely to be genuine or be involved in fundamental signalling pathways crucial to *S. aureus* survival.

The relatively large quantities of bacterial cells (~0.5 g) required to generate high-complexity CLASH libraries made it practically and ethically challenging to perform CLASH under physiological infection conditions. Therefore, as an alternative approach, we adopted an in vitro system to mimic two different environments encountered during host infection. Previous studies have shown that specific culture media are able to recapitulate the bloodstream and intracellular environment. A commercially available eukaryotic cell culture medium, RPMI 1640, induces a similar transcriptomic response in *S. aureus* as human blood plasma[29]. Most importantly, the lack of iron in this medium induces a strong upregulation in iron-responsive genes, a phenomenon also observed in human plasma[29]. Additionally, low phosphate, low magnesium (LPM) media was designed to have a similar salt composition as eukaryotic cytoplasm and has been used in various bacterial infection studies[36–41]. We therefore utilised LPM media at pH 7.6 to examine the nutritional adaptations to the intracellular environment. Finally, LPM at

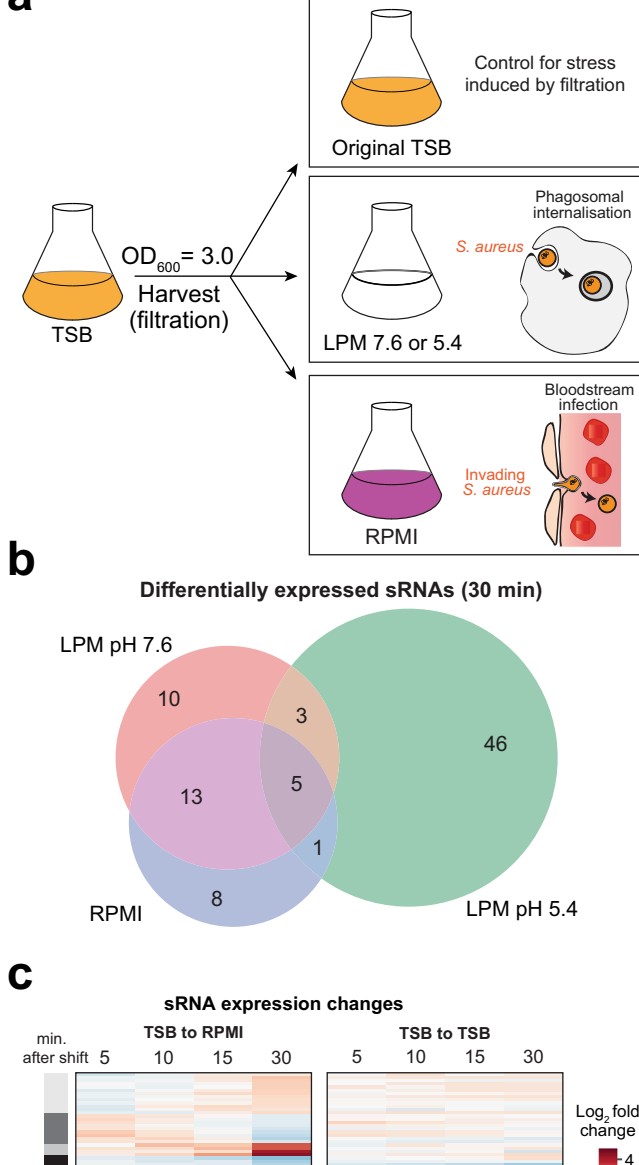

**Fig. 1 Mimicking the transition to the bloodstream and intracellular environment. a** Experimental set up for shift experiments. *S. aureus* was grown to OD$_{600}$ ~3 in TSB and harvested through vacuum filtration. Cells were then resuspended in RPMI to simulate the bloodstream or LPM pH 5.4 to simulate intracellular phagosomes. As controls, cells were shifted into LPM pH 7.6 to model for the nutritional profile of this medium, or back into their original TSB to model for any stresses incurred by the shift. **b** Number of sRNAs with significantly changed expression after 30 min of stress. Each stress condition is uniquely coloured, with sRNAs found in LPM pH 7.6 shown in red, LPM pH 5.4 in green and RPMI in blue. **c** Changes in the expression levels of individual sRNAs during the shift to RPMI and shift back to the original TSB medium. The darker the red in colour, the higher the increase in gene expression. The darker the blue in colour, the stronger the reduction in gene expression. Expression was compared to data obtained from the TSB (t = 0) sample. Only sRNAs that were differentially expressed during the time-course, according to DESeq2[116], are shown.

pH 5.4 was used to investigate the response to acidic stress which is encountered in cellular compartments such as phagolysosomes.

As mRNA half-lives in bacteria can vary from seconds to minutes[42], we focused on the initial phases of stress adaptation. We utilised a novel cell harvesting device[43,44] that facilitates the transfer of cells from one medium to another in under one minute, enabling stress adaptation studies at high temporal resolution. We grew *S. aureus* in tryptic soya broth (TSB) to post-exponential phase (OD$_{600}$ ~3.0) to induce expression of virulence genes and then rapidly transferred the cells to either RPMI, LPM pH 7.6 or LPM pH 5.4 media (Fig. 1a). We used RNAtag-Seq[45] to examine the transcriptomic response of *S. aureus* to these media under our experimental conditions. We took samples 5, 10, 15 and 30 minutes after the shift to the new media. A very high correlation between replicate experiments was observed (Supplementary Fig. 2), demonstrating the reproducibility of the results. We observed dramatic and dynamic changes in sRNA gene expression (see example in Fig. 1b, c), the majority of which were specific to the infection-mimicking media. To control for mechanical stresses induced by the rapid vacuum filtration, we shifted *S. aureus* back into their original TSB medium.

Data from these TSB control cells imply that the rapid filtration has minimal impact on gene expression as most genes displayed a linear expression pattern (i.e., either continued to rise or continued to fall) after the shift back to TSB. However, the cells shifted to the stressful media showed a markedly different gene expression behaviour (Fig. 1c and Supplementary Fig. 3), suggesting that the induced changes in gene expression are largely due to the changes in media composition.

Although our RNA-seq analyses were carried out at very early time-points following the shift from TSB, we already observed transcriptomic changes that were previously detected after hours of growth in similar media. For example, after shifting to RPMI, we observed upregulation in iron-related transcripts, including members of the *isd*, *feu*, *fhu* and *sir* operons (Supplementary Fig. 4a)[28,29]. Additionally, previous work studying transcriptome changes in *S. aureus* in response to blood and serum exposure identified upregulation of amino acid metabolism, immune evasion proteins, virulence factors and transcripts involved in iron acquisition[28], some of which were recapitulated in our RPMI shifts (Supplementary Fig. 4a, b). With regards to LPM pH 5.4, we observed strong upregulation in many amino acid biosynthesis pathways (Supplementary Fig. 4c), which matches previous observations of *S. aureus* internalised into human macrophages and epithelial cells[29]. Further evidence of metabolic remodelling was also observed with upregulation in TCA cycle members. The gamma haemolysin cytolytic toxins (*hlgA*, *hlgB* and *hlgC*), which are also highly upregulated in human blood, have been hypothesised to play a role in *S. aureus* escape from internalisation of polymorphonuclear granulocytes[28]. Interestingly, we observed rapid (within 5 min) upregulation of transcripts encoding these toxins in LPM pH 7.6, whereas the induction in LPM pH 5.4 and RPMI medium was more modest (Supplementary Fig. 4d). We conclude that our shift experiments recapitulate key aspects of the human bloodstream and intracellular environments, facilitating their use as model systems.

### RNase III CLASH robustly detects RNA-RNA interactions in *S. aureus*. To be able to effectively apply the CLASH method to Gram-positive bacteria, the cell lysis and affinity purification steps required optimisation (see Supplementary Fig. 5 and Supplementary Data for further details). Because the available evidence suggests that Hfq is unlikely to play a major role in chaperoning sRNA-target interactions in *S. aureus*, the RNase III protein was

used as a bait to capture sRNA-target duplexes. To purify the protein, we generated a strain in which RNase III was fused to a HIS6-TEV-3xFLAG (HTF) tag at its C-terminus. Subsequent CLASH experiments were performed in two different methicillin-resistant *S. aureus* strains; USA300 LAC, a representative of sequence type (ST) 8, and JKD6009, an ST239 clone. For USA300, we performed CLASH on cells grown in TSB and after shifting to RPMI, LPM pH 7.6 and LPM pH 5.4 medium. For JKD6009, we only performed CLASH in the TSB and RPMI growth conditions (Supplementary Fig. 5c). Because the bulk of gene expression changes were already detected 15 min after the shift, CLASH experiments were performed on samples harvested at this timepoint. Sequencing data from 3 or 4 independent biological replicates were merged and significantly enriched interactions were identified using hyb[46] combined with a probabilistic pipeline[47], as previously described[33,34].

Analysis of the USA300 data was challenging as the quality of the annotation of this genome (USA300 FPR3757) was not equal to that of JKD6009. To improve the annotation of USA300, we used the Rockhopper software[48] on our RNA-seq data to map untranslated regions (UTRs) and identify novel transcripts. Our updated genome annotation is included in the accompanying Gene Expression Omnibus deposition. This revealed that many annotated sRNAs overlapped with UTRs, as was observed previously in other strains[10]. Because it was unclear whether these UTRs harboured genuine sRNAs, we focused our analyses on sRNAs that are transcribed as independent transcriptional units[10]. For convenience, we hereafter refer to these bona fide sRNAs as "bf sRNAs". The complete list of all sRNA target interactions can be found in Supplementary Data 3 and 4, and further detailed in the included Supplementary Data[49–51] and Figures documentation.

Many of the predicted sRNA-target interactions in USA300 had poorer folding energies compared to chimeras identified in JKD6009 and experimentally verified *S. aureus* sRNA-mRNA interactions captured through CLASH (Fig. 2a). Manual inspection of the data revealed that many of the USA300 chimeras with poor folding energies consisted mainly of sequences with low GC content (Fig. 2b) that mapped to multiple annotated features in the USA300 genome. These were therefore likely incorrectly assigned as intermolecular interactions. As a result, we only considered interactions that contained a bf sRNA and had a minimum folding energy (MFE) equal or smaller than -10 kcal/mol, which removed most of these likely false-positive interactions. The remaining sRNA-target interactions had GC contents and folding energies closer to that of captured known interactions, which acted as positive controls (Fig. 2a, b).

Because RNase III is an endonuclease that cleaves double-stranded RNA substrates generally consisting of relatively long stem structures interrupted by few bulges[52], we reasoned that interactions obtained from CLASH should have a strong folding potential and contain such structures. To test this, we used RNADuplex[53] to compute the hybridization potential (in kcal/mol) of each half of the filtered chimeras (Fig. 2c). This showed that the data were statistically significantly enriched for structured RNAs compared to randomised, shuffled data. Moreover, RNA structural motif analyses revealed that the filtered chimeras were highly enriched for structures with long stems and only a few single nucleotide bulges (Fig. 2d), fitting the established mode of RNase III binding to its targets[54] and giving further credibility to identified interactions.

Overall, we obtained thousands of unique hybrids in the RNase III-HTF data (Fig. 2e). Within each experimental condition, we also detected hundreds of fragments containing bf sRNAs (Fig. 2f). Very few chimeras were detected in the CLASH data from the parental strains, suggesting that those interactions

detected through RNase III CLASH are specific. As such, we conclude that RNase III CLASH effectively captures RNA-RNA duplexes in *S. aureus*.

The types of interactions obtained were overall similarly abundant across the different strains and conditions (Fig. 3a). Notably, we obtained 721 hybrids between mRNAs and transcripts antisense to mRNAs (mRNA$_{AS}$), agreeing with previous RNase III RIP-seq experiments that identified mRNA$_{AS}$ transcripts as major targets of RNase III[27]. Of these mRNA-mRNA$_{AS}$ interactions, 72% (JKD6009) and 53% (USA300) were between cognate RNAs (Supplementary Fig. 6a); this confirms that we can capture significant numbers of canonical mRNA-mRNA$_{AS}$ interactions but also raises the interesting possibility of cis-encoded mRNA$_{AS}$ transcripts regulating distinct targets in trans. We also recovered a large number of sRNA-mRNA interactions (~11% of each condition, on average), consistent with established RNase III biology[14,55–57]. Of note, an average of 7% of the interactions were between sRNAs and other sRNAs (Fig. 3a). Examples of such interactions have recently been identified in *S. aureus*[17,19,58]. Finally, we also recovered a significant amount of mRNA–mRNA interactions. Given that UTRs are known to be a source of trans-acting regulatory RNAs in other bacterial species[5,59,60], these may contain such examples. Consistent with this idea, Mediati et al.[61] identified a long 3′ UTR in *S. aureus* that is not processed from the mRNA but functions as a non-coding RNA in regulating vancomycin resistance. We conclude that we recovered examples of all known RNase III target categories.

Overall, we identified 42 bf sRNA-target interactions in USA300, represented by 855 unique hybrids, and 48 bf sRNA-target interactions in JKD6009, represented by 1689 unique hybrids. These were primarily between sRNAs and mRNAs, but a noteworthy number of sRNA-sRNA interactions were also identified (Fig. 3b). As such, we conclude that RNase III recognises many bf sRNA-target duplexes. The interactome is visualised in Supplementary Fig. 7.

We reasoned that bf sRNAs predicted to base-pair at or near the mRNA Shine-Dalgarno (SD) sequence and/or start codon would most likely have an impact on the mRNA or protein steady-state levels. Indeed, a large number of our bf sRNA-mRNA interactions (Fig. 3c) included these ribosome binding sites, implying a canonical mode of sRNA-mediated regulation[62]. We also found examples of bf sRNAs base-pairing with the coding sequences as well as the extreme 3′ end of transcripts involved in operons (Supplementary Fig. 6b). This included the RNAIII-*murQ* interaction that was also detected by Mediati et al.[61] and was shown to be a functional interaction.

To assess the quality of our CLASH data, we firstly looked for experimentally verified interactions (Supplementary Fig. 8). Several type I toxin-antitoxin systems are well characterised in *S. aureus*, where an unstable antisense RNA represses the translation of a more stable, toxic mRNA. The best characterised of these is between *sprA1*/SprA1$_{AS}$, and *sprA* has been identified as an RNase III target through RIP-seq experiments[7,8,27]. These interactions were by far the most abundant in the data and were detected in all strains and conditions tested. Additionally, the in silico folded structures of the corresponding chimeric reads are consistent with the published literature. We also identified several known sRNA-mRNA interactions, including RsaA-*mgrA*; RsaA-*HG001_01977* (annotated here as SAA6008_01954 and SAUSA300_1921); RsaE-*opp3B*; RsaE-*purH* and SprX-*spoVG* (Supplementary Fig. 8). Again, the predicted structure of the chimera halves was consistent with published literature[15,17,26,63]. A detailed description of these verified interactions is provided and discussed in the Supplementary Data. Overall, we conclude that RNase III CLASH reliably detects sRNA-mRNA interactions in *S. aureus*.

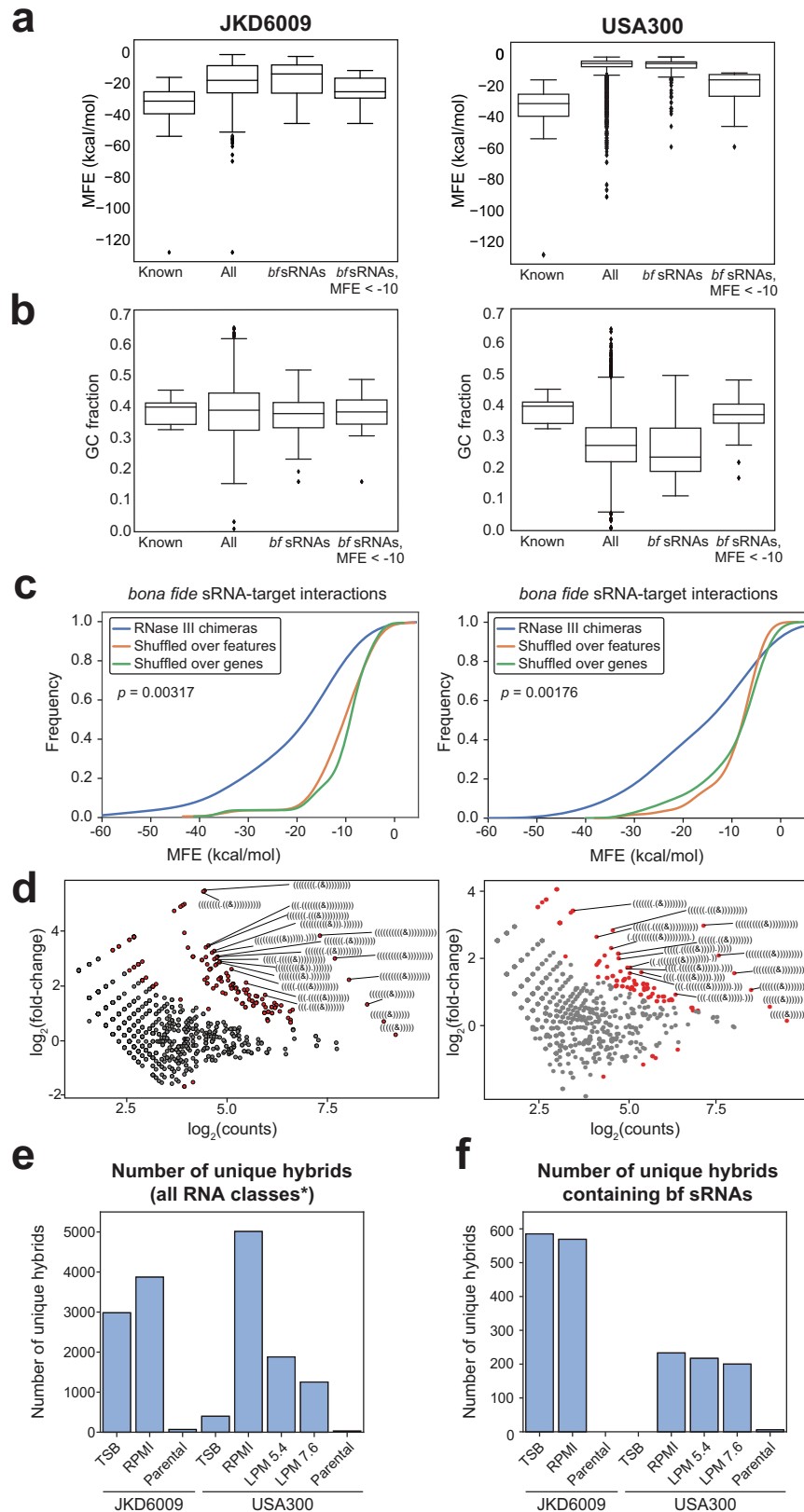

**RsaI directly and specifically binds RsaE in vitro**. We identified six unique sRNA-sRNA interactions using RNase III CLASH. Two stood out due to their being represented by a relatively large number of hybrids; RsaI-RsaE and RsaA-RNAIII. We detected interactions between RsaA-RNAIII in JKD6009 in TSB and RPMI (Supplementary Fig. 9a, b) and RsaI-RsaE interactions could be detected in both strains, but primarily in the RPMI, LPM 7.4 and LPM 5.4 media (Supplementary Fig. 10a). We were able to demonstrate specific binding between RsaA and RNAIII in vitro (Supplementary Fig. 9c and d). However, we did not pursue this interaction further as its functional significance was unclear (see Supplementary Fig. 9e–g and Supplementary Data for a description of the results).

**Fig. 2 Folding and structural analyses of hybrids identified through RNase III CLASH. a** Boxplots showing the minimum folding energy (MFE) of identified RNA-RNA interactions in all independent RNase III CLASH datasets ($n = 4$). Plotted are the previously verified interactions ("Known") captured through CLASH; all the interactions identified by CLASH; only those containing a bona fide sRNA (bf sRNAs); and those containing a bona fide sRNA and filtered for MFE < -10 kcal/mol. The boxplot extends from first to the third quartile values of the data, with a line at the median. The whiskers extend from the edges of box to show the interquartile range multiplied by 1.5. Outliers are plotted as separate dots. **b** As in **a** but with reference to GC content. **c** Cumulative distribution of the MFE of the filtered interactions involving bona fide sRNAs in JKD6009 (left) and USA300 (right). Folding energies were calculated using RNADuplex[53]. As controls, interactions were shuffled randomly against other partners of the same class (orange line) or randomly across the gene (green line) of their partner identified through CLASH. Significance was tested with the Kolmgorov-Smirnov test. **d** Enriched structural motifs (red dots) in hybrids identified through RNase III CLASH. The incidence of each structure generated by RNADuplex was counted and compared to interactions randomly shuffled against different partners. Significance was calculated using a one-sided Fisher's exact test and Benjamini-Hochberg correction was applied to account for multiple tests. The x-axis displays the total number of counts for each structure in the data, whereas the y-axis indicates the log₂-fold difference for each structure between the experimental data and the randomly shuffled data. **e** Total number of unique hybrids identified in each experimental condition. The parental controls for each strain were merged. *tRNA-tRNA and rRNA-rRNA chimeras were excluded due to their high sequence similarity, meaning that we could not unambiguously determine if these represented intermolecular or intramolecular interactions. **f** As in **e**, but with respect to hybrids involving a bf sRNA.

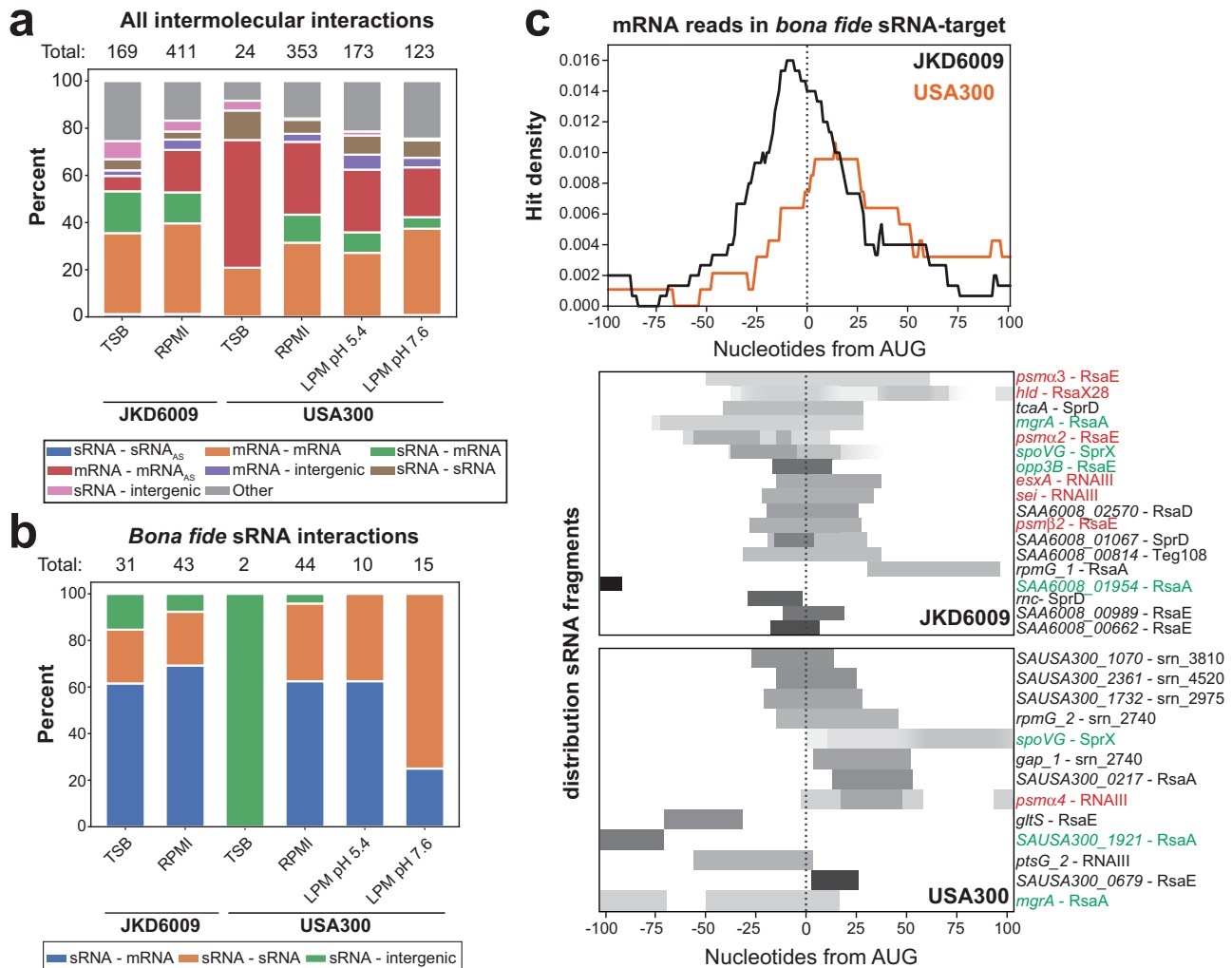

**Fig. 3 RNase III CLASH captures canonical sRNA – mRNA interactions. a** Categories of identified intermolecular RNA interactions. Here, a single interaction can be represented by many hybrids. tRNA-tRNA and rRNA-rRNA chimeras were excluded due to their high sequence similarity, meaning that we could not unambiguously determine if these represented intermolecular or intramolecular interactions. "Total" indicates the total number of unique RNA-RNA interactions identified in each dataset. Colour of each stacked bar denotes the type of interaction represented. **b** As in **a**, but only for interactions containing a bona fide sRNA. **c** Top: distribution of the mRNA fragments in bf sRNA-mRNA interactions around the translational start codon (AUG). Orange line shows data from USA300 and the black line from JKD6009. Bottom: heatmaps showing the read distribution for the mRNA fragments for each individual interaction. Interactions highlighted in green are those that have previously been experimentally verified, and in red are those interactions involving toxins. Interactions coloured in black are other novel interactions identified in this study.

The interaction between RsaI and the highly conserved RsaE (RoxS in *Bacillus subtilis*) has previously been proposed in Gram-positive bacteria[17,19,64]. However, precisely how these sRNAs base-pair and the functionality of this interaction was unclear. RsaE is characterized by a sequence duplication comprising the characteristic UCCCC seed motif. Our CLASH data imply that the two G-rich regions of RsaI base-pair with the 5′ and the 3′ UCCCC motifs of RsaE when cells are transferred to RPMI and LPM medium (Supplementary Fig. 10b). The reason we primarily detected the RsaI-RsaE interaction in RPMI and LPM media could be explained by the expression levels of the two sRNAs under these stress conditions; RsaI was rapidly upregulated in response to RPMI and LPM exposure but not in TSB (Supplementary Fig. 10c). RsaE remained relatively stable in the TSB control and RPMI, while decreasing in the two LPM media.

We performed electrophoretic mobility shift assays (EMSAs) with (mutant) RsaI and RsaE transcripts (Supplementary Fig. 11). This revealed that the 3′ UCCCC motif in RsaE is essential for binding RsaI in vitro (Supplementary Fig. 11a). Furthermore, these data confirmed that both GGGG motifs in RsaI are involved in the base-pairing interactions as only mutations in both G-tracts completely disrupted the interaction with RsaE (Supplementary Fig. 11b; RsaI mut 3). However, base-pairing could be partially restored with this G-tract mutant when compensatory mutations were made in RsaE (Supplementary Fig. 11c; RsaE mut 1 and RsaI mut 3).

Because the RsaE-RsaI interaction was recovered using RNase III as bait, it is logical to assume that this sRNA-sRNA duplex is a substrate for RNase III-mediated cleavage. To test this possibility, we 5′-end radiolabelled RsaE or RsaI, mixed them respectively with cold RsaI or RsaE and increasing concentrations of recombinant RNase III in presence of $Mg^{2+}$ (enzyme activator) or $Ca^{2+}$ (enzyme inhibitor). Consistent with this idea, we found that RNase III specifically cleaves a fraction of RsaE and RsaI at a single position but only when they formed a duplex in vitro (Supplementary Fig. 12a, b). However, RNase III cleavage was only detected in the stem where the 5′ UCCCC motif of RsaE is base-paired to the 3′ GGGG motif of RsaI (Supplementary Fig. 12c). We conclude that RsaE specifically base-pairs with RsaI in vitro and that this duplex can be cleaved by RNase III.

**RsaI and RsaE are trimmed at the 3′ end**. Because RsaE regulates multiple metabolic pathways[15–17] and RsaI plays a role in glucose starvation[19], we hypothesised that this interaction may mediate metabolic remodelling when entering the host environment. This led us to a model where RsaI is upregulated during stress and then base-pairs with RsaE to inhibit RsaE's regulation of its mRNA targets. Given that RsaI's interaction with RsaE was primarily detected under host infection conditions and that RsaI has been proposed to play a role in the infection process[11], we decided to test this hypothesis during growth in human serum to better mimic physiologically-relevant infectious conditions. Northern blot analyses showed that RsaE species slightly longer than 100 nt could be detected in human serum, which is approximately the expected length (102 nt; Fig. 4a, lane 2). However, in TSB, shorter RsaE species accumulated, indicating that in rich medium RsaE is processed (Fig. 4a, lane 1). Similarly, we found that RsaI accumulates as slightly shorter species in human serum (Fig. 4a, lane 2).

In addition to the detection of alternative sRNA species, quantifying the total levels of all forms of RsaE revealed that these were significantly lower in human serum compared to TSB, implying RsaE downregulation in this environment (Fig. 4a, lanes 1 and 2; Fig. 4b; Supplementary Fig. 13a, lanes 1). In opposition to this, all forms of RsaI were comparable in TSB and human

serum (Fig. 4a, lanes 1 and 2; Fig. 4b). As a result of this regulation, RsaI levels in human serum are therefore increased relative to RsaE (Fig. 4a, b). This is an important point as their relative expression levels likely determines the regulatory impact.

To gain more insights into the processing of RsaE and RsaI, we mapped the extremities of these molecules in TSB and human serum using Nanopore sequencing (Fig. 4c, d). This revealed that both RsaE and RsaI undergo trimming of the U-tract of the transcription terminator, resulting in heterogenous RNA sub-species of varying lengths. This was particularly evident for RsaE in TSB, mirroring the northern blot data. We found that full-length RsaE has an 8-nucleotide terminator sequence composed of 7 Us and a terminal A (UUUUUUUA). In human serum, RsaE is observed primarily as the full-length product or with a two-nucleotide trim. In comparison, RsaE in TSB is only very rarely present as the full length, and instead exhibits a variety of trimmed subspecies containing terminators between 2 and 6 nucleotides in length with the most abundant being 4 (Fig. 4c). Overall, we conclude from the northern blot data and Nanopore sequencing that RsaE undergoes 3′-end trimming in TSB, with full length (or only a small degrees of trimming) RsaE observed in human serum.

Like RsaE, RsaI shows differential 3′-end trimming in TSB and human serum. We found RsaI to exhibit a terminator of 7 Us. In TSB, terminator lengths between 2 and 5 Us were most prominent, although the full-length subspecies was also observed at significant levels. In comparison, RsaI in human serum was most prominently 4 Us in length, although other subspecies were also observed (Fig. 4d).

**RsaI and RNase III primarily regulate RsaE activity, not stability, under host infection conditions**. Because RsaI directly binds RsaE and this duplex can be cleaved by RNase III in vitro, we next asked whether these molecules regulate the stability of RsaE in human serum. Deleting RsaI (*ΔrsaI*) or RNase III (*Δrnc*) in USA300 did not significantly influence RsaE processing or steady state levels in TSB or human serum (Fig. 4a, b, lanes 3–6). This was also observed in a double *ΔrsaI-Δrnc* mutant (Supplementary Fig. 13a, lanes 1–4). We conclude that RsaI and RNase III do not significantly impact RsaE stability.

Do RsaI and RNase III then regulate RsaE activity? Deleting RsaI or RNase III led to a strong and significant reduction in several tested RsaE targets, but only when the cells were grown in human serum (Fig. 5a). This implies that deleting RsaI or RNase III results in increased RsaE activity under infection-mimicking conditions. The abundance of RsaE target mRNAs in the WT strain did not differ significantly between TSB and human serum (Supplementary Fig. 13b), further supporting the conclusion that the decrease in target mRNA stability observed in the *ΔrsaI* and *Δrnc* strains is due to the removal of RsaI and RNase III expression respectively, and not the effect of the growth in human serum. Overall, these data suggest that binding of RNase III to the RsaI-RsaE duplex is necessary and sufficient to regulate the levels of RsaE targets in human serum.

The observation that the RsaI and RNase III-dependent regulation of RsaE targets was only observed in human serum implies that high levels of RsaI relative to RsaE are required to regulate RsaE targets. To test this hypothesis, we overexpressed RsaI in cells grown in TSB. Consistent with our predictions, very high levels of RsaI (Fig. 5b, lanes 3 and 4, quantified in Fig. 5c) increased the expression of RsaE targets (Fig. 5d). This was independent of the presence of RNase III as equivalent results were obtained when RsaI was overexpressed in the *Δrnc* strain (Fig. 5b–d).

Vice versa, to examine if RsaE can regulate RsaI, we overexpressed RsaE in the WT and *Δrnc* background and

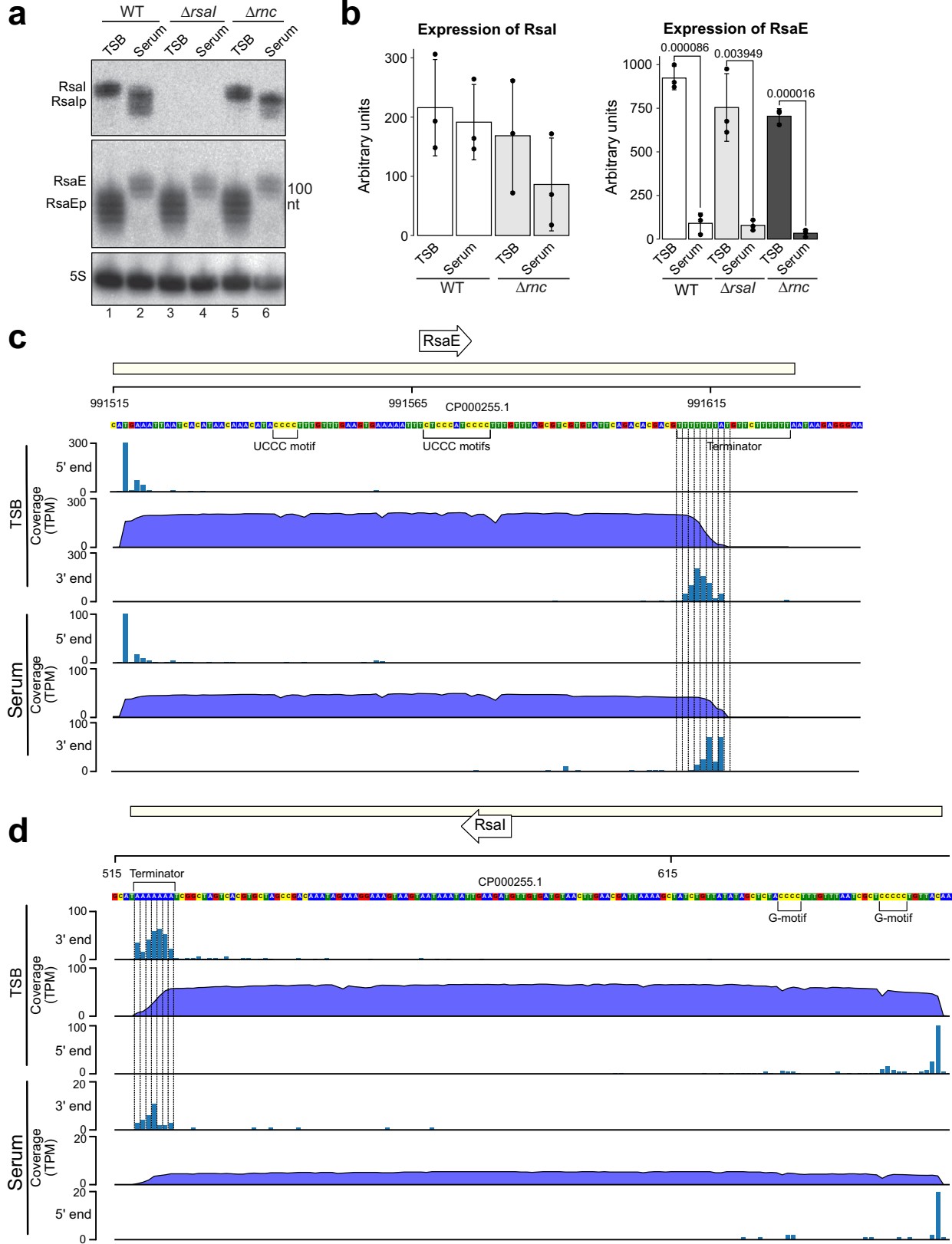

examined the stability of RsaI. First, RsaE was overexpressed using the constitutive *amiA* promoter. No significant changes in RsaI levels were observed in TSB or human serum conditions (Supplementary Fig. 13c, compare lanes 3, 4, 7 and 8 with lanes 1 and 2 and the quantification of the results below the image). However, there was only a modest overexpression of RsaE using

this constitutive promoter, indicating that RsaE levels are tightly controlled. Supporting this idea, previous attempts to overexpress RsaE have resulted in decreased cell viability in rich media conditions[15]. We therefore decided to use a previously described tetracycline-inducible RsaE construct[17]. After a 15-min induction, RsaE levels were substantially upregulated in TSB

**Fig. 4 RsaI and RsaE medium-dependant expression and processing. a** Northern blot analysis of USA300 parental (WT), Δ*rsaI* and Δ*rnc* strains grown in TSB and human serum. Cells were grown to exponential phase in TSB, diluted in 10 mL of human serum to OD$_{600}$ 0.05 and grown for another 3 h. RsaIp and RsaEp indicates processed forms of the corresponding sRNAs. Three independent biological replicate experiments were performed, with a representative experiment shown here. **b** Quantification of the RsaE and RsaI levels represented in **a**. Arbitrary units indicate the signal intensities of the bands as measured by the Fuji AIDA software. Shown are the averages and standard deviation calculated from three independent replicate experiments. Values above bars display respective *p* value, obtained from Student's unpaired, two-tailed *t* test. Images and raw data used to generate figures (**a**, **b**) are provided in the Source Data file. **c** Genome browser visualisation of RsaE and its mapped 5′- and 3′-ends, Total RNA was extracted from cells grown in TSB or human serum and then the exact sequence of RsaE was identified through Nanopore cDNA sequencing. Reads are expressed as transcripts per million (TPM). DNA nucleotides are coloured, A in blue, T in green, C in yellow and G in red. **d** As in **c** but for RsaI.

(Supplementary Fig. 13d, lanes 3, 7). However, the cell was still able to strongly repress RsaE overexpression in serum (Supplementary Fig. 13d, lanes 4, 8). Note that RsaE expressed from this inducible plasmid accumulates as a longer species in USA300. The fact that we observed strong suppression of RsaE expression in human serum using two different promoters suggests that post-transcriptional regulation plays an important role in suppressing RsaE levels under these conditions. Despite this large increase in RsaE in TSB, we could not detect significant changes in RsaI levels (Supplementary Fig. 13d, lanes 3 and 7) or RsaE mRNA targets (Supplementary Fig 13e). It therefore seems that we were unable to express RsaE to sufficiently high levels in USA300 to impose changes on these interacting RNAs, possibly because RsaE levels are very tightly controlled.

Collectively, these data suggest that when RsaI is expressed at sufficiently high levels relative to RsaE in human serum, RsaI can sponge RsaE and prevent it from downregulating its targets in an RNase III-dependent manner. However, this dependency on RNase III can be overcome by expressing very high levels of RsaI. We therefore hypothesise that the main function of RNase III here is to act as an RNA chaperone by stabilising the RsaI-RsaE base-pairing interactions.

**Toxin expression in *S. aureus* is under extensive sRNA-mediated regulation.** Strikingly, we identified many sRNAs fused to fragments of toxin-encoding mRNAs, including phenol-soluble modulins (PSMs) (Fig. 6a, Supplementary Fig. 14). In JKD6009, we identified RsaE interacting with members of both the alpha and beta PSMs (α/βPSMs), *psmα2, psmα3* and *psmβ2*, during growth in TSB (Supplementary Fig. 7). These cytolytic peptides[65] are crucial for *S. aureus* virulence[66] through inducing blood cell lysis[66,67], phagosomal escape[68,69] and detachment from biofilms[70,71]. The stability of the αPSM operon RNA transcript is known to be regulated by the sRNA Teg41, which is predicted to bind within the coding sequence of *psmα4*[72]. Our CLASH data indicate that RsaE uses its 3′ UCCCC motif to base-pair with the Shine-Dalgarno (SD) sequences of *psmα2* and *psmα3* mRNAs (Fig. 6b and Supplementary Fig. 14a). However, similar base-pairing interactions can also be drawn with the SD sequences of the *psmα1* and *psmα4* transcripts and with the 5′ UCCCC motif of RsaE (Fig. 6c).

We also found an interaction between RNAIII and *psmα4* in both JKD6009 and USA300, primarily found after the shift to both LPM media (Fig. 6a, Supplementary Fig. 14b). In silico analyses predict that RNAIII binds to a sequence in *psmα4* that normally sequesters the SD sequence. This suggests RNAIII could stimulate PSMα4 production by liberating the SD sequence from a stem structure, making it more accessible to ribosomes, in a similar way as it regulates α-toxin[73].

Finally, we also identified sRNA-toxin interactions with the PSMs. We identified an interaction between helix 9 of RNAIII and *esxA* (Fig. 6a), which is important for the intracellular survival of *S. aureus* in infected epithelial cells through interfering with apoptosis[74]. Finally, after shifting JKD6009 to RPMI, we

identified RsaX28 interacting with the *hld* coding sequence of RNAIII, and RNAIII interacting with *sei*, encoding for enterotoxin I (Supplementary Fig. 14c). Thus, the expression of *S. aureus* toxins appears to be under extensive sRNA-mediated regulation.

**RsaE regulates haemolytic activity by enhancing phenol-soluble modulin production.** Given RsaE's roles in metabolic regulation, we hypothesised that regulation of PSMs by RsaE could represent a direct link between metabolism and virulence at the post-transcriptional level. Because the αPSMs (in particular PSMα3) display greater cytotoxicity than the βPSMs[75] we decided to further characterize the predicted interaction between RsaE and αPSMs. Consistent with our CLASH data, our EMSAs showed that RsaE can interact with *psmα3* in vitro, although complex formation is inefficient (Fig. 6d). This suggests that a chaperone may be required to stabilize these duplexes. Nevertheless, the interaction was specific as mutations in RsaE's C-rich motifs abrogated duplex formation. Interestingly, although RsaE's two UCCCC motifs can act independently to regulate several mRNA targets[17], we found that mutation of just the 3′ UCCCC motif can completely abolish binding to *psmα3*. However, we were unable to verify the predicted interactions between RsaE and *psmα1* and *psmα4* by EMSA.

The observation from CLASH that RsaE interacts with the SD sequences of the *psmα2* and *psmα3* suggests that RsaE base-pairing would inhibit the translation of these toxins as it would block the association of the 30 S ribosomal subunit. To address this, we attempted to validate these interactions in *S. aureus* using a GFP-reporter assay[76]. Here, the RBS and a portion of the coding sequence of each PSM was fused to GFP. However, none of the *psmα*-GFP fusions were expressed at sufficiently high levels in vivo. Therefore, as an alternative approach for testing the functional significance of this interaction in vivo, we investigated the role of RsaE in regulating cytotoxic activity. We performed these validation experiments in the USA300 background as this strain is known to secrete high levels of PSMs compared to other clinical strains[77]. As the secreted αPSMs are involved in the lysis of host blood cells[66,67,78], we reasoned that deleting RsaE would increase haemolytic activity, whereas overexpression should decrease it. Culture supernatants were incubated with whole human blood and the degree of lysis was measured by optical absorbency.

Much to our surprise, overexpression of RsaE using the constitutive *amiA* promoter resulted in a 1.5 to 2-fold increase in haemolytic activity, whereas deletion of RsaE reduced haemolytic activity by approximately 40% relative to the wild type (Fig. 6e). The defect in haemolytic activity of the Δ*rsaE* strain could be restored (and even increased) by overexpressing RsaE from a plasmid (Fig. 6e). Identical results were obtained with butanol extracts of culture supernatants, which enriches for PSMs[72] (Fig. 6f). As we uncovered this interaction using RNase III as a bait protein for CLASH, we also tested the haemolytic activity in

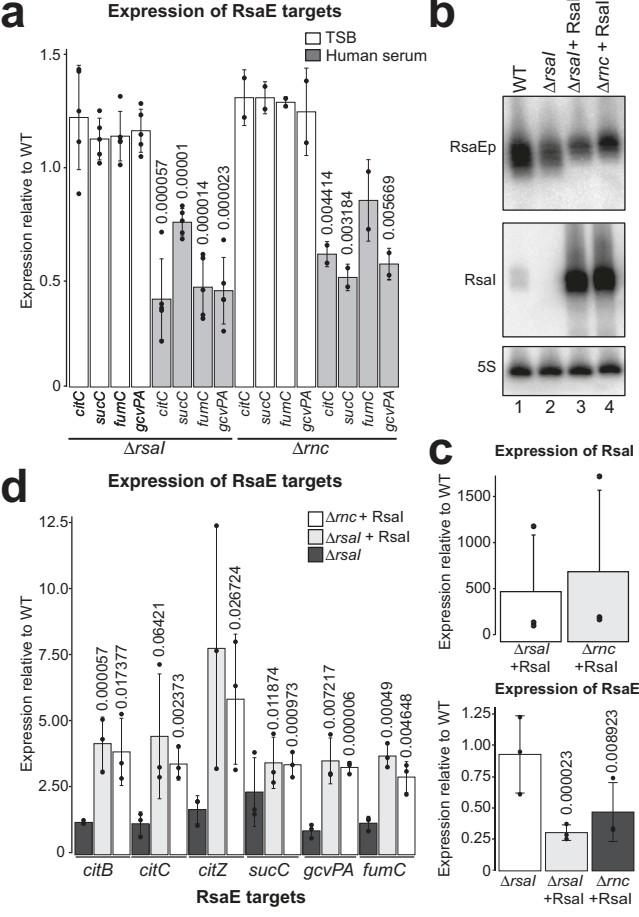

**Fig. 5 RsaI and RNase III regulate RsaE activity in human serum. a** RsaI and RNase III regulate RsaE activity in human serum. Shown is the expression of RsaE targets in TSB and human serum in the Δ*rsaI* and Δ*rnc* strains relative to the parental strain (USA300) as measured by RT-qPCR. Averages and standard deviation were calculated from five (Δ*rsaI*) or two (Δ*rnc*) experimental replicates and three technical replicates. Values above bars display respective *p* value (mutant vs WT), obtained from Student's two-tailed, unpaired *t* test The bar fill denotes the condition tested, white for TSB, grey for human serum. **b** Overexpression of RsaI and impact on the levels of RsaE. Strains (WT, Δ*rsaI* and Δ*rsaI* complemented with a plasmid expressing RsaI) were grown to OD$_{600}$ ~3.0 in TSB and RNA was analysed by Northern blot analyses to detect RsaE, RsaI and 5 S rRNA. RsaEp indicates processed RsaE. Shown is a representative Northern blot results from three independent experiments. **c** Quantification of RsaI and RsaE Northern blot results described in **b**. Shown are the averages and standard deviations calculated from three independent experiments. **d** Expression of RsaE targets in Δ*rnc* and Δ*rsaI* strain complemented with a plasmid expressing RsaI. Shown are the averages and standard deviations calculated from three independent experiments. Values are represented as relative to the WT. Values above bars display respective *p* value (mutant vs WT), obtained using Student's two-tailed, unpaired *t* test. The bar fill denotes the tested strain; dark grey for Δ*rsaI*, light grey for Δ*rsaI* pICS3:RsaI and white for Δ*rnc* pICS3:RsaI. Images and raw data used to generate these results are provided in the Source Data file.

an RNase III deletion mutant (Δ*rnc*). Deleting RNase III almost completely abolished haemolytic activity (Fig. 6e).

Finally, we also performed label-free quantitative mass-spectrometry on culture supernatants from Δ*rsaE* and Δ*rsaE* complemented with RsaE on a plasmid (triplicate experiments; Supplementary Fig. 15a). This showed that deleting RsaE substantially reduced the expression of PSMα1 and PSMα4 levels

in Δ*rsaE* culture supernatants (Supplementary Fig. 15b-c). We identified two unique PSMα2 peptides in our data, however, the intensities were too low to be able to do a reliable quantification. The level of PSMα3 and βPSM toxins were not significantly affected, whereas reduced δ-toxin levels were found in supernatants of the Δ*rsaE* strain (Supplementary Fig. 15b). Importantly, toxin production was restored to roughly wild-type levels when RsaE was reintroduced in the Δ*rsaE* strain from a plasmid (Supplementary Fig. 15d).

Taken together, these data imply that RsaE positively regulates the expression of αPSM-mediated haemolysis. We propose that RsaE acts to couple metabolic pathways to virulence through post-transcriptional regulation of its target mRNAs.

**RNAIII is required for optimal *esxA* toxin production**. We applied numerous approaches to validate the interaction identified between helix 9 of RNAIII and *esxA* (Fig. 7a). Firstly, we recapitulated the interaction in vitro using an EMSA. The CLASH data suggested that RNAIII's ninth helix makes extensive base-pairing interactions with the *esxA* coding sequence just after the SD sequence (Fig. 7a); our EMSAs demonstrated that the interaction was specific to this region of RNAIII (Fig. 7b, c).

We were also able to confirm the interaction in vivo using a GFP reporter assay[76]. Here, a fragment of the *esxA* 5′-UTR and coding sequenced was fused to GFP and expressed constitutively from a plasmid using the *amiA* promoter. The impact of ectopic RNAIII expression on the translation of *esxA* could then be inferred by fluorescence. Importantly, this system uncouples *esxA* expression from its endogenous promoter, thus facilitating interrogation of only post-transcriptional effects of RNAIII. RNAIII overexpression led to a ~12-fold increase in expression of an EsxA-sfGFP fluorescent reporter, indicating that, as with α-haemolysin[73], RNAIII base-pairing enhances *esxA* translation (Fig. 7d). We also created an *esxA* seed mutant in which each G or C nucleotide predicted to interact with RNAIII was complemented. Although this *esxA mutant-sfGFP* construct was less stable than the wild type, expressing an RNAIII mutant containing compensatory mutations increased the levels of this GFP reporter ~4-fold (Fig. 7d). As controls, we confirmed that the expression of RNAIII did not affect fluorescence of GFP alone nor GFP fused to the 5′ UTR of a transcript that is not regulated by RNAIII (*gyrB*; Supplementary Fig. 16). We conclude that RNAIII directly binds the *esxA* mRNA to enhance its translation.

To investigate the significance of this interaction at the protein level, we created a Δ*rnaiii* mutant and examined EsxA expression through Western blotting (Fig. 7e, f). EsxA levels were almost undetectable in the RNAIII deletion mutant but were completely restored when RNAIII was expressed from a plasmid. Deletion of RNAIII also reduced the levels of *esxA* mRNA (Fig. 7e, f), suggesting that RNAIII binding to *esxA* stabilises the mRNA. Deletion of RNase III, the bait protein used for the CLASH experiments, did not strongly impact *esxA* mRNA levels, suggesting that the endonuclease does not play a significant role in regulating the stability of the *esxA* mRNA (Fig. 7g, h). We conclude that RNAIII directly controls the levels of the EsxA toxin.

## Discussion

Microorganisms must constantly adapt their transcriptome to respond to changes in their environment. When pathogenic bacteria systematically infect their host, they must derive essential nutrients from the bloodstream; an environment that is usually depleted of crucial cofactors such as iron. To overcome this problem, *S. aureus* has a wide array of virulence factors that enable the bacterium to extract nutrients through host cell

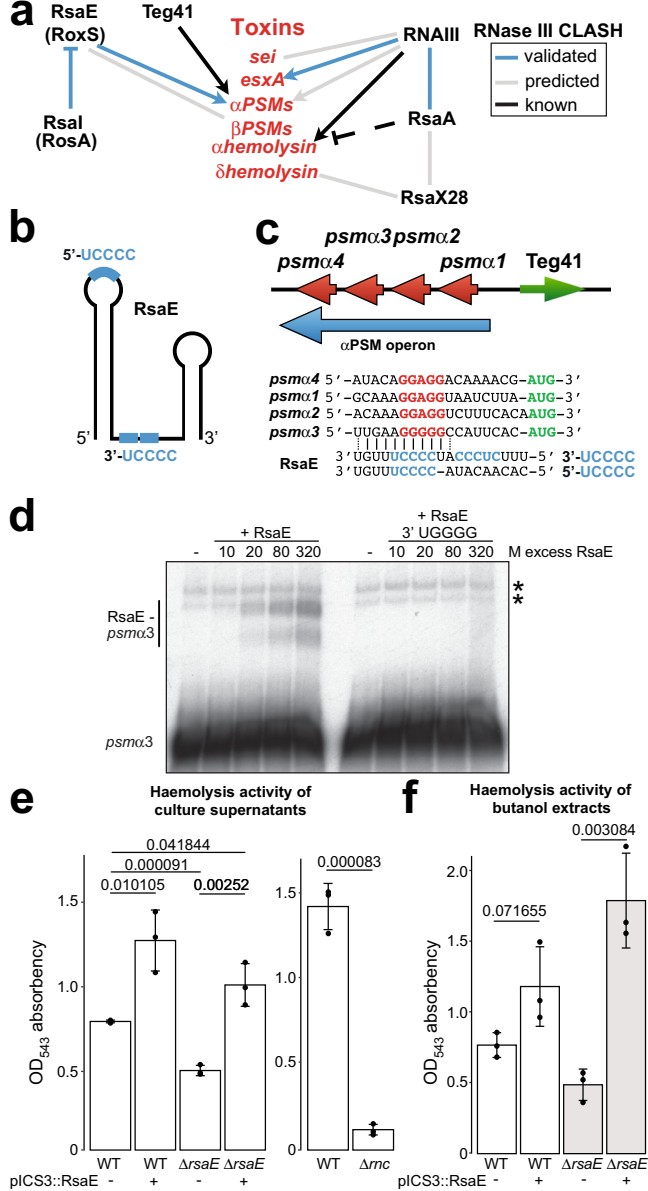

**Fig. 6 CLASH predicts toxin production is extensively regulated by sRNAs. a** Predicted interactions between sRNAs and sRNAs, and sRNAs with mRNAs encoding for toxins (indicated in red). Black line indicates interactions experimentally verified by previous groups. Grey lines indicate interactions predicted by CLASH and blue lines indicate interactions predicted by CLASH that we experimentally verified. Dashed lines indicate indirect regulation. **b** Schematic representation of the structure of the RsaE sRNA. Blue lines and blue text indicate the RsaE UCCCC seed sequences. **c** Schematic representation of the αPSM operon (blue arrow), the location of the four αPSM genes in the operon (red arrow) and the location of the Teg41 sRNA in this genomic location. Below that, a multiple sequence alignment of *psmα1-4* Shine-Dalgarno (red) and translational start codon (green) regions aligned to the RsaE seed sequence motifs. CLASH detected interactions between the 3′ UCCCC motif (light blue) of RsaE with the *psmα2* and *psmα3* transcripts but similar base-pairing interactions can also be drawn with all the PSM transcripts and with the 5′ UCCCC motif (light blue). **d** The 3′ UCCCC motif of RsaE is involved in base-pairing with *psmα3*. EMSA between radiolabelled RsaE and *psmα3* transcript containing the binding sites predicted by CLASH. As a negative control, an RsaE fragment was used in which the 3′ UCCCC motif was mutated to UGGGG. Asterisks (*) indicate self-oligomerisation of the *psmα3* RNA as these are produced in the absence of RsaE and do not increase in abundance when increasing amounts of RsaE is added. Results from a single experiment is shown and the raw data is provided in the Source Data file. **e** Haemolytic activity of supernatants from USA300 LAC WT, RsaE overexpressing strains, *ΔrsaE* and *Δrnc* (RNase III) deletion mutants. Supernatant from overnight cultures was mixed with human blood and the degree of blood cell lysis was measured through optical absorbency. Shown are the averages and standard deviations calculated from three independent experiments. Values above bars display respective *p* value, obtained using Student's two-tailed, unpaired *t* test. **f** As in **e** but after performing butanol extraction on supernatants to enrich for PSMs. Raw images and data used to generate (**d**–**f**) are provided in the Source Data file.

Gram-positive bacteria. Given the roles of RsaE and RsaI in regulating the carbon state of the cell, we speculated that these sRNAs could also contribute to the virulent capacity of *S. aureus*. Indeed, we found that RsaE not only regulates metabolic genes but also regulates expression of cytolytic toxins. To the best of our knowledge, RsaE is the first Gram-positive sRNA that impacts significantly both metabolism and the production of clinically relevant toxins.

Our data suggest that RsaI base-pairing with RsaE induces some cleavage by RNase III in vitro but does not significantly affect RsaE steady state levels in vivo. This was unexpected as if the RsaI-RsaE duplex is indeed a substrate of RNase III, one would expect to see an increase in RsaE levels in *Δrnc*, *ΔrsaI* or the *Δrnc-ΔrsaI* double mutant. This was not the case. However, it is possible that in vivo only a small fraction of the RsaI-RsaE duplex is cleaved by RNase III. It has been proposed that RNase III can function as a non-catalytic RNA-binding protein[87]. Thus, it is possible that the RsaE-RsaI duplex is largely resistant to degradation and that the main function of RNase III here is to stabilise the RsaI-RsaE interaction in vivo or by preventing RsaE from base-pairing with other targets. Thus, RsaI acts as a true sponge for RsaE, where RsaE is inactivated without alteration in its stability. This is mechanistically similar to the regulation of *S. aureus* RNAIII by SprY[58] and *Salmonella* MicF by OppX[88].

One surprising finding was that several different species of RsaE and RsaI accumulate in the tested media, with shorter RsaE species accumulating in rich TSB and shorter RsaI species appearing in human serum, independent of RNase III activity. Processing of RsaI was not previously observed in *S. aureus*,

disruption. Considering the importance of nutrients for survival within the host, the coupling of nutrient sensing to the expression of virulence genes offers an elegant means to respond to such conditions (reviewed in[79]).

Several *S. aureus* sRNAs have already been found to regulate metabolism. One of the better understood sRNAs, RsaE, is known to regulate the TCA cycle, carbon flux, amino acid metabolism and biofilm formation[15–17,19]. In *B. subtilis*, RsaE is involved in regulating the redox state of the cell in response to nitric oxide stress[17]. RsaI, which regulates RsaE activity, is activated when glucose is scarce and subsequently inhibits genes involved in glucose catabolism[19]. Connections between changes in the TCA cycle and virulence have previously been identified in *S. aureus*, which mainly involves controlling the activity of transcription factors[79–83]. For example, the transcriptional regulator CcpE has been shown to drive expression of the TCA cycle whilst also regulating many virulence-associated genes[84]. Additionally, deletion of CcpA, a transcriptional regulator that can repress the TCA cycle[85], lowers RNAIII levels[86]. However, how sRNAs directly connect metabolism to virulence is not well explored in

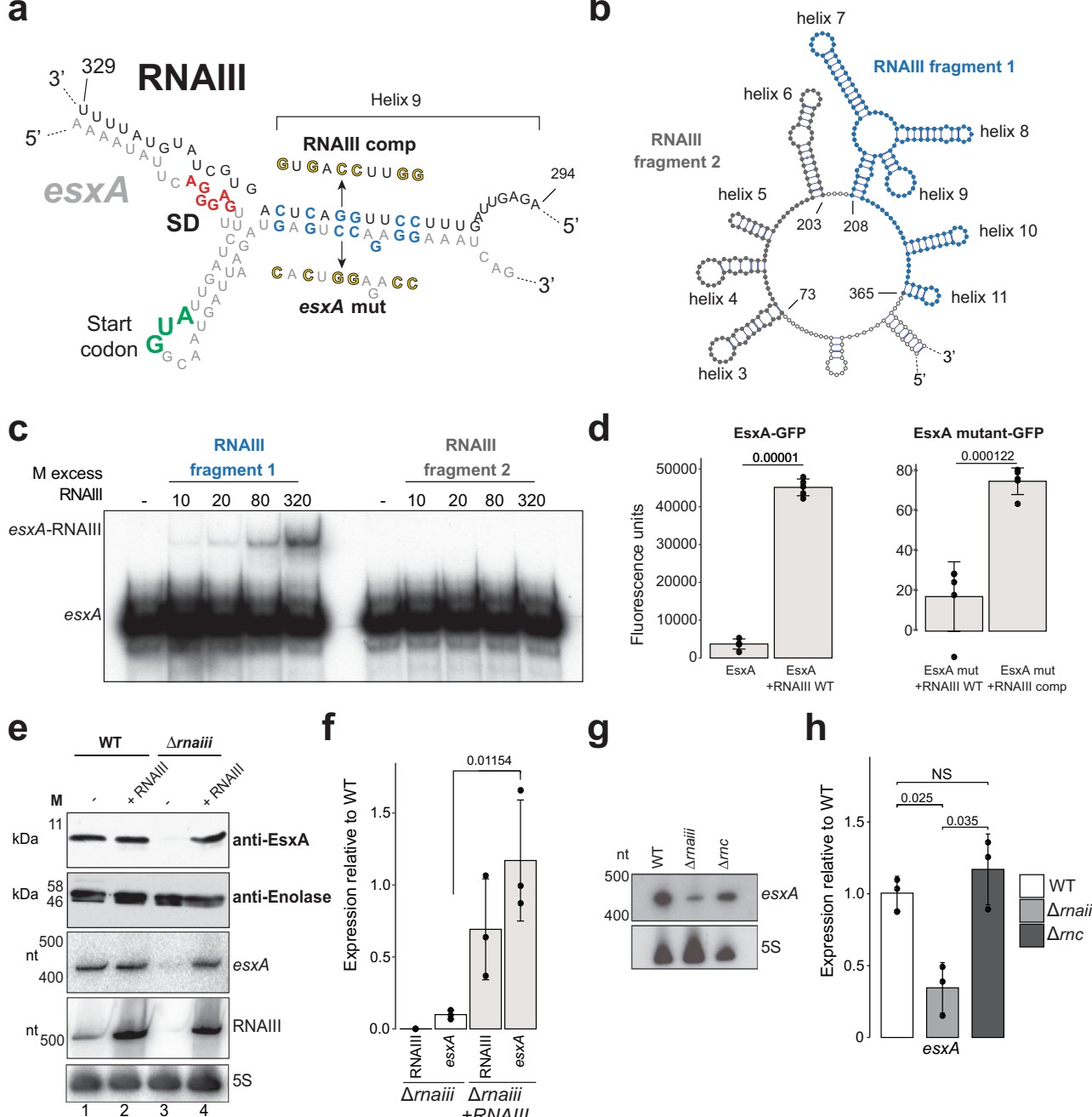

**Fig. 7 RNAIII base-pairing with the *esxA* mRNA is required for EsxA production. a** Predicted base-pairing interactions between RNAIII and *esxA* as found through CLASH. Green text: translational start codon; red text: Shine-Dalgarno (SD) sequence, blue text: nucleotides that were mutated (yellow) to generate the seed and the compensatory mutants. **b** Secondary structure of the region containing RNAIII fragments 1 (blue) and 2 (dark grey) that were used for in vitro binding assays. **c** EMSA between radiolabelled *esxA* and RNAIII fragment 1 (blue text; containing helix 9) and RNAIII fragment 2 (grey text). Results from a single experiment is shown. **d** In vivo GFP-fusion reporter assay[76]. A portion of the 5′ UTR and CDS of *esxA* was fused to sfGFP and expressed in RN4220. FACS was used to measure GFP fluorescence of WT and *esxA* seed mutant upon RNAIII overexpression. "RNAIII comp": mutant predicted to base-pair with the *esxA* seed mutant. Shown are the averages and standard deviations for five independent biological replicate experiments. Values above bars display *p*-value (mutant vs WT; student's two-tailed, unpaired *t* test). **e** Stability of EsxA protein and mRNA upon RNAIII deletion and restoration. The Eno1 protein signal was used as a loading control. Northern blotting was performed to detect *esxA*, RNAIII and 5 S rRNA levels. Shown are representative results from three independent experiments. Statistical significance was measured through Student's unpaired, two-tailed *t* test. **f** Quantification of the results in **e**. Shown are averages and standard deviations. Values above bars display *p* value (mutant vs WT; student's two-tailed, unpaired *t* test). **g** Stability of *esxA* mRNA upon deletion of RNase III. Shown are northern blot results from one of three independent biological replicate RNA samples that were analysed by RT-qPCR (**h**). **h** Expression of esxA in WT, RNAIII and RNaseIII mutants. Shown are the mean and standard deviation of three independent biological replicates. Values above bars display *p* values (mutant vs WT) obtained using Student's two-tailed, unpaired *t* test. Strains tested: WT (white), *Δrnaiii* (light grey) and *Δrnc* (dark grey). Raw images and data used to generate (**c**–**h**) are provided in the Source Data file.

presumably because different growth conditions were used. Additionally, we also used high resolution acrylamide gels. In *B. subtilis*, RNase Y is known process RoxS by cleaving around 20 nucleotides from the 5′ end and processing of RsaE has also been previously observed in *Staphylococcus epidermidis*. These cleavage events expand RsaE base-pairing potential with other mRNA targets, presumably by enhancing base-pairing potential with the UCCCC motif in the 5′ end of RsaE[18,89] that is normally sequestered in a stem-loop structure. However, our Nanopore sequencing revealed that in the conditions tested here, RsaE and RsaI undergo differential trimming of U-rich terminator sequence at the 3′-end. This was particularly prominent for RsaE in TSB versus human serum. Investigation into the regulatory outcome of this is beyond of scope of this manuscript, but previous studies show that the U-tail of sRNAs is important in dictating their regulatory potential. In *E. coli*, shortening the U-tails of SgrS, RhyB, MicA and MicF inhibits their ability to silence their mRNA targets as they are no longer bound by Hfq[90,91]. As such, we hypothesise that 3′-end trimming of RsaE and RsaI observed here may impact their regulation of target mRNAs in response to specific environment conditions, perhaps by altering the stability of the sRNA. Our EMSAs were performed with RsaE transcripts that lacked the terminator sequences. Therefore, this element is not required in vitro for forming stable base-pairing interactions. The shorter form of RsaE appears to be an active form as overexpression of RsaI in TSB resulted in increased levels of RsaE mRNA targets, likely because RsaE was sponged by RsaI (Fig. 5b–d). However, in vivo it may be the case that trimming of the RsaE U-tract in TSB prevents binding of specific proteins that contribute to the formation or stabilisation of RsaE base-pairing interactions. *S. aureus* PNPase, RNase R and YhaM have been shown to exhibit 3′–5′ exonuclease activity, and future studies will aim to identify the RNase responsible for RsaI and RsaE 3′ trimming. We hypothesise that the impact of an sRNA on gene expression is not only dictated by its expression levels but also by its maturation and processing.

Several sRNAs were shown to have a role in regulating post-transcriptionally the production of toxins in *S. aureus*. Currently, RNAIII[73] and SSR42[92] are known to regulate the translation of α-toxin and Teg41 is known to stimulate production of αPSMs[72]. The work presented here adds several additional sRNA players to this growing list. We identified RsaE interacting with both α- and β−PSMs; RNAIII interacting with *esxA, sei* and *psmα4*; and RsaX28 interacting with *hld*. We further validated several of these interactions. Of particular interest was the observed interactions identified between RsaE and αPSM transcripts. Our CLASH analyses identified base-pairing interactions between RsaE and the Shine-Dalgarno sequences (SD) of *psmα2* and *psmα3*. Using EMSAs, we were able to verify that RsaE can base-pair with the *psmα3* toxin mRNA in vitro, albeit inefficiently. Consistent with our CLASH data, this interaction required the 3′ UCCCC motif of RsaE. The base-pairing of RsaE with *psmα* SD sequences suggests a canonical mode of sRNA-mediated regulation where the sRNA prevents translation of the toxins by blocking the 30 S subunit access to the ribosome binding site. We therefore predicted that deleting RsaE would result in increased expression of these cytolytic toxins. Instead, culture supernatants from cells lacking RsaE showed significantly reduced cytolytic activity in our hae-molysis assay. Additionally, our mass-spectrometry data showed that RsaE deletion reduced substantially the accumulation of PSMα1 and PSMα4 in culture supernatants, providing an explanation for the observed decrease in haemolytic activity in this strain. However, since the levels of the δ-toxin were also significantly reduced in this mutant, we cannot exclude the possibility that at least some of the changes in haemolysis activity was the result of changes in the level of this toxin. Regardless,

these data suggest that RsaE has a positive influence on PSM expression.

The observation that there were no differences in the levels of PSMα3 in culture supernatants of the WT and Δ*rsaE* strain implies that RsaE does not impact the expression of this specific toxin. It is possible that base-pairing of RsaE with the αPSM operon only directly impacts a subset of the PSMα toxins, such as PSMα1 and PSMα4 that were significantly reduced in the RsaE deletion mutant. The fact that we identified the interaction between RsaE and *psmα2* and *psmα3* when using RNase III as a bait protein indicates that RsaE base-pairing could trigger RNase III-dependent cleavage of the mRNAs. The four αPSMs are produced as a single polycistronic mRNA that is highly structured and the SD and/or translational start codon are predicted to be sequestered within stem or stem-loop structures for all the *psmα* transcripts (Supplementary Fig. 17). Interestingly, even though these toxins are encoded on a single operon, they are differentially expressed at the protein level; in USA300, PSMα1 and PSMα4 are expressed the highest. As such, RNase III may be guided to particular *psmα* transcripts to cleave near the SD sequence, which would liberate the individual toxin mRNAs from the operon and offer a way of differentially regulating the translation of the individual toxins. We are currently testing this hypothesis.

Previous RNA capture and sRNA overexpression experiments on RsaE did not identify any toxin mRNAs as putative binding partners, and instead identified targets mostly involved in the TCA cycle and amino acid metabolism[17]. Such a discrepancy could be explained by the different strains and growth conditions used. We performed our analyses on cells grown in late exponential phase ($OD_{600}$ 3.0) when several virulence factors in *S. aureus* (such as RNAIII and α-toxins) are highly expressed, increasing the likelihood of detecting such interactions. Additionally, αPSMs tend to be expressed at elevated levels in clinical strains, especially in the USA300 isolate that we used[77]. Alternatively, the number of interactions between RsaE and *psmα* transcripts may be low relative to RsaE and its metabolic targets, thus requiring the capture of a bait protein (e.g., RNase III) for enrichment.

Although RsaE was previously hypothesised to play an indirect role in regulating virulence[22], our findings suggest a direct and central role. We hypothesise that RsaE acts as a molecular switch, balancing TCA cycle activity with virulence. Collectively, our data reinforce that toxin production, virulence and metabolism are interconnected tightly.

RNAIII is known to regulate a wide variety of targets, including immune evasion proteins, pro-virulence transcription factors, toxins and a regulator of cell wall integrity[11]. Through RNase III CLASH, we identified novel targets pertaining to several of these classes. In particular, we showed that RNAIII stimulates the production of EsxA, a toxin involved in bacterial persistence and spread during infection[74,93,94]. Interestingly, the *agr* locus is known to contribute to *esxA* transcription[95], and therefore *agr* and RNAIII are likely to act in tandem to regulate EsxA production at both the mRNA and protein level as part of a coherent feed forward loop. This type of regulatory network ensures that target genes (*esxA* in this case) are rapidly induced. Overall, this interaction further expands RNAIII's involvement in *S. aureus* invasion and expansion. Future mechanistic studies will be required to investigate exactly how RNAIII stimulates *esxA* translation and how RNase III contributes towards this.

We also identified RNAIII as interacting with *psmα4* and sta-phylococcal enterotoxin I (*sei*), classical toxins that mediate cytolysis and T-cell activation respectively, and *saeR*, which is part of a 2-component system involved in up regulation of virulence factors in response to phagocyte-derived stimuli[96] (Supplementary Figs. 6, 12). Future experiments will be necessary

to further interrogate the functional outcomes of these, but it is likely that RNAIII is even more deeply integrated into the virulence networks of *S. aureus* than previously thought.

Although RNase III in *S. aureus* has been found to bind several sRNAs, including RsaA, RsaE, RsaI and RNAIII[27], we were surprised by the plethora of sRNA-target interactions identified through RNase III CLASH. This is because deletion of RNase III does not affect the growth of *S. aureus* in culture[27,97], in contrast to its essentiality in *B. subtilis*[98]. However, the diverse set of interactions we have identified as being targeted by RNase III imply that this enzyme may play important roles in non-standard laboratory conditions, such as during infection. Indeed, our observation that removal of RNase III activity led to an almost complete loss of the haemolytic capacity of culture supernatant strongly supports this idea. Further exploration into this idea may place RNase III as a potential drug target.

Remarkably, we observed little overlap between the captured RNase III interactomes of USA300 and JDK6009. This is best exemplified by our identified interaction between RsaA and RNAIII, which was by far the most abundant sRNA-sRNA interaction in JKD6009. Yet, we did not detect this interaction in the USA300 CLASH data. Although CLASH is a stochastic method, we were able to identify interactions common to both strains, such as that between RsaA and *mgrA*. One potential explanation for the difference in RNase III interactomes is that these strains exhibit different growth dynamics; we observed that USA300 grows faster and to higher optical densities than JKD6009 (Supplementary Data 5). Although CLASH was performed on these strains at the same optical density (OD$_{600}$ ~3.0), the difference in growth rates, and therefore the time required to reach this density, may have a strong effect on sRNA expression or activity. This may, at least in part, explain the differences between the JKD6009 and USA300 RNA-RNA interactomes. As such, the growth rate of different *S. aureus* isolates may be an underappreciated contributing factor when examining sRNA-target interactions. Future studies could seek to perform RNase III capture across different growth stages and compare these between strains.

In this manuscript we have mostly focused on interactions involving verified, trans-acting sRNAs as well as interactions that involve base-pairing with ribosome binding sites. However, it is important to note that our CLASH data contains many other classes of RNA-RNA interactions. For example, our focus on trans-acting sRNAs excludes several experimentally verified interactions, such as toxin-antitoxin systems involving anti-sense sRNAs[8,99]. Interestingly, interactions between two different mRNAs were the most abundant class within our CLASH data, and we identified an increased number of mRNA-mRNA interactions after exposure to RPMI and LPM versus TSB, suggesting that these contribute to stress adaptation. It is unclear why mRNA-mRNA interactions were predominant, however, there is evidence suggesting that these could be biologically meaningful. Mediati et al.[61] demonstrated that an mRNA-mRNA interaction in *S. aureus* regulates resistance to vancomycin, one of the last drugs available to battle multidrug-resistant *S. aureus* infections. Similarly, in *Listeria monocytogenes*, it has been found that the 3′ UTR of an mRNA encoding a haemolysin binds the 5′ UTR of an mRNA encoding a protein chaperone. This interaction prevents nuclease-mediated decay of the chaperone, promoting its translation, and this subsequently contributes to virulence[100]. Our data may contain other such examples of regulatory mRNA-mRNA interactions, which may form a greater component of the stress adaptation apparatus than previously thought.

It is also possible that many of the interactions that were labelled as mRNA-mRNA intermolecular interactions in our data may in fact represent interactions between processed sRNAs

and their mRNA substrates. In *E. coli* and *Salmonella*, many mRNA transcripts undergo cleavage to generate functional sRNAs[34,59,101–103]. With regards to *S. aureus*, the sRNA Teg49 was originally thought to be an independently expressed sRNA that has a role in regulating virulence, but further study revealed Teg49 to be a cleavage product of the *sarA* 5′ UTR[104,105]. Such processing events may be a widespread phenomenon in *S. aureus*, and a portion of our mRNA-mRNA interactions may represent interactions between mRNAs and processed sRNAs that have been derived from mRNAs.

We also identified many mRNA transcripts bound to their cognate anti-sense RNA. This confirms previous observations of RNase III being a major player in controlling sense-anti-sense mRNA duplexes[27] and expands the list of genes regulated in such fashion in *S. aureus*. Finally, we have identified intergenic regions interacting with both mRNAs and sRNAs. Intergenic regions have long been used as a resource for discovering novel sRNAs, and these interactions may yield similar results. We therefore believe that our CLASH data will be a rich resource for the identification of novel sRNAs, potentially involving unique regulatory methods.

## Methods

**Bacterial strains and culture conditions.** An overview of all *E. coli* and *S. aureus* strains used in this study is provided in Supplementary Data 1. *S. aureus* USA300 and JKD6009 strains served as parental strains. *S. aureus* RN4220 served as an intermediate for transducing plasmids into USA300 and JKD6009. The RN450 strain was used to produce and harvest 80α phage for transduction of USA300 and JKD6009. The *E. coli* DH5α strain was used for general plasmid propagation. *S. aureus* strains were grown in tryptic soy broth (TSB; Oxoid) under aerobic conditions at 37 °C with shaking at 180 rpm, while *E. coli* was grown in lysogeny broth under the same conditions. The media was supplemented with antibiotics when appropriate at the following concentrations: ampicillin, 100 μg/mL; chloramphenicol, 15 μg/mL; erythromycin, 10 μg/mL. For inducing sRNA expression from the pRMC2 vector, anhydrotetracycline was used at 1 μg/mL. CLASH was performed in *S. aureus* JKD6009 *rnc*::HTF and USA300 *rnc*::HTF alongside the untagged parental strains. Tagged and sRNA deletion strains were generated through allelic exchange using the pIMAY plasmid[106]. RsaE inducible pRMC2 with P$_{txyl/tetO}$ promoter was kindly provided by Philippe Bouloc.

**Construction of sRNA and mRNA-GFP expression vectors.** All oligonucleotides and DNA fragments used for cloning are listed in Supplementary Data 2 and were purchased from Integrated DNA Technologies (IDT). For inducible sRNA expression, the pRMC2 expression vector was used[107]. The sRNAs were synthesised as gBlocks with flanking 5′ KpnI and 3′ EcoRI sites. These gBlocks were cloned into the pJET 1.2 cloning vector (Thermo Fisher) through blunt-end ligation and the insert confirmed through Sanger sequencing (Edinburgh Genomics, Edinburgh, UK). Positive inserts were then excised through KpnI and EcoRI digestion and ligated into digested pRMC2 overnight at 16 °C using T4 DNA ligase (NEB). For constitutive sRNA expression, the pICS3 vector was used[76]. Here, sRNAs were cloned under the control of the *amiA* promoter from *Streptococcus*. sRNAs were synthesised as gBlocks (IDT) in the form of 5′ KpnI—*PamiA*—sRNA—EcoRI 3′. These sRNAs were inserted into pICS3 in the manner described above. For the FACS analyses, the 5′ UTR and a small portion of the coding sequence of *esxA* were synthesised as gBlocks (IDT), flanked by 5′ PstI and 3′ EcoRV restriction sites. These were verified using pJET 1.2 as above and then cloned into the pCN33 shuttle vector containing sfGFP[76,108].

**Western blotting.** Strains for western blotting were lysed as described in the CLASH protocol. Forty mg of protein was resolved on 8% or 15% polyacrylamide gels and transferred to a nitrocellulose membrane. Membranes were blocked for 1 h in blocking solution (5% non-fat milk, 0.1% Tween-20 in PBS). Primary antibody probing was performed overnight at 4 °C using anti-EsxA[109] (1/500) or the anti-TAP antibody (1/5000, ThermoFisher) to detect tagged RNases. The membrane was then washed three times in PBS (PBS with 0.1% Tween-20) and visualised using an HRP-linked goat anti-rabbit antibody (1/500, Abcam) and Pearce enhanced chemiluminescence solutions (ThermoFisher).

**Reverse-transcription quantitative PCR.** The qRT-PCR analyses were performed on RNA samples extracted from cells that had underwent nutrient shift and on various strains grown to OD$_{600}$ 3. Total RNA was extracted using a guanidium thiocyanate, acidic phenol:chloroform-based extraction[34,110]. Briefly, cells were resuspended in 550 μL of GTC phenol buffer (4 M guanidium thiocyanate, 50 mM Tris pH 8, 10 mM EDTA, 100 mM beta mercaptoethanol, 2% sarcosyl, 100 mM

sodium acetate pH 5.2, 50% acidic phenol pH 4.3). Zirconia beads were added, and the cells lysed by vortexing. Afterwards, 300 µL of chloroform was added and the mixture centrifuged. A second phenol:chloroform extraction was then performed on the aqueous layer for further clean-up and the RNA precipitated using ethanol and glycogen.

Isolated RNA was ttreated with DNase I (TURBO DNase; Thermo Fisher) for 1 h at 37 °C in the presence of 2 U of SUPERasin. RNA was subsequently purified using RNAClean XP beads (Beckmann Coulter) diluted to a concentration of 5 µg/µL. The qPCRs were then performed using the Luna Universal One-Step RT-qPCR kit (NEB) according to the manufacturer's instructions using 5 ng of RNA. The PCR was run on a LightCycler 480 (Roche). Analysis of the qPCR data was performed as previously described[34]. Briefly, the IDEAS2.0 software was used to calculate Ct values using the absolute quantification/fit points method with default parameters, and the fidelity of the PCR was examined through melt curve genotyping analyses. To calculate the relative fold-change of genes, the $2^{\wedge}(\Delta\Delta Ct)$ method was employed using 5 S rRNA as a control. Each qPCR experiment was performed in technical triplicate. For final data analyses, the mean and standard error of the mean of three biological triplicates was calculated and plotted. All oligonucleotides used for qPCR analyses are listed in Supplementary Data 2.

**RNase III activity assay**. Over-expression and purification of RNase III from *E. coli* were performed as described with the following modifications[54]. After clearing of bacterial lysate, nucleic acids were removed by digestion with the addition of 500 U of Micrococcal Nuclease S7 (Sigma) for 1 h at 25 °C. The Ni-NTA beads were washed three times with buffer A (25 mM Tris HCl pH 8, 8% ammonium sulfate, 0.1 mM EDTA) containing 50 mM imidazole before elution with buffer B (25 mM Tris HCl pH 8, 1 M NH₄Cl, 1 mM DTT) in the presence of 100 mM imidazole or 200 mM imidazole. After dialysis and concentration, the RNase III was stored in 30 mM Tris HCl, 500 mM KCl, 0.1 mM EDTA, 0.1 mM DTT and 50% glycerol, and was used directly for activity assay with 5′-end radiolabelled RNAs[54]. In these assays, 10 mM MgCl₂ or 10 mM CaCl₂ was used to modulate RNase III activity.

**Media shifts and UV cross-linking**. *S. aureus* was grown overnight, diluted into fresh TSB the day after and grown to an OD₆₀₀ value of 3. For CLASH, 65 mL of OD₆₀₀ 3 cells were then cross-linked with 250 mJ of 254 nm UV using a Vari-X-Linker and harvested using vacuum filtration[43]. The cells were then flash-frozen in liquid nitrogen. The remaining cells in TSB were harvested through vacuum filtration and resuspended in an identical volume of either preheated RPMI 1640 (Gibco), LPM pH 7.6 or LPM pH 5.4. The cells were then incubated at 37 °C, shaking at 180 rpm. After 15 min, 65 mL of cells were cross-linked and harvested as before. For RNAtag-seq, 65 mL of OD₆₀₀ 3 cells were harvested and flash-frozen. The remaining cell suspension was harvested through vacuum filtration and resuspended in an identical volume of either RPMI 1640, LPM pH 7.6 or LPM pH 5.4 (or back into the original TSB medium as a control). Samples were taken and vacuum harvested after 5, 10, 15 and 30 min before being flash-frozen.

**CLASH**. Cells were removed from the filters through two washes with 10 mL of phosphate buffered saline (PBS) and pelleted through centrifugation at 4600 g for 10 min, 4 °C. The supernatant was discarded, and the cell pellets weighed. The pellets were then resuspended in 2 volumes of TN150-lysostaphin (50 mM Tris pH 7.8, 150 mM NaCl, 100 mg/mL lysostaphin, 0.1% NP-40, 0.5% Triton X-100), and 60 U of DNase RQ1 (Promega) and 200 U of SUPERasin (Invitrogen) were added. The cells were incubated for 1 h at 20 °C for the lysostaphin to degrade the outer cell wall. The cells were then transferred to 15 mL conical tubes and lysed through bead beading with 0.1 mm zirconia beads (Biospek Products) for 5 min. Afterwards, 2 volumes of TN150-antipeptidase (50 mM) Tris pH 7.8, 150 mM NaCl, 0.1% NP-40, 0.5% Triton X-100, 10 mM EDTA, 1 mini cOmplete protease inhibitor per 10 mL (Roche) was added. The beads were then separated from the lysate by centrifugation at 4600 × g for 20 min at 4 °C and then the lysate transferred to 1.5 mL tubes. The insoluble and soluble fractions of the lysate were then separated through centrifugation at 20,000 × g for 20 min at 4 °C. Magnetic anti-FLAG M2 beads (Sigma Aldrich) were washed three times in TN150 (50 mM Tris pH 7.8, 150 mM NaCl, 0.1% NP-40, 0.5% Triton X-100), with 75 µL taken for each sample. The washed beads were then distributed equally between the cleared lysates and incubated for 2 h at 4 °C with rotation. Following capture, the beads were washed three times in TN1000 (50 mM Tris pH 7.8, 1 M NaCl, 0.1% NP-40, 0.5% Triton X-100) for 10 min at 4 °C with rotation. The beads were then rinsed three times in TN150 and then resuspended in a final volume of 250 µl TN150. To cleave the RNases from the FLAG beads, homemade TEV protease was added, and the samples incubated for 2 h at room temperature with rotation. Following cleavage, an extra 350 µL of TN150 was added to the samples and the eluate collected following separation from the beads using a magnetic rack. The eluates were then RNase digested with 1 µL of a 1:100 dilution of RNace-It (Agilent) for 7 min at 20 °C. The RNase digestion was stopped with the addition of 0.4 g of guanidium hydrochloride (GuHCl; Sigma Aldrich). Following digestion, 100 µL of nickel-NTA agarose beads (Qiagen) was added, prewashed in wash buffer 1 (50 mM Tris pH 7.8, 0.1% NP-40, 5 mM β-mercaptoethanol, 0.5% Triton X-100, 300 mM NaCl, 10 mM imidazole, 6 M GuHCl) and the proteins were captured overnight. The capture solutions were then transferred to Pierce SnapCap spin columns (Thermo

Fisher, 69725). The harvested beads were washed three times with wash buffer 1 and three times with NP-PNK (50 mM Tris pH 7.8, 0.1% NP-40, 5 mM β-mercaptoethanol, 0.5% Triton X-100, 10 mM MgCl₂). Afterwards, the RNAs were dephosphorylated on-column using 4 U of TSAP (Promega) in the presence of 80 U of rRNasin (Promega) in 1X PNK buffer (50 mM Tris pH 7.8, 10 mM MgCl₂, 10 mM β-mercaptoethanol, 0.1% Triton X-100) for 1 h at 20 °C. The beads were then washed once with wash buffer 1 to inactivate the enzyme and then three times with NP-PNK to remove residual guanidium. The RNAs were then radiolabelled at the 5′ end using 30 U of T4 PNK and 3 µL of ³²P-ATP in 1X PNK buffer for 100 min at 20 °C. 1 mM of cold ATP was then added, and the reaction left to proceed for another 40 min to ensure complete 5′ phosphorylation of the RNAs. The beads were then washed three times with wash buffer 1 and three times with NP-PNK. Sequencing adaptors were then ligated to the ends of the RNAs. First, an L5 adaptor (Supplementary Data 2) was ligated to the 5′ end using 200 mmoles of adaptor and 40 U of T4 RNA ligase in the presence of 80 U of rRNasin and 1 mM ATP in 1X PNK buffer, for 16 h at 16 °C. The beads were subsequently washed once with wash buffer I and three times with NP-PNK. Afterwards, 60 mmoles of App_PE adaptor (Supplementary Data 2) was ligated onto the 3′ end using 600 U of T4 RNA Ligase 2 truncated K227Q (NEB). This reaction was carried out in 1X PNK buffer with 10% PEG-8000 and 30 U of rRNasin for 6 h at 25 °C. Afterwards, the beads were washed once in wash buffer I and three times in wash buffer two (50 mM Tris pH 7.8, 10 mM β-mercaptoethanol, 0.1% NP-40, 0.5% Triton X-100, 50 mM NaCl, 10 mM imidazole). The protein-RNA complexes were then eluted from the beads through addition of 200 µL of elution buffer (wash buffer two with 250 mM imidazole), repeated for a total of two times. The proteins were then pooled and precipitated through addition of trichloroacetic acid (Sigma Aldrich) to a final concentration of 20% and left to precipitate on ice for 20 min. The samples were then centrifuged at 20000 x g for 20 min at 4 °C. The pellets were then washed with 800 µL of acetone, dissolved in 20 µL of loading buffer (Novex) and resolved on a 4-12% Bis-Tris gel. The protein-RNA complexes were visualised through autoradiography and the gel piece containing these was excised. The RNAs were then extracted through incubation in 4 mL of extraction buffer (50 mM Tris pH 7.8, 0.1% NP-40, 5 mM β-mercaptoethanol, 1% SDS, 5 mM EDTA, 50 mM NaCl, 60 mg/mL proteinase K) at 55 °C for 2 h. Following this, the RNAs were purified through phenol:chloroform extraction and then resuspended in 20 µL of DEPC-treated water. The RNAs were reverse transcribed using the PE_reverse primer and SuperScript IV according to the manufacturer's instructions. Sample RNA was subsequently degraded through addition of 10 U of RNase H (NEB). Afterwards, the cDNA was purified through RNAClean XP beads (Beckmann Coulter) and resuspended in a final volume of 11 µL. Half of this cDNA was then used as a template for PCR with Pfu polymerase, using BC reverse and P5 forward primers. The cycling conditions were as follows: 95 °C for 2 min; 24 cycles of 95 °C for 20 s, 52 °C for 30 s and 72 °C for 1 min; final extension of 72 °C for 5 min. The PCR product was treated with 40 U of Exonuclease I (NEB) to degrade free primer and the DNA library purified using RNAClean XP beads. The library was then resolved on a 2% MetaPhor (Lonza) gel and 175-300 bp fragments were excised and gel extracted through a MinElute column. The library was quantified using a 2100 Bioanalyzer and a DNA HS assay (Agilent). Individual libraries were then pooled together to produce an equimolar solution and sequenced through 75 bp paired-end sequencing on an Illumina HiSeq 4000 or NovaSeq 6000 platform (Edinburgh Genomics).

**Flow cytometry**. Overnight cultures of strains expressing pCN33-*target-gfp* and pICS3-*sRNA* were diluted 1:40 into 2 mL of PBS. Translation of GFP fusions was monitored on the LSRFortessa Special Order Research Product (BD) from a 500 µL aliquot of PBS-diluted samples on a 530/30 nm bandpass filter. Sample acquisition of 100,000 events was performed on the built-in Diva (LSRFortessa SORP) software. Median fluorescence intensities (MFI) were determined from the entire population (no gating) using the flowJo software where FSC and SSC were used to gate any fluorescence attributed to cellular background. The average MFI and standard deviations were calculated and plotted. To determine significant differences, a two-sample student's *t*-test (assuming unequal variance) was used.

**Northern blotting**. Total RNA was then extracted through acid guanidinium thiocyanate-phenol-chloroform extraction as described above. Total RNA was resolved on an 8% polyacrylamide TBE-urea gel, transferred to a nitrocellulose membrane via electroblotting and then crosslinked to the membrane through exposure to 1200 mJ of 254 nm radiation.

For hybridisation, membranes were firstly prehybridised in 10 mL of UltraHyb (Am-bion). Membranes were then probed with a ³²P-labelled DNA oligonucleotide (Supplementary Data 2) at 42 °C for 20 h. Membranes were washed twice in 2×SSC with 0.5% SDS for 10 min. Membranes were then imaged using a phosphorimager screen and FujiFilm FLA-5100 scanner using the IP-S filter. For 5 S rRNA, imaging was also performed through autoradiography.

**Electrophoretic mobility shift assays**. In vitro transcription and radiolabelling of RNA was carried out using a MEGAscript T7 transcription kit (ThermoFisher) as per the manufacturer's instructions. The RNAs were refolded in structure buffer (25 mM Tris, 150 mM KCl, 1 mM MgCl₂) using a thermal cycler by heating to

95 °C for 1 min, slowly cooled to 25 °C for 5 min and finally incubated at 25 °C for 20 min. Binding reactions between the radiolabelled RNA and cold RNA were then set up in 1:0, 1:10, 1:20, 1:80 and 1:320 molar ratios and incubated at 25 °C for 20 min. Native loading buffer was added to a 1X concentration (10% sucrose, 0.1X TBE, 0.04% bromophenol blue) and then RNA complexes were resolved on an 6% acrylamide TBE gel. The gel was dried under vacuum for 1 h at 80 °C and imaged as described above.

**RNAtag-Seq**. Total RNA was extracted from cells shifted to RPMI or TSB as described above. The cDNA libraries were generated utilising the RNAtag-Seq protocol[45]. Briefly, 500 ng of total RNA from each tested condition was fragmented by incubation at 92 °C for 6 min in 2X FastAP buffer and then snap chilled on ice. Eight units of TURBO DNase (Thermo Fisher), 10 units of FastAP (Thermo Fisher) and 40 units of rRNasin (Promega) were then added and the mixture and incubated at 37 °C for 30 min. The RNA was purified using RNAClean XP beads (Beckman Coulter) and eluted in 10 μL of water. To this, 2 μL of a unique barcoded RNAtag-Seq RNA primer (100 μM) was added and the sample was heat denatured at 70 °C for 2 min. The rRNA adaptor was then ligated onto the sample by mixing with ligation buffer (final concentration in reaction: 1X T4 RNA ligase buffer, 9% DMSO, 1 mM ATP, 20% PEG 8000, 36 units T4 RNA ligase). Reactions were incubated for 90 mi at 22 °C and the reaction inactivated by the addition of 80 μL RLT buffer. RNA was subsequently purified using phenol:chloroform extraction. rRNA was then depleted from the sample using a MICROBexpress kit (Invitrogen). Following elution, RNA was again purified using RNAClean XP beads. cDNA was produced using the AM2 primer and SuperScript IV (Invitrogen) as per manufacturer's instructions. The remaining RNA was degraded by addition of 100 mM sodium hydroxide and incubated at 70 °C for 12 min. The solution was then neutralised with addition of 100 mM acetic acid. The cDNA was purified using RNAClean XP beads and eluted in 11 μL. For addition of the 3′ linker, 1 μL of 80 μM 3Tr3 primer was added and ligated onto the cDNA using CircLigase (Lucigen) as per manufacturer's instructions for 5 h at 60 °C. cDNA was purified again using RNAClean XP beads and eluted in 15 μL of water. The cDNA libraries were then amplified using AccuPrime polymerase (ThermoFisher) as per manufacturer's instructions using the 2 P primers. Afterwards, 40 units of exonuclease I (NEB) were added, and the reaction incubated at 37 °C for 1 h to degrade the PCR primers. Amplified libraries were then separated on a 6% TBE polyacrylamide gel and fragments above the primer-dimer band purified.

Libraries were sequenced on an Illumina HiSeq 4000. The adaptor and oligonucleotide sequences are provided in Supplementary Data 2.

**Haemolytic activity assays**. For comparative analysis of supernatants, culture growth was synchronized. Overnight cultures of each strain were diluted 1:100 in 10 mL of fresh TSB and grown for 3 h. These 3-h cultures were subsequently diluted into 25 mL of fresh TSB to a starting OD$_{600}$ of 0.05. The cultures were then grown for 15 h. Culture supernatants were extracted by centrifugation of the cell suspension at 20,0000 × g. Supernatants were then diluted 1:2 in haemolysis assay buffer (40 mM CaCl$_2$, 1.7% NaCl). In total, 200 μL of diluted supernatant was incubated at 37 °C in a tube revolver with 25 μL of whole human blood for 10 min. The samples were centrifuged at 5500 × g and 100 μL of the supernatant was transferred to a 96-well plate. The degree of erythrocyte lysis was determined by reading the absorbance of the samples at OD$_{543}$. Butanol extractions of PSMs from *S. aureus* supernatants were performed[111]. Overnight cultures were centrifuged, and the supernatant collected. Afterwards, 1-butanol was added to a final concentration of 25% and the mixture centrifuged. The upper phase was finally collected. Extractions were then incubated with human blood and the degree of haemolysis measured as described above.

**Proteomics**

*Sample preparation*. Cultures of parental USA300 pICS3, ΔrsaE pICS3 and ΔrsaE pICS3-RsaE were grown overnight in TSB at 37 °C with 180 rpm shaking. The following day, each culture was diluted 1:100 into 25 mL of fresh TSB and grown for 3 h Cultures were then re-inoculated into another 25 mL of fresh TSB for a starting OD$_{600}$ of 0.05 and grown for 15 h at 37 °C with shaking. Samples were grown in biological triplicates.

The following day, 1 mL of culture was centrifuged at 10,000 × g for 1 min and 500 μL of culture supernatant was moved to a new tube. 2 mL of freezing-cold acetone was added, and solutions were incubated at −20 °C for 1 h to precipitate the proteins. Precipitated proteins were then pelleted through centrifugation at 13,000 × g for 30 min at 4 °C. Pellets were washed with 70% ethanol and then resuspended in resuspension buffer (50 mM Tris pH 7.8, 150 mM NaCl, 0.5% SDS, 5 mM MgCl$_2$, 5 mM CaCl$_2$). Protein concentrations were measured using a Qubit system and then 50 μg of protein placed in a new tube and samples made up to identical volumes through addition of resuspension buffer.

Protein extracts were treated with 10 mM DTT at 56 °C for 30 min and then diluted 1:8 with UA buffer (8 M urea, 100 mM Tris pH 8). Extracts were passed through a FASP column (Expedeon) through centrifugation at 20,000 × g and washed with 200 μL of UA buffer. In total, 100 μL of IAA buffer (50 mM iodoacetamide in UA buffer) was then added, and the samples stored in darkness for 10 min at room temperature. Afterwards, the IAA buffer was passed through

the column by centrifugation and the column was washed twice with 100 μL of UA buffer. The column was then washed twice with 100 μL of ABC buffer (50 mM ammonium bicarbonate in water). The column was transferred to a new 2 mL tube and 40 μL of TWR buffer was added (1 μg trypsin in 0.1% triflouroacetic acid (TFA)), and the samples left to digest overnight at 37 °C. The following day, 40 μL of ABC buffer was added and the peptides were collected through centrifugation before being acidified with 10% TFA. Peptides were then desalted using C18-StageTips[112]. Briefly, two pieces of C18 filters (Empore 2215) were placed on the tips and activated with 15 μL methanol, followed by an equilibration step with 50 μL 0.1% TFA. Samples were passed through the StageTips and washed with 50 μL 0.1% TFA on the tips and subsequently eluted with 40 μL 80% acetonitrile (ACN), 0.1% TFA.

The tryptic peptides eluted from StageTips were lyophilised and resuspended in 0.1% TFA. Samples were analysed on a Q Exactive Plus mass spectrometer connected to an Ultimate Ultra3000 chromatography system (Thermo Scientific, Germany) incorporating an autosampler. Five μL of each tryptic peptide sample was loaded on an Aurora column (IonOptiks, Australia, 250 mm length), and separated by an increasing ACN gradient, using a reverse-phase 120 min gradient (from 3%–40% ACN) at a flow rate of 400 nL/min. Data was acquired with the mass spectrometer using the following settings: MS 70k resolution in the Orbitrap, 350 to 1500 precursor scan, 1.4 m/z Quad isolation; MS/MS obtained by HCD fragmentation (26% HCD collision energy), read out in the orbitrap with a resolution of 17.5k with a cycle-time of 2 seconds.

**Nanopore cDNA sequencing**. Nanopore cDNA libraries were prepared using total RNA isolated from wild type USA300 grown in TSB or human serum. Total RNA was DNase I (TURBO DNase; Thermo Fisher) treated for 1 h at 37 °C in the presence of 2 U of SUPERasin. RNA was subsequently poly(A) tailed using *E. coli* poly(A) polymerase (NEB) as per manufacturer's instructions. Afterwards, rRNA was removed using the NEBNext rRNA Depletion Kit (Bacteria) according to the manufacturer's instructions. After each enzymatic step, RNA was purified using RNAClean XP beads (Beckmann Coulter). cDNA barcoded libraries were prepared with Nanopore cDNA-PCR Sequencing kit (SQK-PCS109). The cDNA was purified without size selection in order not to exclude sRNAs. The pooled barcoded libraries were sequenced on MinION using an R9 flow cell.

**Computational analyses**

*Pre-processing of raw sequencing data*. Raw sequencing data were first processed using the pyCRAC package[113]. In particular, the CRAC_Pipeline_PE.py script was used that automates almost the entire processing pipeline. The pipeline firstly demultiplexed the raw sequencing data based on the in-read barcode sequences found in the L5 adaptors using pyBarcodeFilter.py. Flexbar (version 3.5.0) was used to then remove the 3′ adaptor sequences and any flanking nucleotides with a Phred score below 23. The reads were then collapsed to remove PCR duplicates and then aligned to either the *S. aureus* JKD6009 (https://www.ncbi.nlm.nih.gov/assembly/GCF_900607245.1/) or USA300 genome (https://bacteria.ensembl.org/Staphylococcus_aureus_subsp_aureus_usa300_fpr3757_gca_000013465/Info/Index/) using Novoalign (version 2.07). To improve the annotations of the USA300 genome, a Gene Transfer Format (GTF) file was generated. This file describes the start and end positions of all annotated sequences and what RNA class they belong to. In order to generate this file, a minimal GTF was obtained from ENSEMBL and UTR annotations were added by analysing the RNAtag-seq data with the Rockhopper2 software[48] (version 2.0.3). The pyReadCounters.py script then used the output from Novoalign to quantify the number of reads for each transcriptional unit.

*Identification of hybrids*. The hyb pipeline (version 0.0)[46] was used to detect chimeric reads as previously described[34]. Briefly, FLASH2[114] was used to merge overlapping paired reads into a single read. These merged reads were then analysed using hyb. The -anti option for the hyb pipeline was used to allow use of a genomic database. Only the uniquely annotated hybrids (.ua_hyb file) were used in subsequent analyses. When visualising hybrids using a genome browser, the.ua_hyb output file was converted to a GTF file using custom scripts. GTF files could then be converted to.sgr files using pyCRAC's pyGTF2sgr.py.

*Filtering hybrids*. To filter the list of produced hybrids for high confidence, multiple approaches were taken. To estimate the false discovery rate of any given RNA-RNA interaction, each interaction was compared to the probability that the same interaction could be generated through spurious, background ligation[33]. Briefly, the probability that a hybrid-half was matched with its pair at random, P(gx), was estimated by dividing its read depth by the total number of mapped reads (N). The background probability for any given interaction, pdf(gx, gy), was estimated by multiplying the probabilities P(gx) and P(gy). The background probability of observing any number of interactions between gx and gy, termed k, was modelled using a binomial distribution: k ~ binomial(p = pdf(gx, gy), N). These calculations were used to assign a *p*-value (i.e., P(X = k)) to each experimentally observed interaction, which were then adjusted using Benjamini-Hochberg multiple testing corrections. All interactions with a Benjamini-Hochberg adjusted p-value higher than 0.05 were discarded. Afterwards, the Vienna 2 package[53] (version 2.5.0) was

used to calculate the minimum folding energy (MFE) of each intermolecular interaction. We then created cumulative distribution plots of the CLASH data to visualise the MFE distribution. As a control, each RNA fragment was shuffled randomly over its partner RNA, or over genes belonging to the same class. Comparing the CLASH data to the shuffled control was then used to generate an MFE cut-off value for which all interactions that did not meet this minimum energy threshold were discarded. Additionally, we also utilised a list of highly curated, trans-acting sRNAs[10]. This allowed us to filter our data for interactions that contain verified sRNAs. These sRNAs are termed bona fide sRNAs. To calculate enriched structural motifs in the CLASH data, the minimum free energy structures of chimeric reads and randomly shuffled chimeric reads were calculated using hybrid-min (UNAFold) with default settings (NA = RNA, t = 37). The structures were converted into the Vienna dot-bracket notation and double-stranded structural motifs of length between 5 and 10 base pairs were extracted. Enrichment of motifs in the chimera set, compared with the shuffled chimera set, was quantified with a Fisher's exact test, and Benjamini-Hochberg correction was applied to account for multiple tests.

*Hybrid distribution plots.* Only statistically significant interactions that contained a bona fide sRNA and had an MFE of less than -10 kcal/mol were considered. pyBinCollector.py was then used to plot the mRNA reads within each interaction relative to the start codon. Each interaction was counted only once to avoid biasing the output for abundant interactions.

*RNase binding to target transcripts.* To calculate where the RNase bound its target transcripts, the pyBinCollector tool from the pyCRAC package (version 1.5.0) was used. Here, each target transcript was divided into 100 equally sized bins and the nucleotide read density for each bin was calculated and the total plotted.

*Interactome plots.* Only statistically significant interactions that contained a bona fide sRNA and had an MFE of less than -10 kcal/mol were considered. These were visualised using the iGraph Python package.

*GO term analysis.* Gene ontology analysis was performed on upregulated and downregulated genes, defined as displaying a $\log_2$ fold change of 1.5 and $p$ value below 0.05, 30 min after the shift to RPMI, LPM pH 7.6 or LPM pH 5.4. KEGG pathway and keyword analysis were performed in R with STRINGdb package[115].

*RNAtag-Seq analysis.* Following pre-processing of the data using the pyCRAC_-pipeline_PE (version 0.6.1), the output from pyReadCounters.py for each experimental condition was merged. In order to normalise the data and to account for variations in sequencing depth, the data was normalised using DESeq2[116]. Data could then be normalised to the t0 sample when desired to examine relative fold change. In order to cluster genes into common expression patterns, the 'Short Time-series Expression Miner' (STEM, version 1.3.13) was used[117]. Data was $\log_2$ normalised and clustered using the STEM clustering method with 50 model profiles to examine which genes showed changed expression following shift to RPMI or LPM.

*Mass spectrometry analysis.* MaxQuant[118] (version 1.6.17.0) was used for mass spectra analysis and peptide identification via Andromeda search engine[119]. Match between runs and LFQ were chosen. Trypsin was chosen as protease with a minimum peptide length of 7 and a maximum of two missed cleavage sites. Carbamidomethyl of cysteine was set as a fixed modification and methionine oxidation and protein N-terminal acetylation as variable modifications. Proteome databases were made using 31 toxin sequences. The first search peptide tolerance was 20 ppm, and the main search peptide tolerance was set at 4.5. Peptide spectrum match (PSM) was filtered to 1% FDR. Protein intensities were log transformed and missing values imputed. Moderate t-test was performed on $\log_2$ transformed protein intensities using the limma package[120].

*Nanopore cDNA data analysis.* Basecalling, demultiplexing and quality analysis were done with gruppy. Reads with mean quality above 7 were kept for further analysis. Next, the orientation of the reads was determined with cdna.claasifier.py (pychopper, https://github.com/nanoporetech/pychopper.git) and mapped to the USA300 genome with minimap2[121] (version2.24; https://github.com/lh3/minimap2). After mapping, sam files were converted into sorted bam files and then to bedgraphs with normalized coverage using Samtools[122] (version 1.9; http://www.htslib.org) and bedtools[123] (version 2.27.1; https://github.com/arq5x/bedtools2) respectively.

**Reporting summary**. Further information on research design is available in the Nature Research Reporting Summary linked to this article.

## Data availability
The next generation sequencing data have been deposited on the NCBI Gene Expression Omnibus (GEO) with accession number GSE166151. The mass spectrometry proteomics data have been deposited to the ProteomeXchange Consortium via the PRIDE[124] partner repository with the dataset identifier PXD025122. Source data are provided with this paper.

## Code availability
The python pyCRAC[113], GenomeBrowser packages (version 1.6.3) and the CRAC[43] and CLASH[34] pipelines used for analysing the data are available from https://git.ecdf.ed.ac.uk/sgrannem/ and pypi (https://pypi.org/user/g_ronimo/). The hyb pipeline used for identifying chimeric reads is available from https://github.com/gkudla/hyb (version 0.0). The structural motifs (version 0.1) code used for identifying enriched structural motifs (Fig. 2d) can be obtained from https://github.com/gkudla/structural_motifs. The scripts for statistical analysis of hyb data[33] is available from https://bitbucket.org/jaitree/hyb_stats/. The FLASH2 algorithm (version 1.2.11) for merging paired reads is available from https://github.com/dstreett/FLASH2. The code used for has also been uploaded to Zenodo[125] and is provided as Supplementary Software.

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

## Acknowledgements

We are grateful to Wei Gao, Tim Stinear and Benjamin Howden for providing the JKD6009 RNase III-HTF strain, Philippe Bouloc for the RsaE over-expression plasmid and Tracy Palmer for the polyclonal EsxA antibody. We would like to thank Pascale Romby and Emma Denham for fruitful discussions and helpful suggestions. We also thank Julia Wong for help with the FACS analyses, Lucas Herrgott for producing recombinant RNase III and Jimi-Carlo Wills and Alexander von Kriegsheim from the proteomics facility at the Institute of Genetics and Molecular Medicine (IGMM) at the University of Edinburgh for performing the mass-spectrometry analyses. Finally, we would like to thank the members of the Granneman lab for critically reading the manuscript. This work was supported by a Medical Research Council Non-Clinical Senior Research Fellowship (MR/R008205/1 to S.G), a Wellcome Trust grant (109093/Z/15/A to S.W.M), a Wellcome Trust Senior Research Fellowship (207507 to G.K), the labEx NetRNA framework (ANR-10-LABX-0036) and the French investment for the future framework (ANR-17-EURE-0023). J.J.T and D.G.M were supported by a grant from the National Health and Medical Research Council Australia (GNT1139313). J.R.F was funded by institute strategic grant funding ISP2: BBS/E/D/20002173 and BBS/E/D/20002174 from the Biotechnology and Biological Sciences Research Council (United Kingdom), SHIELD grant MR/N02995X/1 from the Medical Research Council (United Kingdom) and a Wellcome Trust collaborative award 201531/Z/16/Z. R.K.C and R.L.Z were supported in part by grant AI128376 from the US National Institute of Allergy and Infectious Diseases (to R.K.C.). The proteomics facility at the IGMM is supported by a Wellcome Trust Multiuser Equipment grant (208402/Z/17/Z to Alexander von Kriegsheim). For the purpose of open access, the corresponding author has applied a creative commons attribution (CC BY) licence to any author accepted manuscript version arising.

## Author contributions

S.W.M, J.J.T., R.J.F, and S.G initiated the project and S.W.M, J.J.T., R.K.C, I.C., R.J.F. and S.G designed the experiments. S.W.M, I.I, P.A, R.L.Z, N.M, L.C, D.G.M and A.C.P performed the experiments and analysed the data. R.F. and G.K developed the software for analysis of enriched structural motifs in the RNase III CLASH data. S.G developed the data analysis

pipelines for processing paired-end CLASH data. S.W.M and S.G drafted the manuscript, and all the authors reviewed the manuscript and approved the final version.

## Competing interests

The authors declare no competing interests.
