## [Peer Review File · Nature Communications]

Reviewer comments, first round review -

Reviewer #1 (Remarks to the Author):

In their manuscript 'RNase III CLASH in MRSA uncovers sRNA regulatory networks coupling metabolism to toxin expression' McKellar et al. adapt and optimize the CLASH approach to Gram-positive bacteria and apply it to *S. aureus* to map RNA-RNA interactions involving RNase III. The authors employ established culture media to mimic various *S. aureus* infection conditions, and the transcription patterns identified (by classical RNA-seq prior to CLASH) revealed a number of typical transcription hallmarks, generally confirming the suitability of these media as model systems. RNase III-CLASH (performed with two clonally unrelated MRSA strains) revealed known RNA interaction patterns which highlighted the robustness of the approach. Furthermore, numerous novel condition- and strain-dependent RNA-RNA interactions were identified from which a few were selected for detailed analysis. Here the authors mainly focused on sRNA-RNA interactions involving sRNAs RsaI, RsaE and RNAIII. Based on a limited number of validation experiments the authors propose (i) RsaI to represent a sponge for RsaE that inactivates the sRNA in concert with RNase III. Moreover, they suggest (ii) RNAIII to positively regulate the EsxA toxin through specific base-pairing and (iii) RsaE to bind to PSM transcripts and enhance translation of the cytolysins.

This is a comprehensive study of general interest revealing a wealth of both known and novel RNA-RNA interactions in *S. aureus*. The manuscript is carefully written and presented, and the study highlights the eminent role of sRNAs in both metabolic and virulence adaptation of this important pathogen. The observed growth- and strain-specific differences in sRNA expression and functions are remarkable and of great interest, although the authors do not address and discuss this aspect in the manuscript.

The first part of the study (covering the CLASH approach) is straightforward and of high quality. The data, which appear to be robust and reproducible, are supported by several bioinformatic analyses, including folding energies of RNase III-bound RNA duplexes and correlation analyses. However, the functional studies on selected sRNA interactions (presented in the second part of the paper) require attention and additional experimentation.

Major comments

RsaI sponge part

The idea of RsaI as a sponge for RsaE is intriguing. However, the data, as they stand now, are not entirely convincing and support only in part the conclusions drawn. The authors are actually aware of these issues and stay therefore a bit vague in their interpretations. By performing a few more control experiments and by paying attention to the molecular details of the putative RNA-RNA interactions, the authors should actually be able to address these concerns and shed more light on the underlying mechanism.

In this respect, the crucial point of the study is the medium-dependent processing of RsaI and RsaE which the authors identified and which is an important finding. Unfortunately, the authors neglect in subsequent experiments the presence of these various RNA species which are (at least for RsaE/RoxS) known to display very different target RNA spectra.

Although processing of RsaI and RsaE itself (which RNase(s) involved; conditions etc.) is not subject of the study, elucidating the interactions of the full-length and processed species with their

respective binding partners is key for understanding the general regulatory mechanism. Also, the fate of the RsaE mRNA targets (i.e. transcript stabilization or degradation) upon binding of either full-length of processed RsaE needs to be established in the first place before any conclusions on their up or down regulation can be drawn in the different *rsal* or *rnc* deletion mutants and media.

For doing these experiments, the authors should streamline their experimental plan and focus on one strain (e.g. USA300) to generate a set of isogenic mutants, keeping in mind that other strains (as shown for JKD6009) may behave differently regarding their RNA-RNA interaction profiles. The latter is an important point for appreciating the huge regulatory potential of *S. aureus* (as a species) and the authors should address this fact explicitly in the discussion.

Below a detailed list of questions and suggestions that may help to improve the manuscript:

1. Demonstrating specific binding of RsaE to RsaI and cleavage of the complex in vitro by RNase III is convincing (ExData Figs.10, 11). Of note, in these in vitro experiments full-length sRNA partners are employed. However, both RsaI and RsaE undergo processing which is obviously strongly condition-dependent (and which does not involve RNase III as the processing enzyme) (Figure 4a). Under the conditions tested (TSB, human serum) either full-length RsaI meets processed RsaE (TSB) or vice versa, processed RsaI will find full-length RsaE as partner (serum). So, what about the binding interaction between processed and full-length sRNA partners? Is this comparable to the full-length molecules and are the respective hybrids equal RNase III targets?
2. From the data shown it is difficult to entirely comprehend the function of RsaI as a sponge of RsaE (Figure 4). The authors state that RsaI and RsaE form hybrids that are targets of RNase III. Thus, if RsaI sponges RsaE, why do RsaE levels then remain unaffected in the *rsal* and *rnc* deletion mutants (Figure 4a)? Actually, RsaE levels are expected to go up in absence of the sponge and/or the processing enzyme? In this respect, what would happen in a *rsal/rnc* double mutant; and what would be the effect of *rsaE* deletion or RsaE overexpression on RsaI levels?
3. In 383: The statement that RsaE levels in serum are lower than in TSB is hard to judge as two different RsaE species (full-length in TSB, processed in serum) are compared (Figure 4a,b). The authors rightly cite previous work showing that such RsaE isoforms are functionally different and may interact with different targets, making a direct comparison difficult, if not inappropriate. Accordingly, the statements in lines 383 and 386 regarding relative RsaE and RsaI levels in the different media are daring. As the authors consider the relative expression ratios as crucial for the RsaI sponge function this issue needs to be addressed.
4. In Figure 4c, the authors show relative expression levels of RsaE targets in various media (in comparison to wildtype). But, what about their absolute transcription? Particularly for the TCA cycle genes, differences in transcript levels between rich medium (TSB) and human serum can be expected. In this respect, if basal transcription is generally low, relative changes are difficult to interpret.
5. Further to Figure 4c: In *rsal* and *rnc* mutant backgrounds, the authors find RsaE targets almost unaffected or slightly increased when bacteria were grown in TSB, while in human serum RsaE target levels are significantly reduced. Apart from the undefined basal RsaE target transcription levels (#4), do the tested RsaE targets actually interact with processed or full-length RsaE or both? In conjunction with the control experiment on RsaI/RsaE isoform interactions suggested above (#1), this information is crucial to appreciate the alleged RsaI sponge function under the various conditions and the subsequent effects on the RsaE targets.
6. What happens to the RsaE targets if RsaE is overexpressed in the *rsal* and *rnc* deletion mutants?
7. Are the RsaE/mRNA target complexes subject to RNase III cleavage too? They do not show up in the sRNA-mRNA CLASH data list, but *citC*, *sucC*, *fumC* and *gcvPA* have been previously identified as RNase III targets (Lioliou et al. 2012). Does this mean the transcripts are protected by RsaE from

RNAse III-mediated cleavage/degradation? Again, adding the isogenic USA300 *rsaE* deletion mutant as control would be very informative (see also #2) to gain insight into the general fate/turnover of the RsaE targets upon binding of the sRNA; the more so as this mutant is already available to the authors.

8. In the in vitro binding experiments (ExData Figs.10, 11), the authors used Ca²⁺ to successfully inhibit RNAse III enzyme activity. However, the Ca²⁺ concentration used is not indicated. In human serum, normal Ca²⁺ concentration is in the range of 1-1.3 mmol/l. Would this be sufficient to influence intracellular Ca²⁺ concentrations in the bacteria and inhibit RNAse III? In other words, is the enzyme generally inactive upon growth of *S. aureus* in human serum (but still able to bind targets)?

Toxin expression part

9. Figure 5e: Please show haemolytic activity of the WT strain.

10. Figure 5e: Why did the authors use an RNase III knockout in yet another (third) strain background? Please, demonstrate the effect in the USA300 background for better comparability.

11. What about the other known phenotypes associated with PSMs such as phagosomal escape and detachment from biofilms? Are they also affected in the RsaE deletion strain?

12. L413-414: The authors state that the RsaE interactions with α/β psm transcripts were observed in JKD6009, but the in vivo follow-up experiments (Figure 5e-f) were done in USA300. Please clarify.

13. Figure 5d: Given the predictions in 5c and the detected changes for PSM α 1 and PSM α 4, but not for PSM α 3 by Mass-spec analysis (Ext data Fig. 13), why was only the interaction for psm α 3 tested?

14. Fig 6D: In this experiment, a gfp vector control without the *esxA* 5'-UTR fragment should be included.

Minor comments:

15. Please check RNA marker sizes. For example, the size of RsaE (which is 102 nt) appears to be too small compared to the 100-nt marker indicated on the gels (both in the manuscript and the source data).

16. Ext data, pg 2 'RsaE targets': Here *S. aureus* JKD6008 is mentioned. Is this a typo or was indeed this VISA-derivative of JKD6009 used in the experiments?

17. Ext. data Fig. 6. How do the authors explain the major differences in the interactome between the two strains used? (see also general comment above).

18. Further to this: Please, state briefly why these two strains were chosen (L95-96). For example, did the authors expect to find fundamentally different interactions due to known differences in sRNA regulation or because the strains are associated with different disease outcomes/virulence?

19. L179: Here the authors specifically mention difficulties with existing annotations but fail to actually list the annotation that was used for either strain in the manuscript. For example, for JKD6009, was the updated annotation from Mediati et al. (parallel submission) used?

20. Supplementary tables 1 and 2: These tables need a comprehensive legend describing what can be found in each column.

21. L188: 'Many of the predicted sRNA-target interactions in USA300 had poorer folding energies compared to chimeras identified in JKD6009 ...' What is the potential reason for this discrepancy? This may feed back to the questions why the two strains were used in the first place.

22. L51: Please, replace 'spp.' with 'enterica' (there is plenty of research in different serovars but not in different species)

23. L181: Please correct typo in 'regions'

Reviewer #2 (Remarks to the Author):

In their manuscript, McKellar and collaborators studied the sRNA regulatory networks using CLASH to address an important question: what is the biological function of sRNAs in *Staphylococcus aureus*? Because Hfq is dispensable for sRNA-mRNA interactions in this bacterium, the authors adapted CLASH using RNase III as a bait, based on the assumption that this double-strand endoribonuclease is involved and plays a key role in sRNA-mediated regulation. First, the authors described how they set up the method and perform CLASH under different in vitro conditions that mimic the environment encountered during infection. They used RPMI to mimic serum, and LPM and pH 7.6 or 5.4 to study nutritional adaptation and acidic stress. CLASH was performed in two different strains (USA300 and JKD6009) although the latter was used only in TSB and RPMI. Overall, there is a long description of the validation of CLASH to prove that the use of RNase III as a bait enable robust RNA-RNA interactions. Then, there are several short stories dealing with RNA-RNA interactions. RsaI is shown to regulate RsaE, thus having an impact on RsaE targets in human serum. RsaE is shown to regulate PSM production although the mechanism of action is not understood. The overall effect of RsaE onto PSM production does not follow the canonical mode of action as the authors described activation of expression instead of repression while the RBS seems targeted. Finally, the authors describe a novel target for RNAIII, which is EsxA toxin. Although there is a huge amount of data and the interplay between RsaI and RsaE interesting, the message is diluted and the manuscript sometimes looks like a collection of small stories without an ending point or a phenotype. Some of the data presented are not all the time convincing and therefore, the manuscript lacks of a clear characterization and understanding of the mechanisms described.

General comment:

Because of the use of two genetic backgrounds, it is sometimes difficult to follow the study and the interpretation. These two strains are substantially different as they do not belong to the same ST.

Line 187: The authors refer to supplementary tables 1 and 2 in the text, but it is likely 3 and 4 instead.

Line 292: The study on SprA1/SprA1as did not prove, to my knowledge, that RNase III is involved in their degradation. The sentence must be modified accordingly.

Figure 3c; bottom. It is not written on each heatmap, whether data were obtained from USA300 or JKD6009. This would help the reader.

Figure 4: This figure and the text is relatively difficult to follow. In panel A the cells are grown for 3h while in panel D RNAs were extracted at an OD of around 3. On figure 4d, we barely see the two lengths of RsaE and RsaEp which is mentioned in the legend does not appear anywhere. On line 397/398, the lanes described in the text do not correspond to the ones in the figure 4d. Some experiments are done in serum and other in TSB which increase complexity to follow. For instance, in the conclusion of the section (from line 404 to 409), it seems that the authors describe data from human serum while statement on line 407 refer to TSB condition. Why the authors did not test the overexpression of RsaI in human serum instead of TSB in figure 4f (or at least in the two media)? To distinguish the two forms of RsaE, qPCR using specific primers could be employed. A mapping of the extremities of the two forms would be relevant especially as the authors mentioned a length of around 75 nts in the discussion. Then, does the modification of the RsaE targets level has an impact

of their encoded proteins?

Extended data fig.10 c: The ESMA presented is not really convincing.

Figure 5d: The retardation is not strong. There is no mention to the amount or concentrations added which renders difficult to have an overall idea of what could be the K_d (even though I agree that we cannot calculate it here based on the low affinity in vitro). What does the stars mean?

Regarding the regulation of toxin production by RsaE: The GFP-reporter assay suffers from the lack of sufficient information. It is not clear why the construct is cytotoxic. How many codons that belong to PSM were cloned in the plasmid? Restricting to a few codons (perhaps only the initiation codon) might help. The interaction is apparently at the RBS, therefore doing the fusion at the AUG should not be a problem. Did the authors tried toeprint assay or in vitro translation assay? It would be of interest to verify whether the modulation of toxin production as an impact on virulence. What about PSM production in *rnc* mutant ?

The stabilizing effect of RNAIII onto *esxA* mRNA is interesting as the GFP-reporter experiment is convincing (even though EMSA does not show a strong affinity in vitro). However, one would have appreciated to have more physiological data to end the story. Also, it raises the question of the potential role of RNase III. This interaction was found by CLASH using RNase III as a bait, which implies degradation of the target. Here, we have a stabilization by RNAIII. What is the *esxA* RNA level in a *rnc* mutant strain?

Line 181: Change 'reegions' by 'regions'.

Line 654: needs rewriting.

Reviewer #3 (Remarks to the Author):

Review for "RNase III CLASH in MRSA uncovers sRNA regulatory networks 1 coupling metabolism to toxin expression", McKellar et al.

Summary

McKellar et al., describe the role and function of sRNAs in mediating MRSA virulence. Using CLASH, they could identify a number of known, as well as hundreds of novel sRNA-RNA interactions when mimicking the host environment. The authors suggest that the production of small membrane-permeabilizing toxin, whose expression is linked to metabolism, is strongly regulated by sRNA. Among the identified sRNA-RNA interactions, they have uncovered two sRNAs, namely RNAIII and RsaE, which regulate the expression of at least 4 cytolytic toxins, which play a role in MRSA-mediated virulence. Overall, a very interesting study with several novel and potentially clinically relevant findings backed up by sound validations. CLASH data generation, adaptation of the original protocols and data analysis is well done.

General remarks:

- The biology of sRNAs in Gram-positive species is not well described (regulation of sRNA-target interaction might differ mechanistically in gram- and gram+). Given the broad readership of Nature Communications, a short introduction into this might be useful.
- Along the same lines, a general scheme of sRNA biogenesis/functioning in bacteria should be shown, which also includes RNase III, the bait protein chosen for CLASH analysis.
- The authors could elaborate in more detail on the clinical potential of targeting sRNAs rather than using broad statements such as "our data has scope for further impact" (last sentence, discussion).

Specific points:

- The composition of RPMI and human blood is rather different. It has apparently been shown

before (ref 19) that bacteria grown in RPMI have a similar transcriptional profile than in human blood. What means “similar” here? It is surprising that the transcriptional profiles are similar as the composition of blood and RPMI is rather different. Are the authors sure that RPMI is a good choice here especially as there are more physiological media available (DOI:<https://doi.org/10.1016/j.cell.2017.03.023> and others). This reviewer is not an expert in microbiology but the other study (Mediati et al.) uses liquid Mueller-Hinton. Might this be a more physiological medium?

- The resolution of Fig. 2D should be improved as it is hardly readable.
- Can the authors explain why mRNA-mRNA interactions are predominant in the CLASH data?
- Independent of RNase III activity, different species of RsaI and RsaE appear in the different media with different transcript lengths. Can the authors speculate how the medium composition might influence transcript length? If so, a highly physiological medium mimicking the composition of human blood as closely as possible is even more important, especially for validation experiments. In cancer cells, the length of 3' UTR regions of tumor suppressor genes can be shortened so that less miRNAs bind and down-regulate their target. Could a similar mechanism be active in bacteria, i.e. *S. aureus*?

We would like to thank the reviewers for their constructive comments and very helpful suggestions.

Reviewer #1

In their manuscript 'RNase III CLASH in MRSA uncovers sRNA regulatory networks coupling metabolism to toxin expression' McKellar et al. adapt and optimize the CLASH approach to Gram-positive bacteria and apply it to *S. aureus* to map RNA-RNA interactions involving RNase III. The authors employ established culture media to mimick various *S. aureus* infection conditions, and the transcription patterns identified (by classical RNA-seq prior to CLASH) revealed a number of typical transcription hallmarks, generally confirming the suitability of these media as model systems. RNase III-CLASH (performed with two clonally unrelated MRSA strains) revealed known RNA interaction patterns which highlighted the robustness of the approach. Furthermore, numerous novel condition- and strain-dependent RNA-RNA interactions were identified from which a few were selected for detailed analysis. Here the authors mainly focused on sRNA-RNA interactions involving sRNAs RsaI, RsaE and RNAIII. Based on a limited number of validation experiments the authors propose (i) RsaI to represent a sponge for RsaE that inactivates the sRNA in concert with RNase III. Moreover, they suggest (ii) RNAIII to positively regulate the EsxA toxin through specific base-pairing and (iii) RsaE to bind to PSM transcripts and enhance translation of the cytolysins.

This is a comprehensive study of general interest revealing a wealth of both known and novel RNA-RNA interactions in *S. aureus*. The manuscript is carefully written and presented, and the study highlights the eminent role of sRNAs in both metabolic and virulence adaptation of this important pathogen. The observed growth- and strain-specific differences in sRNA expression and functions are remarkable and of great interest, although the authors do not address and discuss this aspect in the manuscript.

The first part of the study (covering the CLASH approach) is straightforward and of high quality. The data, which appear to be robust and reproducible, are supported by several bioinformatic analyses, including folding energies of RNase III-bound RNA duplexes and correlation analyses. However, the functional studies on selected sRNA interactions (presented in the second part of the paper) require attention and additional experimentation.

Major comments:

1) RsaI sponge part

The idea of RsaI as a sponge for RsaE is intriguing. However, the data, as they stand now, are not entirely convincing and support only in part the conclusions drawn. The authors are actually aware of these issues and stay therefore a bit vague in their interpretations. By performing a few more control experiments and by paying attention to the molecular details of the putative RNA-RNA interactions, the authors should actually be able to address these concerns and shed more light on the underlying mechanism.

In this respect, the crucial point of the study is the medium-dependent processing of RsaI and RsaE which the authors identified and which is an important finding. Unfortunately, the authors neglect in subsequent experiments the presence of these various RNA species which are (at least for RsaE/RoxS) known to display very different target RNA spectra. Although processing of RsaI and RsaE itself (which RNase(s) involved; conditions etc.) is not subject of the study, elucidating the interactions of the full-length and processed species with their respective binding partners is key for understanding the general regulatory mechanism.

Also, the fate of the RsaE mRNA targets (i.e. transcript stabilization or degradation) upon binding of either full-length or processed RsaE needs to be established in the first place before

any conclusions on their up or down regulation can be drawn in the different *rsaI* or *rnc* deletion mutants and media.

For doing these experiments, the authors should streamline their experimental plan and focus on one strain (e.g. USA300) to generate a set of isogenic mutants, keeping in mind that other strains (as shown for JKD6009) may behave differently regarding their RNA-RNA interaction profiles. The latter is an important point for appreciating the huge regulatory potential of *S. aureus* (as a species) and the authors should address this fact explicitly in the discussion.

Below a detailed list of questions and suggestions that may help to improve the manuscript:

1. Demonstrating specific binding of RsaE to RsaI and cleavage of the complex *in vitro* by RNase III is convincing (ExData Figs.10, 11). Of note, in these *in vitro* experiments full-length sRNA partners are employed. However, both RsaI and RsaE undergo processing which is obviously strongly condition-dependent (and which does not involve RNase III as the processing enzyme) (Figure 4a). Under the conditions tested (TSB, human serum) either full-length RsaI meets processed RsaE (TSB) or vice versa, processed RsaI will find full-length RsaE as partner (serum). So, what about the binding interaction between processed and full-length sRNA partners? Is this comparable to the full-length molecules and are the respective hybrids equal RNase III targets?

Response: To further examine the processing of RsaE and RsaI and to identify the RNA species present in TSB and human serum, we have now performed Nanopore sequencing on total RNA extracted from cells grown in these conditions. The manuscript has been updated with these data (Fig. 4), which are now discussed on page 11 of the revised manuscript. This revealed that the polyU tracts at the 3'-end of these sRNAs, which are part of the intrinsic transcription terminators, were exonucleolytically trimmed in response to the nutritional environment. This was particularly prominent for RsaE in TSB. These trimming events result in heterogeneous subpopulations of RsaE and RsaI, varying from the full-length transcript to trimmed subspecies up to 7 nucleotides shorter. This was to our surprise as we had previously hypothesised that RsaE and RsaI processing in TSB and serum would be at the 5'-end to free the sRNA seed sequence, as reported in *S. epidermidis* RsaE and *B. subtilis* RoxS. We evidently overestimated the degree of processing.

To the best of our knowledge, 3'-end trimming of sRNAs has not been reported previously in Gram-positive bacteria and we have discussed potential regulatory outcomes for this, and potential RNases involved in the Discussion (page 17). As these RNA subspecies only vary in the length of their U-rich sequences in the intrinsic terminators, which have never been described to be involved in any base-pairing interactions, we do not envisage that this has any effect on the ability of the sRNA seed sequence to base-pair with other molecules, including the interaction between RsaI and RsaE. In fact, all our EMSAs and cleavage assays were performed with an RsaE transcript lacking the terminator sequence, demonstrating that it is not essential for binding RsaI *in vitro*. As such, we believe that cleavage assays on RsaE/RsaI subspecies will not provide any new insights. We do believe that this 3'-end trimming might impact the stability of RsaE and that the shorter versions are more stable. This is also mentioned on page 17 of the revised manuscript in the Discussion section.

2. From the data shown it is difficult to entirely comprehend the function of RsaI as a sponge of RsaE (Figure 4). The authors state that RsaI and RsaE form hybrids that are targets of RNase III. Thus, if RsaI sponges RsaE, why do RsaE levels then remain unaffected in the *rsaI* and *rnc* deletion mutants (Figure 4a)? Actually, RsaE levels are expected to go up in absence of the sponge and/or the processing enzyme? In this respect, what would happen in a *rsaI/rnc* double mutant; and what would be the effect of *rsaE* deletion or RsaE overexpression on RsaI levels?

Response: We agree that our observation that the RsaI-RsaE duplex can be cleaved by RNase III *in vitro* would imply that the duplex is, by default, degraded. As such, one would then expect that RsaE levels would increase upon RsaI or RNase III deletion. We clearly do not see this. This has puzzled us for some time and there is no simple answer to this. We favour a model where RNase III here mostly acts as an RNA chaperone. It is possible that *in vivo* only a small fraction of the RsaI-RsaE duplex is cleaved by RNase III. We showed that the duplex can be specifically cleaved by RNase III *in vitro*, however, the efficiency appears to be low. It has been proposed that RNase III can function as a non-catalytic RNA-binding protein (Calin-Jageman and Nicholson, NAR 2003). Thus, we speculate that the RsaE-RsaI duplex is largely resistant to RNase III degradation, but it is still efficiently bound by the protein. Therefore, we hypothesise that main function of RNase III here is to stabilise the RsaI-RsaE interaction *in vivo* or by preventing RsaE from base-pairing with other targets. We now discuss this on page 16 of the revised manuscript.

With respect to the proposed sponging activity, several groups have shown that sponging of an sRNA by another does not necessarily have to result in degradation of the target sRNA. For example, SprY binds to RNAIII and inactivates it but this interaction does not alter the stability of RNAIII (Le Huyen et al NAR 2021; see Figueroa-Bossi and Bossi, 2018 and Denham 2020 for further support of this definition). Therefore, we believe that RsaI (Figs 4 and 5) in that sense functions similarly as SprY.

We have also performed the experiment suggested by the reviewer to examine RsaE levels in a Δ rsaI- Δ rnc double mutant and included the data in the revised manuscript (Supplementary Figure 13a). These deletions had no noticeable effect on RsaE stability in TSB or human serum, again contributing to the idea that RsaI primarily regulates RsaE activity, not stability.

Even though RsaI does not induce RsaE degradation, deletion of RsaI led to downregulation of RsaE targets in comparison to WT when cells were grown in human serum (Fig. 5a). This indicates that RsaE has increased activity in the absence of RsaI. Collectively, these data justify the idea of RsaI acting as a sponge of RsaE. We have reworked the description of these results in the Discussion (pages 16 and 17).

To investigate if RsaE can regulate the activity of RsaI, as suggested by the reviewer, we analysed RsaI levels in strains lacking RsaE and in strains in which we overexpressed RsaE under the control of pAmiA promoter of pICS3 plasmid. We have used a RsaE sequence which includes the both polyU terminators found in the RsaE gene. Upon successful transformation of wild type and RNase III mutant USA300 cells, we observed reduced growth of the cells. This is consistent with the previously observed cytotoxic effect of RsaE overexpression (Bohn et al., Nucleic Acids Research 2010). Despite this, we were able to extract total RNA from cells grown in TSB and human serum with and without the plasmid for RsaE overexpression. Northern blot analysis showed only a modest increase in RsaE (around 1.5- to 2-fold) when over-expressed from the pICS3 plasmid (Supplementary Figure 13c, lanes 3 and 4). No significant changes in the levels of RsaI was observed when RsaE was expressed from this plasmid. The same was observed in the Δ rsaE strain (Supplementary Figure 13c, lanes 5 and 6). Overexpressing RsaE in the Δ rsaE background revealed that this plasmid was unable to induce high levels of RsaE expression (Supplementary Figure 13, lanes 7 and 8 and quantification of the results). It is possible that with this plasmid we were unable to express RsaE at high enough levels to impact RsaI levels. However, it is important to point out that over-expression of RsaE from the same plasmid significantly (1.5-2-fold) increased the haemolytic activity of USA300 (Figure 6e). Thus a ~2-fold increase of RsaE level is sufficient to alter toxin production.

To get around the toxicity issue and to see if we could express RsaE at higher levels, we used a previously described tetracycline-inducible RsaE pRMC2 construct (kindly provided by

Philippe Bouloc; Rochat et al NAR 2018). We induced RsaE expression for 15 minutes in USA300, resulting in high over-expression of RsaE. Despite this, we could not detect strong changes in the levels for the RsaE targets (see new Supplementary Fig. 13e). Therefore, it appears that we are unable to express RsaE in USA300 at the levels required to significantly impact its target levels. However, we should note that RsaE expressed from this plasmid runs higher than endogenous RsaE.

Although the RsaE over-expression did not give satisfactory results, what is interesting from these data is that we found that RsaE is strongly regulated in human serum, even when overexpressed from plasmids using two different promoters. This suggests that RsaE levels are post-transcriptionally regulated in human serum.

3. In 383: The statement that RsaE levels in serum are lower than in TSB is hard to judge as two different RsaE species (full-length in TSB, processed in serum) are compared (Figure 4a,b). The authors rightly cite previous work showing that such RsaE isoforms are functionally different and may interact with different targets, making a direct comparison difficult, if not inappropriate. Accordingly, the statements in lines 383 and 386 regarding relative RsaE and RsaI levels in the different media are daring. As the authors consider the relative expression ratios as crucial for the RsaI sponge function this issue needs to be addressed.

Response: Following up from point 1, we now know that these various isoforms of RsaE are truncates of the 3'-end. While these species of RsaE differ in their length it is unlikely that to influence their seed sequence motif and therefore to affect their functionality. As such, we have quantified all these species as a collective of 'RsaE' or 'RsaI'. We also repeated the experiments and ran the RNA samples on lower percentage polyacrylamide gels to make the bands more compact and quantified the results (Supplementary Figure 13a and c). These results again show a strong and significant reduction in RsaE levels in human serum.

4. In Figure 4c, the authors show relative expression levels of RsaE targets in various media (in comparison to wildtype). But, what about their absolute transcription? Particularly for the TCA cycle genes, differences in transcript levels between rich medium (TSB) and human serum can be expected. In this respect, if basal transcripton is generally low, relative changes are difficult to interpret.

Response: We have analysed the mRNA levels of all tested RsaE targets in the WT strain during growth in TSB and human serum (see new Supplementary Figure 13b). This revealed that there are no significant differences in the expression of these mRNAs in the two growth conditions. We have addressed this on page 11 and 12 of the revised manuscript. Thus, the changes we observe in the Δ rsaI and Δ rnc mutants are a result of these deletions, not because the mRNA targets are expressed at very different levels.

5. Further to Figure 4c: In rsaI and rnc mutant backgrounds, the authors find RsaE targets almost unaffected or slightly increased when bacteria were grown in TSB, while in human serum RsaE target levels are significantly reduced. Apart from the undefined basal RsaE target transcription levels (#4), do the tested RsaE targets actually interact with processed or full-length RsaE or both? In conjunction with the control experiment on RsaI/RsaE isoform interactions suggested above (#1), this information is crucial to appreciate the alleged RsaI sponge function under the various conditions and the subsequent effects on the RsaE targets.

Response: As described above, we no longer propose that RsaE is processed at the 5'-end to regulate its base-pairing potential with its target mRNAs. Since the difference between the processed and unprocessed RsaE is only a few nucleotides and does not involve seed sequences, we feel it is safe to assume that both the processed and unprocessed RsaE can interact with the same mRNA substrates. Having now performed many Northern blots on RNA from TSB and human serum, we reproducibly see that RsaE is much more abundant in TSB

compared to serum. We do believe that the shorter RsaE form is able to interact with the RsaI. Our EMSAs were performed with an RsaE transcript lacking the terminator sequences. Secondly, overexpression of RsaI in TSB (where the shorter RsaE form is most abundant) significantly increased the levels of RsaE targets (Figure 5b and c). Therefore, we favour the model that RsaE is expressed at such high levels in TSB that RsaI is unable to make a significant impact on RsaE mRNA targets in this medium.

6. What happens to the RsaE targets if RsaE is overexpressed in the *rsaI* and *rnc* deletion mutants?

RsaE over-expression in TSB should reduce RsaE target levels irrespective of RsaI because RsaI does not impact RsaE activity in TSB unless it is overexpressed. In serum the levels should go further down unless this is unfavourable for the cell. To study the role of RsaI in sponging RsaE, we have measured the expression changes of RsaE targets in *rsaI*- and RNase III deletion strains in TSB and human serum. The RsaE targets were selected based on previously published papers and our hypothesis is that in absence of RsaI or RNase III there will be no sponging of RsaE, and RsaE will be free to downregulate the levels of its targets. Indeed, we observed strong and significant reduction of RsaE targets in human serum but not in TSB. Considering the issues with overexpressing RsaE and that we already observe a strong and specific effect upon deletion of RsaI and RNase III in the regulation of RsaE targets, we do not think the proposed experiments will be informative.

7. Are the RsaE/mRNA target complexes subject to RNase III cleavage too? They do not show up in the sRNA-mRNA CLASH data list, but *citC*, *sucC*, *fumC* and *gcvPA* have been previously identified as RNase III targets (Lioliou *et al.* 2012). Does this mean the transcripts are protected by RsaE from RNase III-mediated cleavage/degradation? Again, adding the isogenic USA300 *rsaE* deletion mutant as control would be very informative (see also #2) to gain insight into the general fate/turnover of the RsaE targets upon binding of the sRNA; the more so as this mutant is already available to the authors.

Response: It is difficult to compare the study carried out by Lioliou *et al.*, 2012, and our own. Lioliou *et al.* used the RN6390 strain and performed their RIP experiments at low optical densities in BHI (rich) medium. They induced RNase III expression at OD₆₀₀ 0.2-0.3 for 90 minutes. This is likely to give a final experimental OD of 0.8-1.2, assuming a 30-minute doubling time. In comparison, we performed our CLASH experiments in the USA300 and JKD6009 backgrounds at OD₆₀₀ 3 in TSB and RPMI medium.

It is known that deletion of RsaE leads to upregulation in its mRNA targets (Geissmann *et al.*, 2009) and that overexpression of RsaE downregulates them (Bohn *et al.*, 2018). Thus, it can be concluded that RsaE binding to its target mRNAs does not lead to RNase III protection, as suggested by the reviewer, but possibly their cleavage. However, these complexes may be degraded too quickly to be captured by RNase III CLASH or may occur during growth stages different to those used in our experiments. This would explain why we did not identify them in our CLASH data.

Performing *in vitro* RNase III cleavage assays between RsaE and its mRNA targets is a significant endeavour and beyond the scope of this manuscript as we are mainly focusing on the RsaI sponging activity. Additionally, examination of the expression of RsaE targets upon RsaE deletion has already been published in previous work (Geissmann *et al.*, 2009), as described above.

8. In the *in vitro* binding experiments (ExData Figs.10, 11), the authors used Ca²⁺ to successfully inhibit RNase III enzyme activity. However, the Ca²⁺ concentration used is not indicated. In human serum, normal Ca²⁺ concentration is in the range of 1-1.3 mmol/l. Would this be sufficient to influence intracellular Ca²⁺ concentrations in the bacteria and inhibit

RNase III? In other words, is the enzyme generally inactive upon growth of *S. aureus* in human serum (but still able to bind targets)?

Response: We used 10 mM CaCl₂ for our cleavage experiments and therefore we believe that RNase III should still have some activity in serum where the concentration around 2-3 mM. We have now added details on our calcium concentrations in the Supplementary Methods (page 2) and discussed this point in the Discussion on page 16.

Toxin expression part

9. Figure 5e: Please show haemolytic activity of the WT strain.

Response: This is the first bar of the now Fig 6e and f. We have changed the labelling in Fig. 6e to make it easier to interpret.

10. Figure 5e: Why did the authors use an RNase III knockout in yet another (third) strain background? Please, demonstrate the effect in the USA300 background for better comparability.

Response: The AH1263 strain a derivative of USA300 that has been cured of a natural plasmid This was an RNase III transposon mutant that was available from a mutant library. However, we have now repeated the experiment using a clean RNase III deletion mutant made in the USA300 LAC background, which we have used for all our validation experiments. The new haemolysis assay results are included in the revised Figure 6e. We obtained identical results with the new USA300 LAC RNase III deletion strain, supporting our original conclusions.

11. What about the other known phenotypes associated with PSMs such as phagosomal escape and detachment from biofilms? Are they also affected in the RsaE deletion strain?

Response: This is an excellent question, and we are addressing this now. However, because RsaE regulates so many targets, performing macrophage/neutrophil infection or biofilm detachment assays with an RsaE deletion mutant is not going to be very informative as we cannot exclude the possibility that any phenotypical changes are directly linked to changes in α PSM levels. For example, in *S. epidermis*, RsaE influences biofilm formation at multiple levels by enhancing the production of PNAGs (by repressing IcaR) and eDNA release (Schoenfelder et al PLOS path. 2019). Hence, we are now making several α PSM operon mutants that can no longer be regulated by RsaE (by mutating part of the base-pairing interaction sites) or in which the AUGs of *psm α 1* and *psm α 4* are mutated so that we can replicate the effect of what we observed in the RsaE deletion mutant on α PSM production, but without having to delete RsaE. However, this is an entirely new project on its own and with the infection and biofilm analyses would take several years to complete.

12. L413-414: The authors state that the RsaE interactions with a/bpsm transcripts were observed in JKD6009, but the in vivo follow-up experiments (Figure 5e-f) were done in USA300. Please clarify.

Response: We performed the validations in USA300 as this strain is known to produce very high levels of PSMs (Berlon *et al*, 2015, Journal of Infection). Thus, we anticipated that performing the validation studies in this strain would be easier as the culture supernatants would have much stronger haemolysis activity. The interactions between RsaE and the PSMs were identified in the JKD6009 strain through CLASH. However, we also identified interactions between sRNAs and PSMs in USA300, such as between RNAIII with *psm α 4*.

13. Figure 5d: Given the predictions in 5c and the detected changes for PSMa1 and PSMa4,

but not for PSMa3 by Mass-spec analysis (Ext data Fig. 13), why was only the interaction for *psma3* tested?

Response: This is because the *psma3* interaction was detected in the CLASH data and because it is the most cytolytic toxin. During the revision period we also performed EMSAs with RsaE and fragments of *psma1* and *psma4* but we were unable to detect any significant complex formation. We now state this on pages 13 and 14 of the revised manuscript. In some experiments we did manage to see a very weak interaction between *psma2* and RsaE, but this was not reproducible and therefore we are hesitant to report these findings.

14. Fig 6D: In this experiment, a gfp vector control without the *esxA* 5'-UTR fragment should be included.

We have now included new FACS data for two GFP controls (Supplementary Figure 16). Here, showed that expression of RNAIII does not impact the produced fluorescence of GFP or a *gyrB*-GFP fusion. We chose *gyrB* as this is not a known target of RNAIII, and thus these experiments demonstrate that the promotion of *esxA*-GFP fluorescence by RNAIII is specific. We now discuss these data on page 15 of the revised manuscript

Minor comments:

15. Please check RNA marker sizes. For example, the size of RsaE (which is 102 nt) appears to be too small compared to the 100-nt marker indicated on the gels (both in the manuscript and the source data).

Response: We have checked the gels again and updated the figure accordingly.

16. Ext data, pg 2 'RsaE targets': Here *S. aureus* JKD6008 is mentioned. Is this a typo or was indeed this VISA-derivative of JKD6009 used in the experiments?

Response: This was indeed a typo and we have edited the manuscript accordingly. Thank you for catching this error.

17. Ext. data Fig. 6. How do the authors explain the major differences in the interactome between the two strains used? (see also general comment above).

Response: We believe this is because both strains have different growth rates and because they reach stationary phase at different densities. We have measured the growth of all the strains we have used and found that USA300 grows to much higher optical densities than the JKD6009 strain. Therefore, we think that the differences in growth dynamics may explain the differences in the interactomes. We have now included this data in Supplementary Table 5 and discuss these findings on page 20 of the revised manuscript.

18. Further to this: Please, state briefly why these two strains were chosen (L95-96). For example, did the authors expect to find fundamentally different interactions due to known differences in sRNA regulation or because the strains are associated with different disease outcomes/virulence?

Response: We did not initially expect such large differences in the identified interactomes. However, we utilised two strains as we reasoned interactions identified in both would be more likely to be real. This guided our decision to investigate the RsaI-RsaE and RsaA-RNAIII interactions, which were identified in both strains. USA300 is also believed to be much more virulent. However, the Reviewer is correct to point out the remarkable differences between the found interactomes, and we have mentioned this in the Discussion (page 20; also see response to comment 17).

19. L179: Here the authors specifically mention difficulties with existing annotations but fail to actually list the annotation that was used for either strain in the manuscript. For example, for JKD6009, was the updated annotation from Mediati et al. (parallel submission) used?

Response: We did not use the annotation from Mediati *et al* as we had already finished our analyses of the JKD data and had started to focus more on USA300 as a model strain. To improve the USA300 annotation file we downloaded a GTF annotation file from ENSEMBL (https://bacteria.ensembl.org/Staphylococcus_aureus_subsp_aureus_usa300_fpr3757_gca_000013465/Info/Index/). We subsequently used Rockhopper and our RNAseq data to map potential UTRs. We now discuss this in more detail on page 7 of the revised manuscript as well as in the Supplementary Methods in the Supplementary Data File. We also uploaded this revised USA300 annotation to GEO.

20. Supplementary tables 1 and 2: These tables need a comprehensive legend describing what can be found in each column.

Response: we have now included detailed legends for the Supplementary Tables on pages 34 and 35 of the Supplementary Data.

21. L188: 'Many of the predicted sRNA-target interactions in USA300 had poorer folding energies compared to chimeras identified in JKD6009 ...' What is the potential reason for this discrepancy? This may feed back to the questions why the two strains were used in the first place.

Response: when inspecting the USA300 CLASH data, we found many hybrids with poor folding energies. Manual examination found these to be repetitive, AT-rich sequences that mapped to multiple annotated features in the USA300 genome, particularly around ribosomal and transfer RNA species. We suspect that these are not actually intermolecular hybrids but potentially the result of poor genome annotation.

To overcome this, we filtered our interactions for those with a minimum folding energy (MFE) of -10 kcal/mol and containing a *bona fide* sRNA. For reference, we found that our positive control interactions (those already validated by the field) always exhibited a minimum MFE of around -20 kcal/mol with a median of -30 kcal/mol (Fig 2a). This filtering resulted in our USA300 hybrids having similar MFE distributions as known interactions (Figure 2a). We detail these analyses and the rationale behind them on pages 7 and 8 of the revised manuscript.

22. L51: Please, replace 'spp.' with 'enterica' (there is plenty of research in different serovars but not in different species)

Response: we have updated this in the text.

23. L181: Please correct typo in 'regions'

Response: we have updated this in the text.

Reviewer #2 (Remarks to the Author):

In their manuscript, McKellar and collaborators studied the sRNA regulatory networks using CLASH to address an important question: what is the biological function of sRNAs in *Staphylococcus aureus*? Because Hfq is dispensable for sRNA-mRNA interactions in this bacterium, the authors adapted CLASH using RNase III as a bait, based on the assumption that this double-strand endoribonuclease is involved and plays a key role in sRNA-mediated regulation.

First, the authors described how they set up the method and perform CLASH under different in vitro conditions that mimic the environment encountered during infection. They used RPMI to mimic serum, and LPM and pH 7.6 or 5.4 to study nutritional adaptation and acidic stress. CLASH was performed in two different strains (USA300 and JKD6009) although the latter was used only in TSB and RPMI. Overall, there is a long description of the validation of CLASH to prove that the use of RNase III as a bait enable robust RNA-RNA interactions. Then, there are several short stories dealing with RNA-RNA interactions. RsaI is shown to regulate RsaE, thus having an impact on RsaE targets in human serum. RsaE is shown to regulate PSM production although the mechanism of action is not understood. The overall effect of RsaE onto PSM production does not follow the canonical mode of action as the authors described activation of expression instead of repression while the RBS seems targeted. Finally, the authors describe a novel target for RNase III, which is EsxA toxin.

Although there is a huge amount of data and the interplay between RsaI and RsaE interesting, the message is diluted and the manuscript sometimes looks like a collection of small stories without an ending point or a phenotype. Some of the data presented are not all the time convincing and therefore, the manuscript lacks of a clear characterization and understanding of the mechanisms described.

1) General comment:

Because of the use of two genetic backgrounds, it is sometimes difficult to follow the study and the interpretation. These two strains are substantially different as they do not belong to the same ST.

Response: we deliberately used two distinct strains with different evolutionary history as we reasoned that discovered interactions in common between them would more likely be genuine, or perhaps play fundamental roles in core stress responses. We mention this on page 5 of the revised manuscript and discuss it further on page 20. We have, however, performed all the validation experiments in the USA300 background.

2) Line 187: The authors refer to supplementary tables 1 and 2 in the text, but it is likely 3 and 4 instead.

Response: Our apologies for the mistake. We have updated this in the text.

3) Line 292: The study on SprA1/SprA1_{AS} did not prove, to my knowledge, that RNase III is involved in their degradation. The sentence must be modified accordingly.

Response: Indeed, RNase III had not been found to degrade SprA1-SprA1_{AS} duplexes, but instead SprA has been found to be a target of RNase III in RIPseq experiments (Lioliou *et al*, 2012). We have updated the manuscript accordingly (page 9).

4) Figure 3c; bottom. It is not written on each heatmap, whether data were obtained from USA300 or JKD6009. This would help the reader.

Response: We thank the reviewer for pointing this out. We have updated the figure accordingly.

5) Figure 4: This figure and the text is relatively difficult to follow. In panel A the cells are grown for 3h while in panel D RNAs were extracted at an OD of around 3. On figure 4d, we barely see the two lengths of RsaE and RsaEp which is mentioned in the legend does not appear anywhere. On line 397/398, the lanes described in the text do not correspond to the ones in the figure 4d. Some experiments are done in serum and other in TSB which increase complexity to follow. For instance, in the conclusion of the section (from line 404 to 409), it seems that the authors describe data from human serum while statement on line 407 refer to TSB condition.

Response: We apologise for causing confusion. Note that the data that was shown in the original Figure 4 has now been split into Figure 4 and 5. With regards to the difference in growth durations, for panels D-F, we grew *S. aureus* to OD₆₀₀ 3 to match the conditions of the CLASH experiments. For panels a-c, we grew *S. aureus* in TSB and human serum for three hours. This is because *S. aureus* is unable to grow to such high optical densities in serum. Our growth of three hours in human serum was chosen to fit in with previous research from our collaborator Ronan Carroll, who was the first to observe changes in RsaE expression in human serum (Carroll *et al*, 2016, mBio). We wanted to replicate these conditions to see if we could reproduce these results. With respect to the RsaI over-expression experiment in Figure 4d (now Figure 5b), the RsaE shown in the Northern blot figure is the processed RsaE. We have now corrected this in the Figure.

In the conclusion paragraph we indeed refer to both serum and TSB data and this is indeed confusing. What the data show is that when RsaI levels are higher relative to RsaE (either by growing cells in human serum or by over-expressing RsaI in TSB), RsaI can sponge RsaE. We have changed the text on page 12 to make this clearer.

6) Why the authors did not test the overexpression of RsaI in human serum instead of TSB in figure 4f (or at least in the two media)?

There are two reasons why we only overexpressed RsaI in TSB medium. With this experiment, we wanted to test if the reason why deletion of RsaI had no impact on RsaE activity in TSB medium was simply because the level of RsaI relative to RsaE was so low that deleting RsaI would not have any impact on RsaE. Therefore, by over-expressing RsaI we wanted to know if we could now see a regulatory impact of RsaI in the rich TSB medium. Secondly, since RsaI over-expression in TSB already showed reduced activity of RsaE, it is reasonable to assume that we would obtain the exact same result if we over-expressed RsaI in human serum, especially since RsaE levels are much lower in human serum.

7) To distinguish the two forms of RsaE, qPCR using specific primers could be employed. A mapping of the extremities of the two forms would be relevant especially as the authors mentioned a length of around 75 nts in the discussion.

Response: Instead of using RT-qPCR, we have established Nanopore sequencing in our lab to determine the exact start and ends of these sRNA isoforms, as described above. This was not a trivial experiment but we found that Nanopore is more reliable at identifying the extremities of RNA compared to RNA-seq. These results indicate that RsaE and RsaI undergo trimming of the U-rich stretch in the intrinsic terminators of these sRNAs. The Nanopore data show that we previously over-estimated the degree of processing. These data are now shown in the revised Figure 4 and discussed on page 11 of the revised manuscript.

8) Then, does the modification of the RsaE targets level has an impact of their encoded proteins?

Response: Yes, RsaE has been shown to impact the levels of the proteins encoded by its mRNA targets (Geissmann et al NAR 2009) and in *B. subtilis* (Durand et al PLOS Genetics 2015).

9) Extended data fig.10 c: The ESMA presented is not really convincing.

Response: We respectfully disagree. These analyses in our view convincingly show that the C-rich motifs specifically base-pair with the G-rich motifs of RsaI and that mutations that disrupt these interactions can be restored using compensatory mutations.

9) Figure 5d: The retardation is not strong. There is no mention to the amount or concentrations added which renders difficult to have an overall idea of what could be the Kd (even though I agree that we cannot calculate it here based on the low affinity *in vitro*). What does the stars mean?

Response: We agree that duplex formation between RsaE and *psmA3* *in vitro* is not efficient. We have addressed this in the Discussion and mention that a chaperone may be required for efficient binding *in vivo*. However, our EMSA still shows that this interaction is specific as mutation of RsaE's CCC motif to GGG completely abrogates the interaction. The stars in this figure indicate non-specific bands, likely a dimer of RsaE (based on the fact that they are observed when *psmA3* is absent and only RsaE RNA is present; lanes marked "-"; see figure legend). We have updated the figure with the relative molar amounts of *psmA3*.

Although the *in vitro* binding is not efficient, we do not solely rely on this experiment for proof of RsaE-mediated regulation of the PSMs. Our mass spectrometry data shows that deletion of RsaE results in decreased abundance of other PSM proteins, and our haemolysis assays with culture supernatants and butanol extractions (which enriches for PSMs) show that deletion of RsaE decreases haemolytic activity.

10) Regarding the regulation of toxin production by RsaE: The GFP-reporter assay suffers from the lack of sufficient information. It is not clear why the construct is cytotoxic. How many codons that belong to PSM were cloned in the plasmid? Restricting to a few codons (perhaps only the initiation codon) might help. The interaction is apparently at the RBS, therefore doing the fusion at the AUG should not be a problem.

Response: The PSM constructs that we used had parts of the coding sequence fused to GFP. We initially had issues with transforming these constructs to RN4220 but our collaborator, Jai Tree, has managed to get this to work. This showed that the constructs were expressed at low levels. We had considered generating more constructs that included less coding sequence, however, we decided not to pursue this further for the following reasons. Firstly, during the revision process were unable to verify the predicted interactions between RsaE and *psmA1* and *psmA4* using EMSA (now mentioned on page 14) and the EMSAs with RsaE and *psmA2* did not generate convincing results, which is why we did not describe these data in the manuscript. The interaction with *psmA3* is also very weak, and we suspect that the interaction is chaperoned by proteins. Therefore, we were very cautious with interpreting these results. Secondly, the culture supernatant MS data show that RsaE does not impact the levels of PSM α 3 toxins but specifically of PSM α 1 and 4. Therefore, we believe that the way RsaE regulates the operon is more complex way than simply inhibiting the formation of translation initiation complexes. Our current model is that RsaE base-pairing with the *psmA3* SD sequence results in cleavage of the operon transcript (see pages 17 and 18 of the discussion). We now have sequencing data that support this idea (which we would be happy to share with this reviewer if needed), but we still have not completely figured out the precise mechanism. We are currently making several *psmA* mutants to test out theories, but it will take several

years to complete these analyses. Therefore, we would prefer to publish this in a separate article.

11) Did the authors tried toeprint assay or in vitro translation assay?

Response: We do not yet have the expertise for these type of experiments in our lab, so we have not tried this. But this is on our to-do list.

12) It would be of interest to verify whether the modulation of toxin production as an impact on virulence. What about PSM production in *rnc* mutant?

Response: We completely agree that understanding how this impacts virulence is an interesting question and this is something that we are focusing on now. However, we provide strong evidence that this is the case: we have shown that deletion of RNase III almost completely abrogates haemolytic activity, as shown in our culture supernatant haemolysis assays (Figure 6e). This observation is also observed after butanol extraction of the supernatants, which enriches for PSMs (Figure 6f), and thus we hypothesise that RNase III is involved in PSM production. However, further analysis of this is beyond of scope of the current manuscript.

13) The stabilizing effect of RNAIII onto *esxA* mRNA is interesting as the GFP-reporter experiment is convincing (even though EMSA does not show a strong affinity in vitro). However, one would have appreciated to have more physiological data to end the story.

Response: We agree that having more physiological data would be very interesting and we are in the process of establishing microphage infection assays to test this hypothesis. However, such experiments are complex and require optimisation. As such, we will be following this up in subsequent, future publications. Secondly, performing such experiments with an RNAIII deletion mutant will not be informative as it regulates many different toxins. Hence, we would have to make a USA300 strain that has silent mutations in the *esxA* coding sequence. This is something that we are trying now.

14) Also, it raises the question of the potential role of RNase III. This interaction was found by CLASH using RNase III as a bait, which implies degradation of the target. Here, we have a stabilization by RNAIII. What is the *esxA* RNA level in a *rnc* mutant strain?

Response: Exactly, this is a very interesting finding, and it is not immediately obvious how this would work mechanistically. We are keen to figure out the mechanism. However, this part of the project has been plagued by the fact that we were not able to form stable base-pairing interactions *in vitro*. We have tried to do RNA-structure probing on the RNAIII-*esxA* interaction, but the complex formation was too inefficient to detect any changes. However, we feel that the *in vivo* data strongly supports a direct interaction between RNAIII and *esxA*.

To address the question of *esxA* mRNA levels in the RNase III mutant, we have now performed Northern blots and RT-qPCR (see the new Figure 7g and Fig 7h). This shows that deleting RNase III has no significant impact on *esxA* mRNA levels. This is now mentioned on page 15 of the revised manuscript. Although binding by RNase III implies that the RNAIII-*esxA* duplex is likely degraded by this enzyme, we do not believe that this necessarily has to be the case. Like with RsaI and RsaE, we suspect that this nuclease is more likely functioning as an RNA chaperone. But clearly RNase III does not seem to be play a key role in regulating *EsxA* production.

15) Line 181: Change 'reegions' by 'regions'.

Response: we have updated the manuscript accordingly.

16) Line 654: needs rewriting.

Response: we have updated the manuscript accordingly.

Reviewer #3 (Remarks to the Author):

Review for “RNase III CLASH in MRSA uncovers sRNA regulatory networks 1 coupling metabolism to toxin expression”, McKellar et al.

Summary

McKellar et al., describe the role and function of sRNAs in mediating MRSA virulence. Using CLASH, they could identify a number of known, as well as hundreds of novel sRNA-RNA interactions when mimicking the host environment. The authors suggest that the production of small membrane-permeabilizing toxin, whose expression is linked to metabolism, is strongly regulated by sRNA. Among the identified sRNA-RNA interactions, they have uncovered two sRNAs, namely RNAIII and RsaE, which regulate the expression of at least 4 cytolytic toxins, which play a role in MRSA-mediated virulence. Overall, a very interesting study with several novel and potentially clinically relevant findings backed up by sound validations. CLASH data generation, adaptation of the original protocols and data analysis is well done.

General

remarks:

1) The biology of sRNAs in Gram-positive species is not well described (regulation of sRNA-target interaction might differ mechanistically in gram- and gram+). Given the broad readership of Nature Communications, a short introduction into this might be useful.

Response: We have now reworked the Introduction section to include details on sRNA biogenesis and functionality, using RNAIII and the Rsa family to illustrate. We also describe the role of RNA-binding proteins in Gram-positives and compare this to what is known for Gram-negatives.

2) Along the same lines, a general scheme of sRNA biogenesis/functioning in bacteria should be shown, which also includes RNase III, the bait protein chosen for CLASH analysis.

Response: We have now included a new Supplementary Figure 1 showing how sRNAs function and are processed in bacteria.

3) The authors could elaborate in more detail on the clinical potential of targeting sRNAs rather than using broad statements such as “our data has scope for further impact” (last sentence, discussion).

Response: We apologise for the lack of clarity in our writing. Here, we are discussing the fact that our CLASH data likely contains many novel regulatory RNAs ‘hidden’ inside mRNAs and intergenic regions that are worth further characterising in future studies. The “scope for further impact” that we were referring to is not in reference to the clinic, but rather the regulatory RNA field. We have now changed the wording on page 21 to make this clearer.

Specific points:

3) The composition of RPMI and human blood is rather different. It has apparently been shown before (ref 19) that bacteria grown in RPMI have a similar transcriptional profile than in human blood. What means “similar” here? It is surprising that the transcriptional profiles are similar as the composition of blood and RPMI is rather different. Are the authors sure that RPMI is a good choice here especially as there are more physiological media available (DOI:<https://doi.org/10.1016/j.cell.2017.03.023> and others). This reviewer is not an expert in microbiology but the other study (Mediati et al.) uses liquid Mueller-Hinton. Might this be a more physiological medium?

Response: In Mäder *et al*, 2016, the authors grew *S. aureus* in many different conditions, including human plasma and RPMI medium and then examined gene expression using tiling

arrays. Using Principal Component Analysis, they found that cells grown in RPMI and human plasma show very similar gene expression patterns. Afterwards, they examined the expression of individual genes involved in amino acid transport, iron metabolism and virulence factors. From this, they found that these analyses “clearly showed that the eukaryotic cell culture medium RPMI resembles the conditions *S. aureus* is faced with in human plasma”. This is primarily thought to be due to the low concentration of free iron in both media, which is an essential cofactor for *S. aureus* and its absence induces virulent behaviour.

However, we accept and agree with the Reviewer’s fundamental point that RPMI medium cannot perfectly recapitulate the conditions *S. aureus* is faced with in the blood. However, each CLASH library requires a large quantity of cells (~0.5g); performing our experiments in replicates and in two different strains would therefore require impractical quantities of human blood. As such, we used the RPMI medium to mimic this condition as best as possible, and then performed validation experiments in human serum in more manageable volumes.

With regards to Mueller-Hinton medium, this is a nutrient-rich broth and so would not mimic the blood condition. We have used tryptic soy broth (TSB) in our studies as a rich medium reference.

4) The resolution of Fig. 2D should be improved as it is hardly readable.

Response: This was caused by a formatting error. This has now been corrected.

5) Can the authors explain why mRNA-mRNA interactions are predominant in the CLASH data?

Response: we are not entirely sure why this is the case. However, we have further expanded our Discussion of this observation on pages 20 and 21 of the revised manuscript. Here, we have mentioned these may be involved in the stress response and may be biologically meaningful, drawing parallels to Mediati *et al*'s accompanying manuscript and observations in *Listeria monocytogenes*.

6) Independent of RNase III activity, different species of RsaI and RsaE appear in the different media with different transcript lengths. Can the authors speculate how the medium composition might influence transcript length? If so, a highly physiological medium mimicking the composition of human blood as closely as possible is even more important, especially for validation experiments.

Response: We have now mapped the extremities of both RsaI and RsaE using nanopore sequencing. This reviewed that the only difference between the processed and unprocessed forms is the length of the U-stretch in the transcription terminator sequence (see revised Figure 4). Our current model is that RsaE is processed by an exonuclease in TSB. We are unsure of the exact mechanism of this, and future studies will focus on elucidating this. To our knowledge, three 3'-to-5' exonucleases exist in *S. aureus* (PNPase, RNase R, and YhaM) and any of these could be involved. We now discuss this in more detail on page 17. We agree that having a physiological medium that more closely resembles human blood would be great to have. However, it is not immediately clear what medium (other than RPMI) would be the best. We are now trying to improve the CLASH protocol so that we can do the experiments with fewer cells. These developments may allow us to perform CLASH experiments under actual infection conditions, which would be the ideal scenario. By improving the library preparation protocol, we hope to be able to accomplish this soon.

7) In cancer cells, the length of 3' UTR regions of tumor suppressor genes can be shortened so that less miRNAs bind and down-regulate their target. Could a similar mechanism be active in bacteria, i.e. *S. aureus*?

Response: This is an interesting parallel to draw, and we are unsure of the answer. The current paradigm in *S. aureus* sRNA functionality is that a single sRNA molecule binds to a single mRNA molecule, most commonly at the 5' UTR or RBS. Only a few examples of alternative binding exist, such as RNAIII binding *coa* in the coding sequence. Identification and validation of sRNAs binding the 3' UTR of their targets in *S. aureus* will be needed to fully answer this point.

We would like to thank the reviewers for their constructive comments and very helpful suggestions.

Reviewer #1

In their manuscript 'RNase III CLASH in MRSA uncovers sRNA regulatory networks coupling metabolism to toxin expression' McKellar et al. adapt and optimize the CLASH approach to Gram-positive bacteria and apply it to *S. aureus* to map RNA-RNA interactions involving RNase III. The authors employ established culture media to mimick various *S. aureus* infection conditions, and the transcription patterns identified (by classical RNA-seq prior to CLASH) revealed a number of typical transcription hallmarks, generally confirming the suitability of these media as model systems. RNase III-CLASH (performed with two clonally unrelated MRSA strains) revealed known RNA interaction patterns which highlighted the robustness of the approach. Furthermore, numerous novel condition- and strain-dependent RNA-RNA interactions were identified from which a few were selected for detailed analysis. Here the authors mainly focused on sRNA-RNA interactions involving sRNAs RsaI, RsaE and RNAIII. Based on a limited number of validation experiments the authors propose (i) RsaI to represent a sponge for RsaE that inactivates the sRNA in concert with RNase III. Moreover, they suggest (ii) RNAIII to positively regulate the EsxA toxin through specific base-pairing and (iii) RsaE to bind to PSM transcripts and enhance translation of the cytolysins.

This is a comprehensive study of general interest revealing a wealth of both known and novel RNA-RNA interactions in *S. aureus*. The manuscript is carefully written and presented, and the study highlights the eminent role of sRNAs in both metabolic and virulence adaptation of this important pathogen. The observed growth- and strain-specific differences in sRNA expression and functions are remarkable and of great interest, although the authors do not address and discuss this aspect in the manuscript.

The first part of the study (covering the CLASH approach) is straightforward and of high quality. The data, which appear to be robust and reproducible, are supported by several bioinformatic analyses, including folding energies of RNase III-bound RNA duplexes and correlation analyses. However, the functional studies on selected sRNA interactions (presented in the second part of the paper) require attention and additional experimentation.

Major comments:

1) RsaI sponge part

The idea of RsaI as a sponge for RsaE is intriguing. However, the data, as they stand now, are not entirely convincing and support only in part the conclusions drawn. The authors are actually aware of these issues and stay therefore a bit vague in their interpretations. By performing a few more control experiments and by paying attention to the molecular details of the putative RNA-RNA interactions, the authors should actually be able to address these concerns and shed more light on the underlying mechanism.

In this respect, the crucial point of the study is the medium-dependent processing of RsaI and RsaE which the authors identified and which is an important finding. Unfortunately, the authors neglect in subsequent experiments the presence of these various RNA species which are (at least for RsaE/RoxS) known to display very different target RNA spectra. Although processing of RsaI and RsaE itself (which RNase(s) involved; conditions etc.) is not subject of the study, elucidating the interactions of the full-length and processed species with their respective binding partners is key for understanding the general regulatory mechanism.

Also, the fate of the RsaE mRNA targets (i.e. transcript stabilization or degradation) upon binding of either full-length or processed RsaE needs to be established in the first place before

any conclusions on their up or down regulation can be drawn in the different *rsaI* or *rnc* deletion mutants and media.

For doing these experiments, the authors should streamline their experimental plan and focus on one strain (e.g. USA300) to generate a set of isogenic mutants, keeping in mind that other strains (as shown for JKD6009) may behave differently regarding their RNA-RNA interaction profiles. The latter is an important point for appreciating the huge regulatory potential of *S. aureus* (as a species) and the authors should address this fact explicitly in the discussion.

Below a detailed list of questions and suggestions that may help to improve the manuscript:

1. Demonstrating specific binding of RsaE to RsaI and cleavage of the complex *in vitro* by RNase III is convincing (ExData Figs.10, 11). Of note, in these *in vitro* experiments full-length sRNA partners are employed. However, both RsaI and RsaE undergo processing which is obviously strongly condition-dependent (and which does not involve RNase III as the processing enzyme) (Figure 4a). Under the conditions tested (TSB, human serum) either full-length RsaI meets processed RsaE (TSB) or vice versa, processed RsaI will find full-length RsaE as partner (serum). So, what about the binding interaction between processed and full-length sRNA partners? Is this comparable to the full-length molecules and are the respective hybrids equal RNase III targets?

Response: To further examine the processing of RsaE and RsaI and to identify the RNA species present in TSB and human serum, we have now performed Nanopore sequencing on total RNA extracted from cells grown in these conditions. The manuscript has been updated with these data (Fig. 4), which are now discussed on page 11 of the revised manuscript. This revealed that the polyU tracts at the 3'-end of these sRNAs, which are part of the intrinsic transcription terminators, were exonucleolytically trimmed in response to the nutritional environment. This was particularly prominent for RsaE in TSB. These trimming events result in heterogeneous subpopulations of RsaE and RsaI, varying from the full-length transcript to trimmed subspecies up to 7 nucleotides shorter. This was to our surprise as we had previously hypothesised that RsaE and RsaI processing in TSB and serum would be at the 5'-end to free the sRNA seed sequence, as reported in *S. epidermidis* RsaE and *B. subtilis* RoxS. We evidently overestimated the degree of processing.

To the best of our knowledge, 3'-end trimming of sRNAs has not been reported previously in Gram-positive bacteria and we have discussed potential regulatory outcomes for this, and potential RNases involved in the Discussion (page 17). As these RNA subspecies only vary in the length of their U-rich sequences in the intrinsic terminators, which have never been described to be involved in any base-pairing interactions, we do not envisage that this has any effect on the ability of the sRNA seed sequence to base-pair with other molecules, including the interaction between RsaI and RsaE. In fact, all our EMSAs and cleavage assays were performed with an RsaE transcript lacking the terminator sequence, demonstrating that it is not essential for binding RsaI *in vitro*. As such, we believe that cleavage assays on RsaE/RsaI subspecies will not provide any new insights. We do believe that this 3'-end trimming might impact the stability of RsaE and that the shorter versions are more stable. This is also mentioned on page 17 of the revised manuscript in the Discussion section.

2. From the data shown it is difficult to entirely comprehend the function of RsaI as a sponge of RsaE (Figure 4). The authors state that RsaI and RsaE form hybrids that are targets of RNase III. Thus, if RsaI sponges RsaE, why do RsaE levels then remain unaffected in the *rsaI* and *rnc* deletion mutants (Figure 4a)? Actually, RsaE levels are expected to go up in absence of the sponge and/or the processing enzyme? In this respect, what would happen in a *rsaI/rnc* double mutant; and what would be the effect of *rsaE* deletion or RsaE overexpression on RsaI levels?

Response: We agree that our observation that the RsaI-RsaE duplex can be cleaved by RNase III *in vitro* would imply that the duplex is, by default, degraded. As such, one would then expect that RsaE levels would increase upon RsaI or RNase III deletion. We clearly do not see this. This has puzzled us for some time and there is no simple answer to this. We favour a model where RNase III here mostly acts as an RNA chaperone. It is possible that *in vivo* only a small fraction of the RsaI-RsaE duplex is cleaved by RNase III. We showed that the duplex can be specifically cleaved by RNase III *in vitro*, however, the efficiency appears to be low. It has been proposed that RNase III can function as a non-catalytic RNA-binding protein (Calin-Jageman and Nicholson, NAR 2003). Thus, we speculate that the RsaE-RsaI duplex is largely resistant to RNase III degradation, but it is still efficiently bound by the protein. Therefore, we hypothesise that main function of RNase III here is to stabilise the RsaI-RsaE interaction *in vivo* or by preventing RsaE from base-pairing with other targets. We now discuss this on page 16 of the revised manuscript.

With respect to the proposed sponging activity, several groups have shown that sponging of an sRNA by another does not necessarily have to result in degradation of the target sRNA. For example, SprY binds to RNAIII and inactivates it but this interaction does not alter the stability of RNAIII (Le Huyen et al NAR 2021; see Figueroa-Bossi and Bossi, 2018 and Denham 2020 for further support of this definition). Therefore, we believe that RsaI (Figs 4 and 5) in that sense functions similarly as SprY.

We have also performed the experiment suggested by the reviewer to examine RsaE levels in a Δ rsaI- Δ rnc double mutant and included the data in the revised manuscript (Supplementary Figure 13a). These deletions had no noticeable effect on RsaE stability in TSB or human serum, again contributing to the idea that RsaI primarily regulates RsaE activity, not stability.

Even though RsaI does not induce RsaE degradation, deletion of RsaI led to downregulation of RsaE targets in comparison to WT when cells were grown in human serum (Fig. 5a). This indicates that RsaE has increased activity in the absence of RsaI. Collectively, these data justify the idea of RsaI acting as a sponge of RsaE. We have reworked the description of these results in the Discussion (pages 16 and 17).

To investigate if RsaE can regulate the activity of RsaI, as suggested by the reviewer, we analysed RsaI levels in strains lacking RsaE and in strains in which we overexpressed RsaE under the control of pAmiA promoter of pICS3 plasmid. We have used a RsaE sequence which includes the both polyU terminators found in the RsaE gene. Upon successful transformation of wild type and RNase III mutant USA300 cells, we observed reduced growth of the cells. This is consistent with the previously observed cytotoxic effect of RsaE overexpression (Bohn et al., Nucleic Acids Research 2010). Despite this, we were able to extract total RNA from cells grown in TSB and human serum with and without the plasmid for RsaE overexpression. Northern blot analysis showed only a modest increase in RsaE (around 1.5- to 2-fold) when over-expressed from the pICS3 plasmid (Supplementary Figure 13c, lanes 3 and 4). No significant changes in the levels of RsaI was observed when RsaE was expressed from this plasmid. The same was observed in the Δ rsaE strain (Supplementary Figure 13c, lanes 5 and 6). Overexpressing RsaE in the Δ rsaE background revealed that this plasmid was unable to induce high levels of RsaE expression (Supplementary Figure 13, lanes 7 and 8 and quantification of the results). It is possible that with this plasmid we were unable to express RsaE at high enough levels to impact RsaI levels. However, it is important to point out that over-expression of RsaE from the same plasmid significantly (1.5-2-fold) increased the haemolytic activity of USA300 (Figure 6e). Thus a ~2-fold increase of RsaE level is sufficient to alter toxin production.

To get around the toxicity issue and to see if we could express RsaE at higher levels, we used a previously described tetracycline-inducible RsaE pRMC2 construct (kindly provided by

Philippe Bouloc; Rochat et al NAR 2018). We induced RsaE expression for 15 minutes in USA300, resulting in high over-expression of RsaE. Despite this, we could not detect strong changes in the levels for the RsaE targets (see new Supplementary Fig. 13e). Therefore, it appears that we are unable to express RsaE in USA300 at the levels required to significantly impact its target levels. However, we should note that RsaE expressed from this plasmid runs higher than endogenous RsaE.

Although the RsaE over-expression did not give satisfactory results, what is interesting from these data is that we found that RsaE is strongly regulated in human serum, even when overexpressed from plasmids using two different promoters. This suggests that RsaE levels are post-transcriptionally regulated in human serum.

3. In 383: The statement that RsaE levels in serum are lower than in TSB is hard to judge as two different RsaE species (full-length in TSB, processed in serum) are compared (Figure 4a,b). The authors rightly cite previous work showing that such RsaE isoforms are functionally different and may interact with different targets, making a direct comparison difficult, if not inappropriate. Accordingly, the statements in lines 383 and 386 regarding relative RsaE and RsaI levels in the different media are daring. As the authors consider the relative expression ratios as crucial for the RsaI sponge function this issue needs to be addressed.

Response: Following up from point 1, we now know that these various isoforms of RsaE are truncates of the 3'-end. While these species of RsaE differ in their length it is unlikely that to influence their seed sequence motif and therefore to affect their functionality. As such, we have quantified all these species as a collective of 'RsaE' or 'RsaI'. We also repeated the experiments and ran the RNA samples on lower percentage polyacrylamide gels to make the bands more compact and quantified the results (Supplementary Figure 13a and c). These results again show a strong and significant reduction in RsaE levels in human serum.

4. In Figure 4c, the authors show relative expression levels of RsaE targets in various media (in comparison to wildtype). But, what about their absolute transcription? Particularly for the TCA cycle genes, differences in transcript levels between rich medium (TSB) and human serum can be expected. In this respect, if basal transcripton is generally low, relative changes are difficult to interpret.

Response: We have analysed the mRNA levels of all tested RsaE targets in the WT strain during growth in TSB and human serum (see new Supplementary Figure 13b). This revealed that there are no significant differences in the expression of these mRNAs in the two growth conditions. We have addressed this on page 11 and 12 of the revised manuscript. Thus, the changes we observe in the Δ rsaI and Δ rnc mutants are a result of these deletions, not because the mRNA targets are expressed at very different levels.

5. Further to Figure 4c: In rsaI and rnc mutant backgrounds, the authors find RsaE targets almost unaffected or slightly increased when bacteria were grown in TSB, while in human serum RsaE target levels are significantly reduced. Apart from the undefined basal RsaE target transcription levels (#4), do the tested RsaE targets actually interact with processed or full-length RsaE or both? In conjunction with the control experiment on RsaI/RsaE isoform interactions suggested above (#1), this information is crucial to appreciate the alleged RsaI sponge function under the various conditions and the subsequent effects on the RsaE targets.

Response: As described above, we no longer propose that RsaE is processed at the 5'-end to regulate its base-pairing potential with its target mRNAs. Since the difference between the processed and unprocessed RsaE is only a few nucleotides and does not involve seed sequences, we feel it is safe to assume that both the processed and unprocessed RsaE can interact with the same mRNA substrates. Having now performed many Northern blots on RNA from TSB and human serum, we reproducibly see that RsaE is much more abundant in TSB

compared to serum. We do believe that the shorter RsaE form is able to interact with the RsaI. Our EMSAs were performed with an RsaE transcript lacking the terminator sequences. Secondly, overexpression of RsaI in TSB (where the shorter RsaE form is most abundant) significantly increased the levels of RsaE targets (Figure 5b and c). Therefore, we favour the model that RsaE is expressed at such high levels in TSB that RsaI is unable to make a significant impact on RsaE mRNA targets in this medium.

6. What happens to the RsaE targets if RsaE is overexpressed in the *rsaI* and *rnc* deletion mutants?

RsaE over-expression in TSB should reduce RsaE target levels irrespective of RsaI because RsaI does not impact RsaE activity in TSB unless it is overexpressed. In serum the levels should go further down unless this is unfavourable for the cell. To study the role of RsaI in sponging RsaE, we have measured the expression changes of RsaE targets in *rsaI*- and RNase III deletion strains in TSB and human serum. The RsaE targets were selected based on previously published papers and our hypothesis is that in absence of RsaI or RNase III there will be no sponging of RsaE, and RsaE will be free to downregulate the levels of its targets. Indeed, we observed strong and significant reduction of RsaE targets in human serum but not in TSB. Considering the issues with overexpressing RsaE and that we already observe a strong and specific effect upon deletion of RsaI and RNase III in the regulation of RsaE targets, we do not think the proposed experiments will be informative.

7. Are the RsaE/mRNA target complexes subject to RNase III cleavage too? They do not show up in the sRNA-mRNA CLASH data list, but *citC*, *sucC*, *fumC* and *gcvPA* have been previously identified as RNase III targets (Lioliou *et al.* 2012). Does this mean the transcripts are protected by RsaE from RNase III-mediated cleavage/degradation? Again, adding the isogenic USA300 *rsaE* deletion mutant as control would be very informative (see also #2) to gain insight into the general fate/turnover of the RsaE targets upon binding of the sRNA; the more so as this mutant is already available to the authors.

Response: It is difficult to compare the study carried out by Lioliou *et al.*, 2012, and our own. Lioliou *et al.* used the RN6390 strain and performed their RIP experiments at low optical densities in BHI (rich) medium. They induced RNase III expression at OD₆₀₀ 0.2-0.3 for 90 minutes. This is likely to give a final experimental OD of 0.8-1.2, assuming a 30-minute doubling time. In comparison, we performed our CLASH experiments in the USA300 and JKD6009 backgrounds at OD₆₀₀ 3 in TSB and RPMI medium.

It is known that deletion of RsaE leads to upregulation in its mRNA targets (Geissmann *et al.*, 2009) and that overexpression of RsaE downregulates them (Bohn *et al.*, 2018). Thus, it can be concluded that RsaE binding to its target mRNAs does not lead to RNase III protection, as suggested by the reviewer, but possibly their cleavage. However, these complexes may be degraded too quickly to be captured by RNase III CLASH or may occur during growth stages different to those used in our experiments. This would explain why we did not identify them in our CLASH data.

Performing *in vitro* RNase III cleavage assays between RsaE and its mRNA targets is a significant endeavour and beyond the scope of this manuscript as we are mainly focusing on the RsaI sponging activity. Additionally, examination of the expression of RsaE targets upon RsaE deletion has already been published in previous work (Geissmann *et al.*, 2009), as described above.

8. In the *in vitro* binding experiments (ExData Figs.10, 11), the authors used Ca²⁺ to successfully inhibit RNase III enzyme activity. However, the Ca²⁺ concentration used is not indicated. In human serum, normal Ca²⁺ concentration is in the range of 1-1.3 mmol/l. Would this be sufficient to influence intracellular Ca²⁺ concentrations in the bacteria and inhibit

RNase III? In other words, is the enzyme generally inactive upon growth of *S. aureus* in human serum (but still able to bind targets)?

Response: We used 10 mM CaCl₂ for our cleavage experiments and therefore we believe that RNase III should still have some activity in serum where the concentration around 2-3 mM. We have now added details on our calcium concentrations in the Supplementary Methods (page 2) and discussed this point in the Discussion on page 16.

Toxin expression part

9. Figure 5e: Please show haemolytic activity of the WT strain.

Response: This is the first bar of the now Fig 6e and f. We have changed the labelling in Fig. 6e to make it easier to interpret.

10. Figure 5e: Why did the authors use an RNase III knockout in yet another (third) strain background? Please, demonstrate the effect in the USA300 background for better comparability.

Response: The AH1263 strain a derivative of USA300 that has been cured of a natural plasmid This was an RNase III transposon mutant that was available from a mutant library. However, we have now repeated the experiment using a clean RNase III deletion mutant made in the USA300 LAC background, which we have used for all our validation experiments. The new haemolysis assay results are included in the revised Figure 6e. We obtained identical results with the new USA300 LAC RNase III deletion strain, supporting our original conclusions.

11. What about the other known phenotypes associated with PSMs such as phagosomal escape and detachment from biofilms? Are they also affected in the RsaE deletion strain?

Response: This is an excellent question, and we are addressing this now. However, because RsaE regulates so many targets, performing macrophage/neutrophil infection or biofilm detachment assays with an RsaE deletion mutant is not going to be very informative as we cannot exclude the possibility that any phenotypical changes are directly linked to changes in α PSM levels. For example, in *S. epidermis*, RsaE influences biofilm formation at multiple levels by enhancing the production of PNAGs (by repressing IcaR) and eDNA release (Schoenfelder et al PLOS path. 2019). Hence, we are now making several α PSM operon mutants that can no longer be regulated by RsaE (by mutating part of the base-pairing interaction sites) or in which the AUGs of *psm α 1* and *psm α 4* are mutated so that we can replicate the effect of what we observed in the RsaE deletion mutant on α PSM production, but without having to delete RsaE. However, this is an entirely new project on its own and with the infection and biofilm analyses would take several years to complete.

12. L413-414: The authors state that the RsaE interactions with a/bpsm transcripts were observed in JKD6009, but the in vivo follow-up experiments (Figure 5e-f) were done in USA300. Please clarify.

Response: We performed the validations in USA300 as this strain is known to produce very high levels of PSMs (Berlon *et al*, 2015, Journal of Infection). Thus, we anticipated that performing the validation studies in this strain would be easier as the culture supernatants would have much stronger haemolysis activity. The interactions between RsaE and the PSMs were identified in the JKD6009 strain through CLASH. However, we also identified interactions between sRNAs and PSMs in USA300, such as between RNAIII with *psm α 4*.

13. Figure 5d: Given the predictions in 5c and the detected changes for PSMa1 and PSMa4,

but not for PSMa3 by Mass-spec analysis (Ext data Fig. 13), why was only the interaction for *psma3* tested?

Response: This is because the *psma3* interaction was detected in the CLASH data and because it is the most cytolytic toxin. During the revision period we also performed EMSAs with RsaE and fragments of *psma1* and *psma4* but we were unable to detect any significant complex formation. We now state this on pages 13 and 14 of the revised manuscript. In some experiments we did manage to see a very weak interaction between *psma2* and RsaE, but this was not reproducible and therefore we are hesitant to report these findings.

14. Fig 6D: In this experiment, a gfp vector control without the *esxA* 5'-UTR fragment should be included.

We have now included new FACS data for two GFP controls (Supplementary Figure 16). Here, showed that expression of RNAIII does not impact the produced fluorescence of GFP or a *gyrB*-GFP fusion. We chose *gyrB* as this is not a known target of RNAIII, and thus these experiments demonstrate that the promotion of *esxA*-GFP fluorescence by RNAIII is specific. We now discuss these data on page 15 of the revised manuscript

Minor comments:

15. Please check RNA marker sizes. For example, the size of RsaE (which is 102 nt) appears to be too small compared to the 100-nt marker indicated on the gels (both in the manuscript and the source data).

Response: We have checked the gels again and updated the figure accordingly.

16. Ext data, pg 2 'RsaE targets': Here *S. aureus* JKD6008 is mentioned. Is this a typo or was indeed this VISA-derivative of JKD6009 used in the experiments?

Response: This was indeed a typo and we have edited the manuscript accordingly. Thank you for catching this error.

17. Ext. data Fig. 6. How do the authors explain the major differences in the interactome between the two strains used? (see also general comment above).

Response: We believe this is because both strains have different growth rates and because they reach stationary phase at different densities. We have measured the growth of all the strains we have used and found that USA300 grows to much higher optical densities than the JKD6009 strain. Therefore, we think that the differences in growth dynamics may explain the differences in the interactomes. We have now included this data in Supplementary Table 5 and discuss these findings on page 20 of the revised manuscript.

18. Further to this: Please, state briefly why these two strains were chosen (L95-96). For example, did the authors expect to find fundamentally different interactions due to known differences in sRNA regulation or because the strains are associated with different disease outcomes/virulence?

Response: We did not initially expect such large differences in the identified interactomes. However, we utilised two strains as we reasoned interactions identified in both would be more likely to be real. This guided our decision to investigate the RsaI-RsaE and RsaA-RNAIII interactions, which were identified in both strains. USA300 is also believed to be much more virulent. However, the Reviewer is correct to point out the remarkable differences between the found interactomes, and we have mentioned this in the Discussion (page 20; also see response to comment 17).

19. L179: Here the authors specifically mention difficulties with existing annotations but fail to actually list the annotation that was used for either strain in the manuscript. For example, for JKD6009, was the updated annotation from Mediati et al. (parallel submission) used?

Response: We did not use the annotation from Mediati *et al* as we had already finished our analyses of the JKD data and had started to focus more on USA300 as a model strain. To improve the USA300 annotation file we downloaded a GTF annotation file from ENSEMBL (https://bacteria.ensembl.org/Staphylococcus_aureus_subsp_aureus_usa300_fpr3757_gca_000013465/Info/Index/). We subsequently used Rockhopper and our RNAseq data to map potential UTRs. We now discuss this in more detail on page 7 of the revised manuscript as well as in the Supplementary Methods in the Supplementary Data File. We also uploaded this revised USA300 annotation to GEO.

20. Supplementary tables 1 and 2: These tables need a comprehensive legend describing what can be found in each column.

Response: we have now included detailed legends for the Supplementary Tables on pages 34 and 35 of the Supplementary Data.

21. L188: 'Many of the predicted sRNA-target interactions in USA300 had poorer folding energies compared to chimeras identified in JKD6009 ...' What is the potential reason for this discrepancy? This may feed back to the questions why the two strains were used in the first place.

Response: when inspecting the USA300 CLASH data, we found many hybrids with poor folding energies. Manual examination found these to be repetitive, AT-rich sequences that mapped to multiple annotated features in the USA300 genome, particularly around ribosomal and transfer RNA species. We suspect that these are not actually intermolecular hybrids but potentially the result of poor genome annotation.

To overcome this, we filtered our interactions for those with a minimum folding energy (MFE) of -10 kcal/mol and containing a *bona fide* sRNA. For reference, we found that our positive control interactions (those already validated by the field) always exhibited a minimum MFE of around -20 kcal/mol with a median of -30 kcal/mol (Fig 2a). This filtering resulted in our USA300 hybrids having similar MFE distributions as known interactions (Figure 2a). We detail these analyses and the rationale behind them on pages 7 and 8 of the revised manuscript.

22. L51: Please, replace 'spp.' with 'enterica' (there is plenty of research in different serovars but not in different species)

Response: we have updated this in the text.

23. L181: Please correct typo in 'regions'

Response: we have updated this in the text.

Reviewer #2 (Remarks to the Author):

In their manuscript, McKellar and collaborators studied the sRNA regulatory networks using CLASH to address an important question: what is the biological function of sRNAs in *Staphylococcus aureus*? Because Hfq is dispensable for sRNA-mRNA interactions in this bacterium, the authors adapted CLASH using RNase III as a bait, based on the assumption that this double-strand endoribonuclease is involved and plays a key role in sRNA-mediated regulation.

First, the authors described how they set up the method and perform CLASH under different in vitro conditions that mimic the environment encountered during infection. They used RPMI to mimic serum, and LPM and pH 7.6 or 5.4 to study nutritional adaptation and acidic stress. CLASH was performed in two different strains (USA300 and JKD6009) although the latter was used only in TSB and RPMI. Overall, there is a long description of the validation of CLASH to prove that the use of RNase III as a bait enable robust RNA-RNA interactions. Then, there are several short stories dealing with RNA-RNA interactions. RsaI is shown to regulate RsaE, thus having an impact on RsaE targets in human serum. RsaE is shown to regulate PSM production although the mechanism of action is not understood. The overall effect of RsaE onto PSM production does not follow the canonical mode of action as the authors described activation of expression instead of repression while the RBS seems targeted. Finally, the authors describe a novel target for RNase III, which is EsxA toxin.

Although there is a huge amount of data and the interplay between RsaI and RsaE interesting, the message is diluted and the manuscript sometimes looks like a collection of small stories without an ending point or a phenotype. Some of the data presented are not all the time convincing and therefore, the manuscript lacks of a clear characterization and understanding of the mechanisms described.

1) General comment:

Because of the use of two genetic backgrounds, it is sometimes difficult to follow the study and the interpretation. These two strains are substantially different as they do not belong to the same ST.

Response: we deliberately used two distinct strains with different evolutionary history as we reasoned that discovered interactions in common between them would more likely be genuine, or perhaps play fundamental roles in core stress responses. We mention this on page 5 of the revised manuscript and discuss it further on page 20. We have, however, performed all the validation experiments in the USA300 background.

2) Line 187: The authors refer to supplementary tables 1 and 2 in the text, but it is likely 3 and 4 instead.

Response: Our apologies for the mistake. We have updated this in the text.

3) Line 292: The study on SprA1/SprA1_{AS} did not prove, to my knowledge, that RNase III is involved in their degradation. The sentence must be modified accordingly.

Response: Indeed, RNase III had not been found to degrade SprA1-SprA1_{AS} duplexes, but instead SprA has been found to be a target of RNase III in RIPseq experiments (Lioliou *et al*, 2012). We have updated the manuscript accordingly (page 9).

4) Figure 3c; bottom. It is not written on each heatmap, whether data were obtained from USA300 or JKD6009. This would help the reader.

Response: We thank the reviewer for pointing this out. We have updated the figure accordingly.

5) Figure 4: This figure and the text is relatively difficult to follow. In panel A the cells are grown for 3h while in panel D RNAs were extracted at an OD of around 3. On figure 4d, we barely see the two lengths of RsaE and RsaEp which is mentioned in the legend does not appear anywhere. On line 397/398, the lanes described in the text do not correspond to the ones in the figure 4d. Some experiments are done in serum and other in TSB which increase complexity to follow. For instance, in the conclusion of the section (from line 404 to 409), it seems that the authors describe data from human serum while statement on line 407 refer to TSB condition.

Response: We apologise for causing confusion. Note that the data that was shown in the original Figure 4 has now been split into Figure 4 and 5. With regards to the difference in growth durations, for panels D-F, we grew *S. aureus* to OD₆₀₀ 3 to match the conditions of the CLASH experiments. For panels a-c, we grew *S. aureus* in TSB and human serum for three hours. This is because *S. aureus* is unable to grow to such high optical densities in serum. Our growth of three hours in human serum was chosen to fit in with previous research from our collaborator Ronan Carroll, who was the first to observe changes in RsaE expression in human serum (Carroll *et al*, 2016, mBio). We wanted to replicate these conditions to see if we could reproduce these results. With respect to the RsaI over-expression experiment in Figure 4d (now Figure 5b), the RsaE shown in the Northern blot figure is the processed RsaE. We have now corrected this in the Figure.

In the conclusion paragraph we indeed refer to both serum and TSB data and this is indeed confusing. What the data show is that when RsaI levels are higher relative to RsaE (either by growing cells in human serum or by over-expressing RsaI in TSB), RsaI can sponge RsaE. We have changed the text on page 12 to make this clearer.

6) Why the authors did not test the overexpression of RsaI in human serum instead of TSB in figure 4f (or at least in the two media)?

There are two reasons why we only overexpressed RsaI in TSB medium. With this experiment, we wanted to test if the reason why deletion of RsaI had no impact on RsaE activity in TSB medium was simply because the level of RsaI relative to RsaE was so low that deleting RsaI would not have any impact on RsaE. Therefore, by over-expressing RsaI we wanted to know if we could now see a regulatory impact of RsaI in the rich TSB medium. Secondly, since RsaI over-expression in TSB already showed reduced activity of RsaE, it is reasonable to assume that we would obtain the exact same result if we over-expressed RsaI in human serum, especially since RsaE levels are much lower in human serum.

7) To distinguish the two forms of RsaE, qPCR using specific primers could be employed. A mapping of the extremities of the two forms would be relevant especially as the authors mentioned a length of around 75 nts in the discussion.

Response: Instead of using RT-qPCR, we have established Nanopore sequencing in our lab to determine the exact start and ends of these sRNA isoforms, as described above. This was not a trivial experiment but we found that Nanopore is more reliable at identifying the extremities of RNA compared to RNA-seq. These results indicate that RsaE and RsaI undergo trimming of the U-rich stretch in the intrinsic terminators of these sRNAs. The Nanopore data show that we previously over-estimated the degree of processing. These data are now shown in the revised Figure 4 and discussed on page 11 of the revised manuscript.

8) Then, does the modification of the RsaE targets level has an impact of their encoded proteins?

Response: Yes, RsaE has been shown to impact the levels of the proteins encoded by its mRNA targets (Geissmann et al NAR 2009) and in *B. subtilis* (Durand et al PLOS Genetics 2015).

9) Extended data fig.10 c: The ESMA presented is not really convincing.

Response: We respectfully disagree. These analyses in our view convincingly show that the C-rich motifs specifically base-pair with the G-rich motifs of RsaI and that mutations that disrupt these interactions can be restored using compensatory mutations.

9) Figure 5d: The retardation is not strong. There is no mention to the amount or concentrations added which renders difficult to have an overall idea of what could be the Kd (even though I agree that we cannot calculate it here based on the low affinity *in vitro*). What does the stars mean?

Response: We agree that duplex formation between RsaE and *psmA3* *in vitro* is not efficient. We have addressed this in the Discussion and mention that a chaperone may be required for efficient binding *in vivo*. However, our EMSA still shows that this interaction is specific as mutation of RsaE's CCC motif to GGG completely abrogates the interaction. The stars in this figure indicate non-specific bands, likely a dimer of RsaE (based on the fact that they are observed when *psmA3* is absent and only RsaE RNA is present; lanes marked "-"; see figure legend). We have updated the figure with the relative molar amounts of *psmA3*.

Although the *in vitro* binding is not efficient, we do not solely rely on this experiment for proof of RsaE-mediated regulation of the PSMs. Our mass spectrometry data shows that deletion of RsaE results in decreased abundance of other PSM proteins, and our haemolysis assays with culture supernatants and butanol extractions (which enriches for PSMs) show that deletion of RsaE decreases haemolytic activity.

10) Regarding the regulation of toxin production by RsaE: The GFP-reporter assay suffers from the lack of sufficient information. It is not clear why the construct is cytotoxic. How many codons that belong to PSM were cloned in the plasmid? Restricting to a few codons (perhaps only the initiation codon) might help. The interaction is apparently at the RBS, therefore doing the fusion at the AUG should not be a problem.

Response: The PSM constructs that we used had parts of the coding sequence fused to GFP. We initially had issues with transforming these constructs to RN4220 but our collaborator, Jai Tree, has managed to get this to work. This showed that the constructs were expressed at low levels. We had considered generating more constructs that included less coding sequence, however, we decided not to pursue this further for the following reasons. Firstly, during the revision process were unable to verify the predicted interactions between RsaE and *psmA1* and *psmA4* using EMSA (now mentioned on page 14) and the EMSAs with RsaE and *psmA2* did not generate convincing results, which is why we did not describe these data in the manuscript. The interaction with *psmA3* is also very weak, and we suspect that the interaction is chaperoned by proteins. Therefore, we were very cautious with interpreting these results. Secondly, the culture supernatant MS data show that RsaE does not impact the levels of PSM α 3 toxins but specifically of PSM α 1 and 4. Therefore, we believe that the way RsaE regulates the operon is more complex way than simply inhibiting the formation of translation initiation complexes. Our current model is that RsaE base-pairing with the *psmA3* SD sequence results in cleavage of the operon transcript (see pages 17 and 18 of the discussion). We now have sequencing data that support this idea (which we would be happy to share with this reviewer if needed), but we still have not completely figured out the precise mechanism. We are currently making several *psmA* mutants to test out theories, but it will take several

years to complete these analyses. Therefore, we would prefer to publish this in a separate article.

11) Did the authors tried toeprint assay or in vitro translation assay?

Response: We do not yet have the expertise for these type of experiments in our lab, so we have not tried this. But this is on our to-do list.

12) It would be of interest to verify whether the modulation of toxin production as an impact on virulence. What about PSM production in *rnc* mutant?

Response: We completely agree that understanding how this impacts virulence is an interesting question and this is something that we are focusing on now. However, we provide strong evidence that this is the case: we have shown that deletion of RNase III almost completely abrogates haemolytic activity, as shown in our culture supernatant haemolysis assays (Figure 6e). This observation is also observed after butanol extraction of the supernatants, which enriches for PSMs (Figure 6f), and thus we hypothesise that RNase III is involved in PSM production. However, further analysis of this is beyond of scope of the current manuscript.

13) The stabilizing effect of RNAIII onto *esxA* mRNA is interesting as the GFP-reporter experiment is convincing (even though EMSA does not show a strong affinity in vitro). However, one would have appreciated to have more physiological data to end the story.

Response: We agree that having more physiological data would be very interesting and we are in the process of establishing microphage infection assays to test this hypothesis. However, such experiments are complex and require optimisation. As such, we will be following this up in subsequent, future publications. Secondly, performing such experiments with an RNAIII deletion mutant will not be informative as it regulates many different toxins. Hence, we would have to make a USA300 strain that has silent mutations in the *esxA* coding sequence. This is something that we are trying now.

14) Also, it raises the question of the potential role of RNase III. This interaction was found by CLASH using RNase III as a bait, which implies degradation of the target. Here, we have a stabilization by RNAIII. What is the *esxA* RNA level in a *rnc* mutant strain?

Response: Exactly, this is a very interesting finding, and it is not immediately obvious how this would work mechanistically. We are keen to figure out the mechanism. However, this part of the project has been plagued by the fact that we were not able to form stable base-pairing interactions *in vitro*. We have tried to do RNA-structure probing on the RNAIII-*esxA* interaction, but the complex formation was too inefficient to detect any changes. However, we feel that the *in vivo* data strongly supports a direct interaction between RNAIII and *esxA*.

To address the question of *esxA* mRNA levels in the RNase III mutant, we have now performed Northern blots and RT-qPCR (see the new Figure 7g and Fig 7h). This shows that deleting RNase III has no significant impact on *esxA* mRNA levels. This is now mentioned on page 15 of the revised manuscript. Although binding by RNase III implies that the RNAIII-*esxA* duplex is likely degraded by this enzyme, we do not believe that this necessarily has to be the case. Like with RsaI and RsaE, we suspect that this nuclease is more likely functioning as an RNA chaperone. But clearly RNase III does not seem to be play a key role in regulating *EsxA* production.

15) Line 181: Change 'reegions' by 'regions'.

Response: we have updated the manuscript accordingly.

16) Line 654: needs rewriting.

Response: we have updated the manuscript accordingly.

Reviewer #3 (Remarks to the Author):

Review for “RNase III CLASH in MRSA uncovers sRNA regulatory networks 1 coupling metabolism to toxin expression”, McKellar et al.

Summary

McKellar et al., describe the role and function of sRNAs in mediating MRSA virulence. Using CLASH, they could identify a number of known, as well as hundreds of novel sRNA-RNA interactions when mimicking the host environment. The authors suggest that the production of small membrane-permeabilizing toxin, whose expression is linked to metabolism, is strongly regulated by sRNA. Among the identified sRNA-RNA interactions, they have uncovered two sRNAs, namely RNAIII and RsaE, which regulate the expression of at least 4 cytolytic toxins, which play a role in MRSA-mediated virulence. Overall, a very interesting study with several novel and potentially clinically relevant findings backed up by sound validations. CLASH data generation, adaptation of the original protocols and data analysis is well done.

General

remarks:

1) The biology of sRNAs in Gram-positive species is not well described (regulation of sRNA-target interaction might differ mechanistically in gram- and gram+). Given the broad readership of Nature Communications, a short introduction into this might be useful.

Response: We have now reworked the Introduction section to include details on sRNA biogenesis and functionality, using RNAIII and the Rsa family to illustrate. We also describe the role of RNA-binding proteins in Gram-positives and compare this to what is known for Gram-negatives.

2) Along the same lines, a general scheme of sRNA biogenesis/functioning in bacteria should be shown, which also includes RNase III, the bait protein chosen for CLASH analysis.

Response: We have now included a new Supplementary Figure 1 showing how sRNAs function and are processed in bacteria.

3) The authors could elaborate in more detail on the clinical potential of targeting sRNAs rather than using broad statements such as “our data has scope for further impact” (last sentence, discussion).

Response: We apologise for the lack of clarity in our writing. Here, we are discussing the fact that our CLASH data likely contains many novel regulatory RNAs ‘hidden’ inside mRNAs and intergenic regions that are worth further characterising in future studies. The “scope for further impact” that we were referring to is not in reference to the clinic, but rather the regulatory RNA field. We have now changed the wording on page 21 to make this clearer.

Specific points:

3) The composition of RPMI and human blood is rather different. It has apparently been shown before (ref 19) that bacteria grown in RPMI have a similar transcriptional profile than in human blood. What means “similar” here? It is surprising that the transcriptional profiles are similar as the composition of blood and RPMI is rather different. Are the authors sure that RPMI is a good choice here especially as there are more physiological media available (DOI:<https://doi.org/10.1016/j.cell.2017.03.023> and others). This reviewer is not an expert in microbiology but the other study (Mediati et al.) uses liquid Mueller-Hinton. Might this be a more physiological medium?

Response: In Mäder *et al*, 2016, the authors grew *S. aureus* in many different conditions, including human plasma and RPMI medium and then examined gene expression using tiling

arrays. Using Principal Component Analysis, they found that cells grown in RPMI and human plasma show very similar gene expression patterns. Afterwards, they examined the expression of individual genes involved in amino acid transport, iron metabolism and virulence factors. From this, they found that these analyses “clearly showed that the eukaryotic cell culture medium RPMI resembles the conditions *S. aureus* is faced with in human plasma”. This is primarily thought to be due to the low concentration of free iron in both media, which is an essential cofactor for *S. aureus* and its absence induces virulent behaviour.

However, we accept and agree with the Reviewer’s fundamental point that RPMI medium cannot perfectly recapitulate the conditions *S. aureus* is faced with in the blood. However, each CLASH library requires a large quantity of cells (~0.5g); performing our experiments in replicates and in two different strains would therefore require impractical quantities of human blood. As such, we used the RPMI medium to mimic this condition as best as possible, and then performed validation experiments in human serum in more manageable volumes.

With regards to Mueller-Hinton medium, this is a nutrient-rich broth and so would not mimic the blood condition. We have used tryptic soy broth (TSB) in our studies as a rich medium reference.

4) The resolution of Fig. 2D should be improved as it is hardly readable.

Response: This was caused by a formatting error. This has now been corrected.

5) Can the authors explain why mRNA-mRNA interactions are predominant in the CLASH data?

Response: we are not entirely sure why this is the case. However, we have further expanded our Discussion of this observation on pages 20 and 21 of the revised manuscript. Here, we have mentioned these may be involved in the stress response and may be biologically meaningful, drawing parallels to Mediati *et al*'s accompanying manuscript and observations in *Listeria monocytogenes*.

6) Independent of RNase III activity, different species of RsaI and RsaE appear in the different media with different transcript lengths. Can the authors speculate how the medium composition might influence transcript length? If so, a highly physiological medium mimicking the composition of human blood as closely as possible is even more important, especially for validation experiments.

Response: We have now mapped the extremities of both RsaI and RsaE using nanopore sequencing. This reviewed that the only difference between the processed and unprocessed forms is the length of the U-stretch in the transcription terminator sequence (see revised Figure 4). Our current model is that RsaE is processed by an exonuclease in TSB. We are unsure of the exact mechanism of this, and future studies will focus on elucidating this. To our knowledge, three 3'-to-5' exonucleases exist in *S. aureus* (PNPase, RNase R, and YhaM) and any of these could be involved. We now discuss this in more detail on page 17. We agree that having a physiological medium that more closely resembles human blood would be great to have. However, it is not immediately clear what medium (other than RPMI) would be the best. We are now trying to improve the CLASH protocol so that we can do the experiments with fewer cells. These developments may allow us to perform CLASH experiments under actual infection conditions, which would be the ideal scenario. By improving the library preparation protocol, we hope to be able to accomplish this soon.

7) In cancer cells, the length of 3' UTR regions of tumor suppressor genes can be shortened so that less miRNAs bind and down-regulate their target. Could a similar mechanism be active in bacteria, i.e. *S. aureus*?

Response: This is an interesting parallel to draw, and we are unsure of the answer. The current paradigm in *S. aureus* sRNA functionality is that a single sRNA molecule binds to a single mRNA molecule, most commonly at the 5' UTR or RBS. Only a few examples of alternative binding exist, such as RNAIII binding *coa* in the coding sequence. Identification and validation of sRNAs binding the 3' UTR of their targets in *S. aureus* will be needed to fully answer this point.

Reviewer comments, second round review -

Reviewer #1 (Remarks to the Author):

The authors have done a great job addressing all our criticism, particularly regarding the processing of RsaE. One intriguing aspect of this work is that RsaE is inactivated by a RNA sponge without much effect on its own transcript levels. This seems mechanistically similar to recently described sponging of the MicF sRNA by the OppX sRNA in enteric bacteria (2022 Molecular Cell, PMID: 35063132). I would recommend this observation be briefly discussed, to make the paper more interesting for researchers concerned with sRNA mechanisms. Other than that, the paper looks good to go.

Reviewer #2 (Remarks to the Author):

In this revised version of the manuscript, the authors have substantially improved their manuscript. Using nanopore sequencing, they clarify the different lengths observed for RsaE. There is a considerable amount of work and CLASH, implemented for the first time in *S. aureus*, revealed novel sRNA-mRNA interactions and shows that RNase III play a role in the sRNA targetome (although not involved in all sRNA-mRNA interaction). Using their approach, the authors uncovered sRNA sponging mechanism (RsaI-RsaE), linked RsaE with toxin production, and RNAIII with EsxA toxin production.

I have only very minor comments:

Line 56: The authors mentioned around 500 transcripts annotated as potential sRNAs. However, a reference is missing. I assume that the authors referred to the SRD since database since some sRNAs appeared as Srn_XXXX in figure 3 for USA300.

Line 244: Since it is now known that RNAIII can be titrated by SprY, the corresponding reference may be added.

Line 510: Please add figure 7f with 7e since they are linked together.

Reviewer #3 (Remarks to the Author):

My queries have all been sufficiently addressed.

Response to Reviewer Comments

We thank all the reviewers for their constructive criticism and very helpful suggestions!

Reviewer #1 (Remarks to the Author): The authors have done a great job addressing all our criticism, particularly regarding the processing of RsaE. One intriguing aspect of this work is that RsaE is inactivated by a RNA sponge without much effect on its own transcript levels. This seems mechanistically similar to recently described sponging of the MicF sRNA by the OppX sRNA in enteric bacteria (2022 Molecular Cell, PMID: 35063132). I would recommend this observation be briefly discussed, to make the paper more interesting for researchers concerned with sRNA mechanisms. Other than that, the paper looks good to go.

Response: We agree. We now discuss the MicF-OppX interaction briefly on pages 16 and 17 of the revised manuscript. A similar mechanism was also proposed for the regulation of RNAIII activity by SprY, which we now also mention in the same paragraph.

Reviewer #2 (Remarks to the Author): In this revised version of the manuscript, the authors have substantially improved their manuscript. Using nanopore sequencing, they clarify the different lengths observed for RsaE. There is a considerable amount of work and CLASH, implemented for the first time in *S. aureus*, revealed novel sRNA-mRNA interactions and shows that RNase III play a role in the sRNA targetome (although not involved in all sRNA-mRNA interaction). Using their approach, the authors uncovered sRNA sponging mechanism (RsaI-RsaE), linked RsaE with toxin production, and RNAIII with EsxA toxin production. I have only very minor comments: Line 56: The authors mentioned around 500 transcripts annotated as potential sRNAs. However, a reference is missing. I assume that the authors referred to the SRD since database since some sRNAs appeared as Srn_XXXX in figure 3 for USA300.

Response: Correct, we were referring to the srd database. We now reference the relevant manuscript on page 3 of the revised manuscript.

Line 244: Since it is now known that RNAIII can be titrated by SprY, the corresponding reference may be added.

Response: Agreed. We now briefly discuss the SprY-RNAIII interaction on page 17 of the revised manuscript.

Line 510: Please add figure 7f with 7e since they are linked together.

Response: We now reference 7f and 7e together in the text.

Reviewer #3 (Remarks to the Author): My queries have all been sufficiently addressed.

Response: Thank you!